# Proteomic signatures of the *APOE ε4* and *APOE ε2* genetic variants and Alzheimer's disease

Lina Lu [1] ✉, Alexa Pichet Binette[1,2,3], Ines Hristovska[1], Shorena Janelidze [1], Bart Smets[4], Irene Cumplido-Mayoral[1], Aparna Vasanthakumar[5], Britney Milkovich[5], The Global Neurodegeneration Proteomics Consortium (GNPC)*, Lijun An [6], Rik Ossenkoppele[1,7,8], Varsha Krish[9], Farhad Imam [9], Sebastian Palmqvist [1,10], Jacob W. Vogel [6], Erik Stomrud[1,10], Oskar Hansson [1] & Niklas Mattsson-Carlgren [1,10] ✉

The *APOE* locus is the strongest genetic factor for Alzheimer's disease, with ε4 increasing and ε2 decreasing risk, yet the basis of these opposing effects remains unclear. Here we performed a multicohort proteomic analysis across plasma and cerebrospinal fluid in GNPC, BioFINDER-2, ADNI, UK BioBank, and PPMI. *APOE*-associated protein alterations are detectable before amyloid pathology and remain stable across age and disease progression. *APOE2*-associated proteins were enriched in pathways related to cellular maintenance and anti-inflammatory processes. By contrast, *APOE4* showed a limited set of upstream mediators linked to cell-cycle and oligodendrocyte precursor cell biology, and a broader group of proteins reflecting vascular, immune, and proteostatic dysfunction shaped by downstream pathology. Comparative analyses highlighted allele-specific mediators and oppositely regulated proteins contributing to differential disease risk. Together, these findings reveal that *APOE2* and *APOE4* shape Alzheimer's disease risk through distinct molecular architectures and identify candidate biomarkers and targets for allele-specific interventions.

The apolipoprotein E (*APOE*) gene is the strongest genetic factor for sporadic Alzheimer's disease (AD), with three main alleles: ε2, ε3, and ε4. Compared to ε3, the ε4 allele (*APOE4*) increases AD risk in a dose-dependent manner, approximately 2- to 3-fold in people with one ε4 allele and up to 12-fold in ε4 homozygotes[1,2] and is linked to amyloid β (Aβ) aggregation[3–5]. By contrast, ε2 (*APOE2*) carriers have a reduced AD risk[6–8] and Aβ burden[9–11] and delayed onset in high-penetrance AD mutation carriers[12].

Recent proteomic studies, including those from the Global Neurodegeneration Proteomics Consortium (GNPC) cohort, have reported widespread *APOE4*-associated protein alterations in plasma, cerebrospinal fluid (CSF), and brain tissue in humans[13,14]. These findings have

expanded our understanding of *APOE4* biology and its pleiotropic effects, particularly in immune regulation. However, many previous studies overlooked early Aβ pathology in cognitively unimpaired (CU) individuals and have not clearly distinguished proteins that likely reflect *APOE4*-related effects from those driven by downstream pathological processes. While some studies have adjusted for *APOE4* status in identifying AD-associated signatures[15,16] or vice versa[14], such approaches may not adequately address the dominance of proteomic changes as well as unique and shared molecular signatures in relation to genotype and disease status. Then, while *APOE4* has been extensively investigated, *APOE2* has received comparatively little systematic attention, despite its well-established protective role, leaving key

mechanistic differences between *APOE4* and *APOE2* in AD risk largely unresolved. Finally, most existing reports relied on single cohorts, tissues, or proteomic platforms, limiting generalizability.

In this Article, we address these gaps through a multicohort, cross-platform proteomic study of *APOE* isoforms in plasma and CSF spanning five well-characterized cohorts (Fig. 1). We systematically compared molecular signatures of *APOE4* and *APOE2* carriers, anchoring analyses on AD diagnosis and Aβ status to distinguish dominant genotype effects from disease-driven protein alterations within an upstream (*APOE* → protein → AD/Aβ) versus downstream (*APOE* → AD/Aβ → protein) mediation framework. By stratifying by age, Aβ status, and clinical stage, we assessed how *APOE* effects emerge and evolve across aging and disease progression. We further grouped *APOE*-related proteins into upstream mediators, AD-mediated proteins, and allele-specific signatures to resolve how distinct molecular mechanisms might explain the divergent AD risk. To refine upstream candidates, we examined their associations with downstream AD phenotypes, assessed replication in CSF, quantified reproducibility across proteomic platforms and tissues, and evaluated central nervous system (CNS) relevance through genetic support from AD genome-wide association studies (GWAS) and spatial co-expression with *APOE* in the human brain. We further tested the temporal stability of *APOE*–protein associations using longitudinal CSF data. Together, this framework delineates distinct *APOE4*- and *APOE2*-associated signatures linked to AD pathology, clarifies their regulation by age and disease stage, and highlights genetically supported, brain-relevant proteins as robust candidates for biomarkers and therapeutic targets.

## Results

### The *APOE*-plasma protein signature and its role in clinical AD in the GNPC cohort

We began by analyzing the GNPC plasma dataset (*N* = 3,289; aged 27–90 years, median 76 years; Supplementary Table 1); 2,069 (62.9%) CU individuals and 1,220 (37.1%) patients with AD dementia were included. *APOE2*-associated proteins were identified by comparing *APOE2* carriers (12 ε2/ε2; 321 ε2/ε3) to ε3/ε3 individuals (*N* = 1,679). To assess the effect of *APOE4* in AD, *APOE4* carriers (1,013 ε3/ε4; 172 ε4/ε4) were compared with ε3/ε3 individuals.

#### *APOE2*-related proteomic alterations are widespread and largely independent of AD diagnosis.
In *APOE2* carriers, 192 proteins were significantly altered compared to ε3/ε3 carriers (Model type 1), with the strongest upregulations in UNG, VPS29, and BIRC2 and the most notable downregulations in BCDIN3D and S100A13 (Fig. 2a). By contrast, 2,896 proteins were significantly altered in AD cases compared to CU individuals (Model type 2), including robust increases in ACHE

and MMP8 (Fig. 2b). After adjustment for each other (Model type 3), the vast majority of *APOE2*- (92%) and AD-associated (98%) proteins remained significant, and the effect sizes of *APOE2* and AD were largely unchanged (Supplementary Fig. 1a,b), indicating largely independent molecular signatures.

Interestingly, 99 proteins (Fig. 2c, red) were uniquely associated with *APOE2* without being associated with AD (for example, VPS29 and BCDIN3D; Supplementary Fig. 2 and Extended Data Fig. 1), suggesting broader biological pathways beyond AD pathophysiology. Similarly, 2,766 proteins were AD specific (Fig. 2c, blue). Only 65 proteins were shared by both *APOE2* and AD (Fig. 2c, black), indicating a limited but biologically relevant overlap.

#### *APOE2* signatures emerge early and remain stable across age and disease status.
We next explored whether *APOE2*-related signatures emerge before clinical onset; 140 of the 192 *APOE2*-associated proteins (73%) were already significantly dysregulated in CU individuals, with effect sizes highly correlated with those in the full cohort (*r* = 0.874, *P* < 2.2 × 10⁻¹⁶) (Fig. 2d). Most alterations were evident across both younger (27–74 years) and older (75–90 years) CU subgroups, indicating that *APOE2*-linked signals appear early in life and remain relatively stable across aging. Only the *APOE2*-specific protein (without being associated with AD) VPS29 showed age-modulated effects, with elevated levels already detectable in early adulthood and showing stronger upregulation in younger than in older *APOE2* carriers, especially in the CU group (Fig. 2e).

In individuals with AD dementia, 51 proteins were significantly associated with *APOE2*, 46 of which overlapped with associations observed in CU individuals and in the full cohort, reflecting a persistent effect across disease stages. No significant *APOE2*–diagnosis interactions were detected, indicating that *APOE2* effects on these proteins are robust and not significantly amplified or diminished by disease status.

To probe the functional relevance of these preclinical changes, we mapped the 140 early dysregulated proteins to Reactome pathways and trained a biologically informed neural network (BINN) to predict AD. The model achieved consistent performance (training accuracy = 0.73; testing accuracy = 0.69) and highlighted UBB, AKT2, and APOB as top contributors, enriched in drug metabolism, intercellular communication, RNA processing, and programmed cell death pathways (Fig. 2f). Enrichment analysis across 81 whole-body cell types indicated predominant involvement of peripheral tissues (Supplementary Table 2). Together, these findings suggest that *APOE2* establishes systemic buffering against cellular stress early in life, with effects that are preserved with age and stable even after AD onset.

---

**Fig. 1 | Overview of cohorts and analytical framework.** We performed different targeted analyses based on the available data for each cohort. The GNPC cohort (*N* = 3,289 individuals; plasma, SomaLogic 7K) was used to identify proteins associated with *APOE4* or *APOE2* and their roles in clinical AD (CU versus AD dementia). Replication was performed in BioFINDER-2 (plasma, SomaLogic 7K). Further validation was conducted in CSF using ADNI (SomaLogic 7K, TMT-MS) and BioFINDER-2 (OLINK). To enable a systematic four-way comparison of *APOE*-associated proteomic signatures across tissue (plasma versus CSF) and platform (SomaLogic versus OLINK), we incorporated plasma OLINK data from the population-based cohort UKBB. Longitudinal *APOE*–protein associations were evaluated in PPMI (CSF, OLINK), providing exploratory insights into temporal stability independent of Aβ pathology. Specifically, differential abundance analysis identified *APOE* (*APOE4* or *APOE2*)-associated proteins and AD diagnosis (or Aβ status)-associated proteins, resulting in different groups of proteins specifically associated with *APOE* or AD or jointly associated with both. Proteins associated with *APOE* were further tested and categorized according to whether they showed stronger evidence for upstream versus downstream mediation using two mediation models, upstream mediation model: *APOE* → protein → AD diagnosis or Aβ status, and downstream mediation

model: *APOE* → AD diagnosis or Aβ status → protein. Stratified analyses by AD diagnosis or Aβ status were conducted to investigate in depth the changes of *APOE*–protein associations. Age-stratified analyses were performed only in CU or Aβ⁻ individuals to investigate how early *APOE*–protein associations change with age. *APOE4* and *APOE2* effects were analyzed separately, with direct comparisons across six genotype groups (ε2/ε2, ε2/ε3, ε2/ε4, ε3/ε3, ε3/ε4, ε4/ε4) to evaluate allele dominance. Functional annotation included cell-type enrichment, GO enrichment, BINNs-enriched Reactome pathway analysis, and protein–protein interaction (PPI) analysis. Associations with 5 AD phenotypes were evaluated to link *APOE*-related proteins to disease features, including tau-PET, Aβ-PET, cortical thickness, cognition (Mini-Mental State Examination (MMSE) and modified Preclinical Alzheimer Cognitive Composite (mPACC)). To assess the robustness of key findings and support their central relevance, we evaluated genetic evidence from AD-associated SNPs in coding genes and examined spatial transcriptomic co-expression with *APOE* in the human brain. Matching superscript numbers indicate which analyses were conducted in which cohorts. A summary of the cohort analysis can be found in Supplementary Fig. 3b. Figure created in BioRender; Lu, L. https://BioRender.com/vvnh8bs (2026).

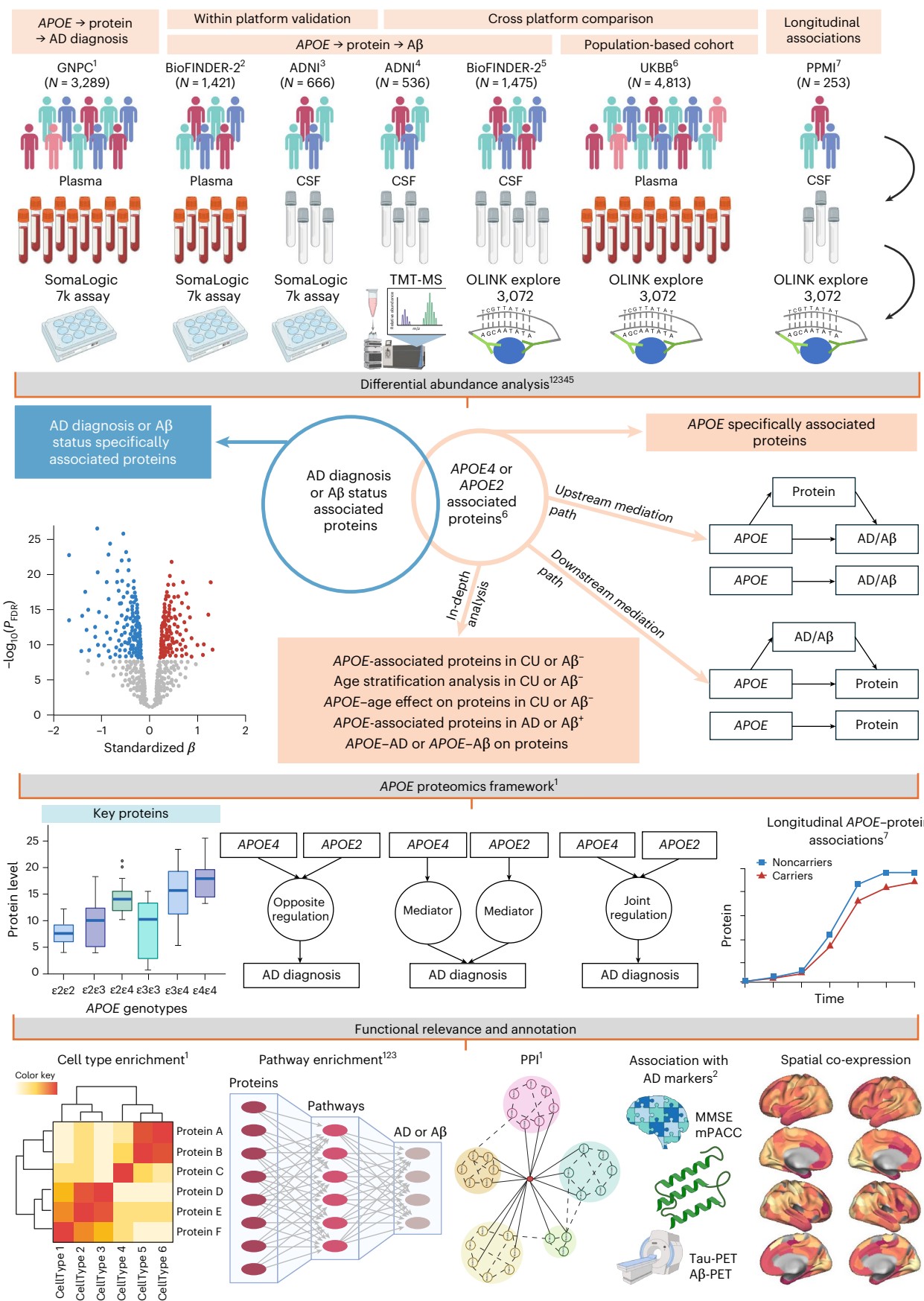

**Mediation analysis maps proteomic alterations relative to disease onset in *APOE2* carriers.** Although *APOE2* effects on these proteins appeared stable with age and disease onset, not all early alterations drive progression, and some changes in AD may simply reflect differences in pathology burden between carriers and noncarriers. To disentangle proteins explaining the *APOE2*–AD association from those reflecting downstream processes, we applied an upstream–downstream mediation framework anchored on disease onset. Among the 192 *APOE2*-associated proteins, 54 proteins (for example, UNG, PCLAF, and ARL2) showed stronger statistical support for the *APOE2* → protein → AD pathway (partial mediation proportions 6–92%) than for the reverse (Fig. 2g,h), indicating that they statistically account for part of the *APOE2*–AD association, consistent with upstream effects. By contrast, 21 proteins (for example, FMR1, PEBP1, and CLUAP1) showed stronger support for the *APOE2* → AD → protein pathway, suggesting that their levels are more consistent with downstream remodeling than upstream effect.

**Mechanistic heterogeneity of *APOE2*-linked proteins.** We further subdivided all 192 *APOE2*-associated proteins into four mechanistic categories based on mediation and diagnostic associations (Fig. 2h). Clustering with linear discriminant analysis (LDA) supported distinctiveness (Fig. 2i). Cell-type enrichment, pathway, and protein–protein interaction (PPI) analyses further supported biological differences between these four groups of proteins (Fig. 2j and Supplementary Table 3):

1. AD mediator proteins (*N* = 54, *APOE2* → protein → AD): Upregulated proteins (*N* = 12, for example, UNG, EIF4G1, UBA2, SIRT1, EIF4G1, and SIRT1) were enriched in undifferentiated cells, adipocytes, erythroid cells, proximal enterocytes, and MeninFibroblast cells (in the Human Brain Vascular (BBB) atlas) and linked to gene expression/translation, protein and epigenetic regulation, energy balance, and DNA repair, suggesting improved cellular maintenance. Downregulated proteins were enriched in oligodendrocytes (in the Religious Orders Study and Memory and Aging Project (ROSMAP) atlas), tied to complex remodeling, immune/inflammatory signaling, lipid metabolism, blood pressure regulation, development, oxidative stress response,

and histone deacetylation, suggesting suppressed stress and immune pathways.

2. AD-mediated proteins (*N* = 21, *APOE2* → AD → protein): Upregulated proteins (*N* = 4; for example, UBB and COPS8) were enriched in astrocytes and related to small protein conjugation, neurodevelopment, and p53-mediated apoptosis, indicating maladaptive activation. Downregulated proteins were enriched in NK cells and club cells and related to pre-messenger-RNA processing, cilium and cell projection assembly, and ERK signaling regulation, indicating loss of essential cellular functions.

3. *APOE2*-specific proteins (*N* = 98): Upregulated proteins (*N* = 28, for example, PHGDH, DLL1, SHH, and FOXO1) were enriched in peripheral cell types such as alveolar cells, with pathway annotations modestly linked to neurodevelopment, vascular development, and stress response. Downregulated proteins were associated with ovarian stromal cells and enriched in processes including catabolism, cell cycle control, immune recombination, and mitochondrial function, suggesting subtle systemic shifts in developmental and metabolic regulation.

4. Nonspecific proteins (*N* = 19): Close to the *APOE2*-specific cluster in LDA space (Fig. 2i), indicating *APOE2*-related functionality. Upregulated proteins were modestly enriched in ubiquitination, vesicle transport, developmental signaling, and stress responses, supporting a minor activation of differentiation and repair-related pathways. Downregulated proteins were associated with mitotic and membrane organization, suggesting reduced proliferative and structural activity.

**_APOE4_-related proteins are easily invaded by AD and produce a pathological cascade reaction.** To complement the *APOE2* analyses, we characterized *APOE4*-associated proteomic alterations. A total of 357 proteins (for example, SPC25, LRRN1, S100A13, TBCA, and NEFL) were significantly altered in *APOE4* carriers, while 3,785 were associated with AD (Fig. 3a–c). Unlike *APOE2*, *APOE4* profiles were more sensitive to disease: effect sizes before and after adjustment for AD were only moderately correlated ($r = 0.654$, $P < 2.2 \times 10^{-16}$), and 39% of *APOE4* associations lost significance, whereas AD signals remained largely unchanged after adjusting for genotype (Supplementary Fig. 1c,d);

**Fig. 2 | *APOE2* plasma protein signature in GNPC. a**, Volcano plot for proteins associated with *APOE2* without adjusting for AD diagnosis (*N* = 2,012 individuals, linear models adjusted for age, sex, mean protein level, and cohorts), with red representing significant association after FDR correction. On the *y* axis, $-\log_{10}$(FDR) above 300 was set to 300 for a better visualization. **b**, Volcano plot shows proteins associated with AD diagnosis without adjusting for *APOE2* (*N* = 2,012 individuals, linear models adjusted for age, sex, mean protein level, and cohorts), with blue representing significant association after FDR correction. **c**, UpSet plot shows the number of proteins associated with *APOE2* or clinical AD with or without adjusting for each other, blue indicating AD-specific proteins, red indicating *APOE2*-specific proteins, black indicating the number of proteins independently associated with both. **d**, Scatter plot shows *APOE2*'s effect size on proteins without adjusting AD diagnosis in the whole cohort (*x* axis, *N* = 2,012 individuals) versus in CU subgroup (*y* axis, *N* = 1,446 individuals) for each protein. Effect sizes were derived from linear models adjusted for age, sex, mean protein level, and cohorts. Spearman correlation was assessed with a two-sided test. Proteins highlighted in red are significantly associated with *APOE2* in both the whole cohort and the CU subgroup. **e**, VPS29 protein levels are plotted against age and stratified by *APOE2* status (ε2⁺ versus ε3/ε3) and clinical diagnosis (CU versus AD). Each line represents a group. Solid lines indicate LOESS (locally estimated scatterplot smoothing)-fitted mean VPS29 levels across age, and shaded bands indicate the 95% confidence intervals around the fitted mean. **f**, BINN-enriched pathway analysis for early dysregulated proteins in *APOE2* carriers; the darker the color, the more important the protein or pathway in predicting AD dementia diagnosis. More features are hidden in the sink for a better visualization. **g**, The heat map summarizes mediation effects and statistical significance for *APOE2*-associated proteins (75 in total were involved in either pathway) across the two mediation pathways: the upstream pathway (top row; *APOE2* → protein → AD, red

label) and the downstream pathway (bottom row; *APOE2* → AD → protein, blue label). Cell colors represent the proportion of mediation. Protein labels on the *x* axis are color coded by the dominant mediation direction, with red indicating stronger upstream mediation and blue indicating stronger downstream mediation. Bold labels denote complete mediation within the dominant pathway, whereas nonbold labels denote partial mediation. Asterisks indicate statistical significance (*+ FDR-corrected significance). For clarity, only selected proteins with the largest mediation effects from each pathway are shown; full results are provided in the Source data. **h**, Subdivision of *APOE2*-associated proteins based on the association between proteins, *APOE2*, and AD. Note that overlapping proteins are preferentially assigned to mediation categories. **i**, The LDA score plot shows the projection score of each group of subdivided proteins in the discriminant direction. Proteins are colored by their assigned groups. **j**, The integrative matrix summarizes differential regulation (red for upregulated and blue for downregulated proteins in *APOE2* carriers), cell-type enrichment based on scaled RNA expression, and functional characterization of each protein. Cell types from the ROSMAP atlas are labeled in black on the *x* axis, while those from the BBB atlas are labeled in red. Gray boxes indicate nominal significance ($P < 0.05$), and black boxes indicate FDR-corrected significance ($P_{FDR} < 0.05$) in cell-type enrichment analysis. GO biological process terms associated with each protein are grouped into broader representative categories; small red boxes indicate the involvement of a given protein in the corresponding process. Cell-type enrichment and GO enrichment analyses were one-sided, with Benjamini–Hochberg adjustment for multiple comparisons. PPIs are annotated using STRING database interactions with a confidence score ≥ 0.7. The number of interactions per protein is shown as a heat map, and direct interactions between proteins are represented by lines, color coded according to their assigned cluster. See Source Data Fig. 2 for detailed statistical summary.

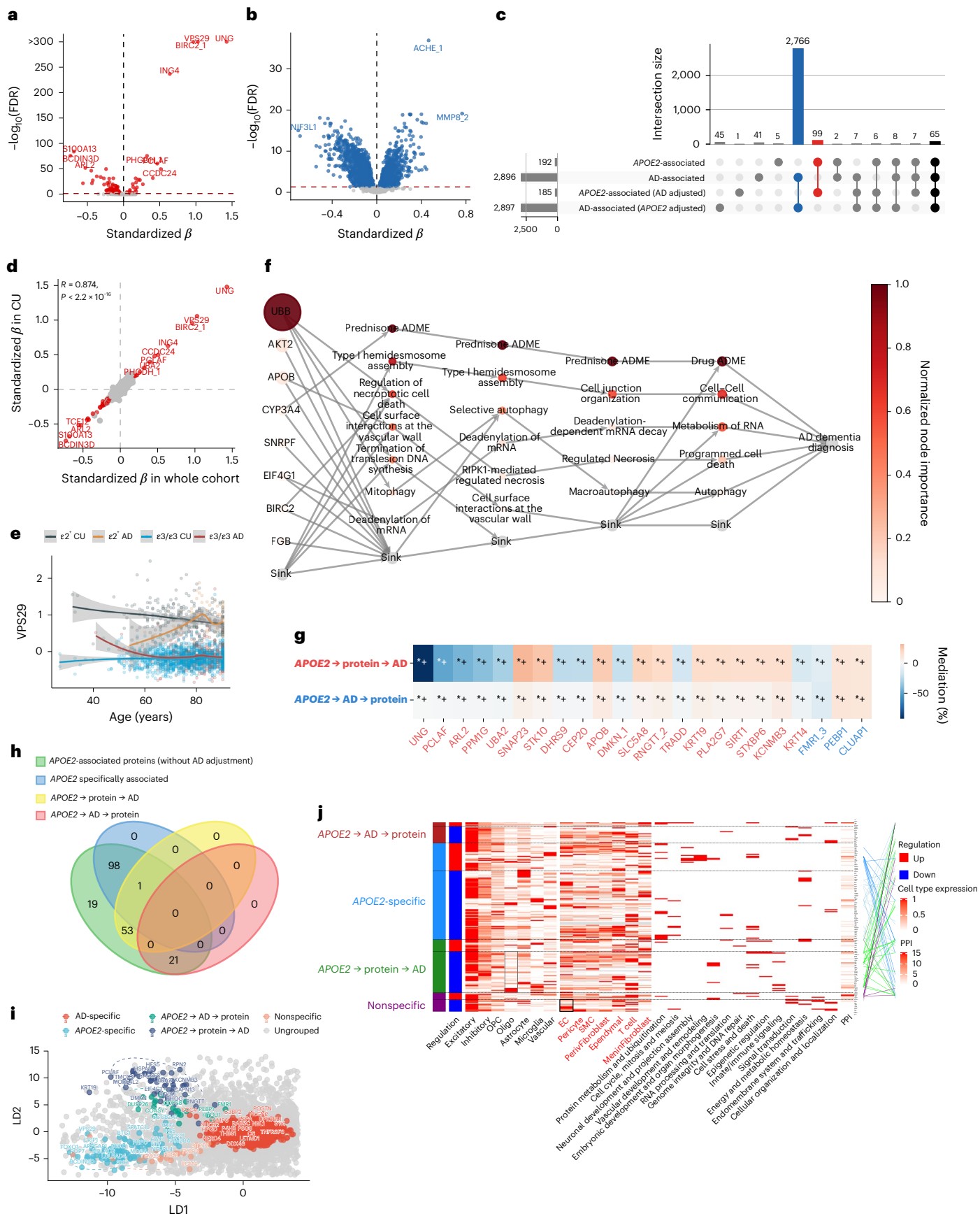

75 proteins were shared between *APOE4* and AD, potentially bridging genetic risk to disease manifestation (Fig. 3c, black).

Despite this sensitivity, 45% of *APOE4* associations were already detectable in CU individuals, stable across aging, and in part amplified in patients with AD (for example, SPC25 and TBCA) (Fig. 3d and Supplementary Fig. 1e,f). Among those early-stage changes, BINNs highlighted key predictors (for example, AKT2, NOTCH1, RIPK1, YWHAQ, BIRC2, and SPC25), enriched in pathways related to gene regulation, metabolism, and DNA repair, with cell-type enrichment pointing to oligodendrocyte precursor cells (OPCs) and inhibitory neurons (Fig. 3e). Comparison with AD-related proteins in ε3/ε3 carriers revealed 66 overlaps, the majority (48/66) of which showed opposite directions of change (Fig. 3f), suggesting distinct or compensatory processes rather than simply latent manifestations of downstream pathology.

Mediation analysis (Fig. 3g) identified 16 proteins (for example, SPC25, LRRN1, ARL2, and NEFL) with stronger *APOE4* → protein → AD effects (up to 49%), suggesting potential upstream contributions, but the majority (200 proteins) followed *APOE4* → AD → protein patterns. Thus, *APOE4* appears to prime the proteome for dysregulation, with its dominant effects unfolding through pathology-driven cascades, contrasting with the more autonomous signatures of *APOE2* (Fig. 3g).

Similarly, we stratified *APOE4*-associated proteins into four mechanistic groups (Fig. 3h–j and Supplementary Table 4). Upregulated AD mediator proteins were enriched in OPCs and smooth muscle cells, related to metabolism (DLD) and mitotic spindle checkpoint control (SPC25), and downregulated proteins were enriched in cell cycle checkpoints (ZW10, ARL2) and spliceosome assembly and RNA capping (SNRPF), suggesting increased proliferation with reduced regulatory control. AD-mediated proteins were upregulated in microglial cells, OPCs, and vascular cells, enriched for vesicle fusion, synaptic signaling, platelet activation, and structural remodeling, and downregulated in protein transport, glycan metabolism, immune chemotaxis, and autophagy, indicating enhanced exocytosis and vascular activity with impaired immune trafficking. *APOE4*-specific proteins were upregulated in macrophages and fibroblasts, linked to immune differentiation, synapse/junction formation, and Notch/IL-6 signaling, and downregulated in protein catabolism, metabolism, mitotic regulation, and RNA processing, suggesting immune and structural activation with impaired cellular homeostasis. Nonspecific proteins showed broad upregulation in developmental, morphogenetic, and signaling pathways but were downregulated in glucose metabolism, hormone secretion, and immune-endocrine functions, reflecting widespread developmental activation with reduced metabolic and immune regulation.

## *APOE4* and *APOE2*: specificity, offset, and synergy

We next compared *APOE2* and *APOE4*-associated proteins, including ε2/ε4 heterozygotes to assess allelic dominance. *APOE2*- and *APOE4*-associated proteins were largely nonoverlapping (Fig. 4a). For example, SPC25 and LRRN1 mediated AD risk in *APOE4* carriers but were not associated with *APOE2*, showing a clear ε4 gene dose–response pattern in CU individuals, with ε2/ε4 carriers resembling ε3/ε4, indicating ε4 dominance (Fig. 4b). By contrast, 26 proteins (for example, UNG and PCLAF) were specific mediators for *APOE2*, showing ε2-dominant patterns even in ε2/ε4 individuals. ARL2 and OTULIN emerged as shared mediators in both *APOE2* and *APOE4* models, suggesting convergence on mitochondrial and immune pathways.

Allele-specific mediation directionality was observed. For instance, SNRPF acted upstream of AD in *APOE4* carriers but downstream in *APOE2* carriers, possibly reflecting detrimental effects of *APOE4*-associated downregulation on RNA splicing[17] and compensatory AD-induced upregulation in *APOE2* carriers. More broadly, 13 *APOE2*-downregulated proteins (for example, SNAP23 and APOB) were upstream mediators in *APOE2* but downstream in *APOE4*, enriched in lipid regulation and organelle fusion, and primarily expressed in macrophages, trophoblasts, and hepatocytes, suggesting early modulation of inflammatory and metabolic responses by *APOE2*. This upstream suppression may help prevent the activation of downstream AD-related processes seen in *APOE4*, pointing to a potential core mechanism by which *APOE2* confers protection against AD.

Beyond mediators, 77 proteins were significantly altered by both *APOE2* and *APOE4* in CU individuals, enriched in autoubiquitination, RNA modification, and autophagy, and expressed in keratinocytes

---

**Fig. 3 | *APOE4* plasma protein signature in GNPC. a**, Volcano plot for proteins associated with *APOE4* without adjustment for AD diagnosis (*N* = 2,864 individuals, linear models adjusted for age, sex, mean protein level, and cohorts), with red representing significant association after FDR correction. On the *y* axis, $-\log_{10}$(FDR) above 300 was set to 300 for a better visualization. **b**, Volcano plot shows proteins associated with AD diagnosis without adjustment for *APOE4* (*N* = 2,864 individuals, linear models adjusted for age, sex, mean protein level, and cohorts), with blue representing significant association after FDR correction. **c**, UpSet plot shows the number of proteins associated with *APOE4* or AD with or without adjusting for each other, blue indicating AD-specific associated proteins, red indicating *APOE4*-specific associated proteins, black indicating the number of proteins jointly associated with both. **d**, Scatter plot shows *APOE4*'s effect size on protein without adjusting AD diagnosis in the whole cohort (*x* axis, *N* = 2,864 individuals) versus in CU subgroup (*y* axis, *N* = 1,751 individuals) for each protein. Effect sizes were derived from linear models adjusted for age, sex, mean protein level, and cohorts. Spearman correlation was assessed with a two-sided test. Red represents proteins associated with *APOE4* in both the whole cohort and in the CU group. **e**, BINN-enriched Reactome pathway analysis for proteins associated with *APOE4* in both the whole cohort and in CU. The darker the dot, the more important the protein and the pathway in the deep learning model predicting AD dementia diagnosis. More features are hidden in the sink for a better visualization. **f**, The scatter plot shows the effect of *APOE4* on proteins in CU subgroup (*x* axis, *N* = 1,751 individuals) versus the effect of AD on proteins in ε3/ε3 carriers (*y* axis, *N* = 1,679 individuals). Only proteins associated with *APOE4* in CU individuals and with AD diagnosis in ε3/ε3 carriers are visualized. Red indicates the same effect direction, while blue indicates an opposite effect direction. **g**, The heat map summarizes mediation effects and statistical significance for *APOE4*-associated proteins (216 in total were involved in either pathway) across the two mediation pathways: the upstream pathway (top row; *APOE4* → protein → AD, red label) and the downstream pathway (bottom row; *APOE4* → AD → protein, blue label). Cell colors represent the proportion of mediation. Protein labels on the *x* axis are color coded by the dominant mediation direction, with red indicating stronger upstream mediation and blue indicating stronger downstream mediation. Bold labels denote complete mediation within the dominant pathway, whereas nonbold labels denote partial mediation. Asterisks indicate statistical significance (*+ FDR-corrected significance). For clarity, only selected proteins with the largest mediation effects from each pathway are shown; full results are provided in the Source data. **h**, Venn plot shows the number of proteins in each category. Note that overlapping proteins are preferentially assigned to mediation categories. **i**, The linear discriminant score plot shows the projection score of all tested proteins in the discriminant direction. Proteins are colored by their assigned groups. **j**, The integrative matrix summarizes differential regulation (red for upregulated and blue for downregulated proteins in *APOE4* carriers), cell-type enrichment based on scaled RNA expression, and functional characterization of each protein. Cell types from the ROSMAP atlas are labeled in black on the *x* axis, while those from the BBB atlas are labeled in red. Gray boxes indicate nominal significance (*P* < 0.05), and black boxes indicate FDR-corrected significance ($P_{FDR}$ < 0.05) in cell-type enrichment analysis. GO biological process terms associated with each protein were grouped into broader representative categories; small red boxes indicate the involvement of a given protein in the corresponding process. Cell-type enrichment and GO enrichment analyses were one-sided, with Benjamini–Hochberg adjustment for multiple comparisons. PPIs are annotated using STRING database interactions with a confidence score ≥ 0.7. The number of interactions per protein is shown as a heat map, and direct interactions between proteins are represented by lines, color coded according to their assigned cluster. See Source Data Fig. 3 for detailed statistical summary.

and club cells. Surprisingly, only a minority showed opposite regulation (Fig. 4c), with ε4-dominant (for example, PHGDH and FOXO1) or ε2-dominant (for example, VPS29 and BIRC2) patterns (Fig. 4d). Most shared proteins (for example, S100A13, BCDIN3D, TBCA, and

ARL2) changed in the same direction, with ε2/ε4 carriers again resembling ε4/ε4 (Fig. 4e). These findings indicate that *APOE2* and *APOE4* do not simply act as mirror images by regulating the same proteins in opposite directions. Instead, their proteomic effects are shaped by

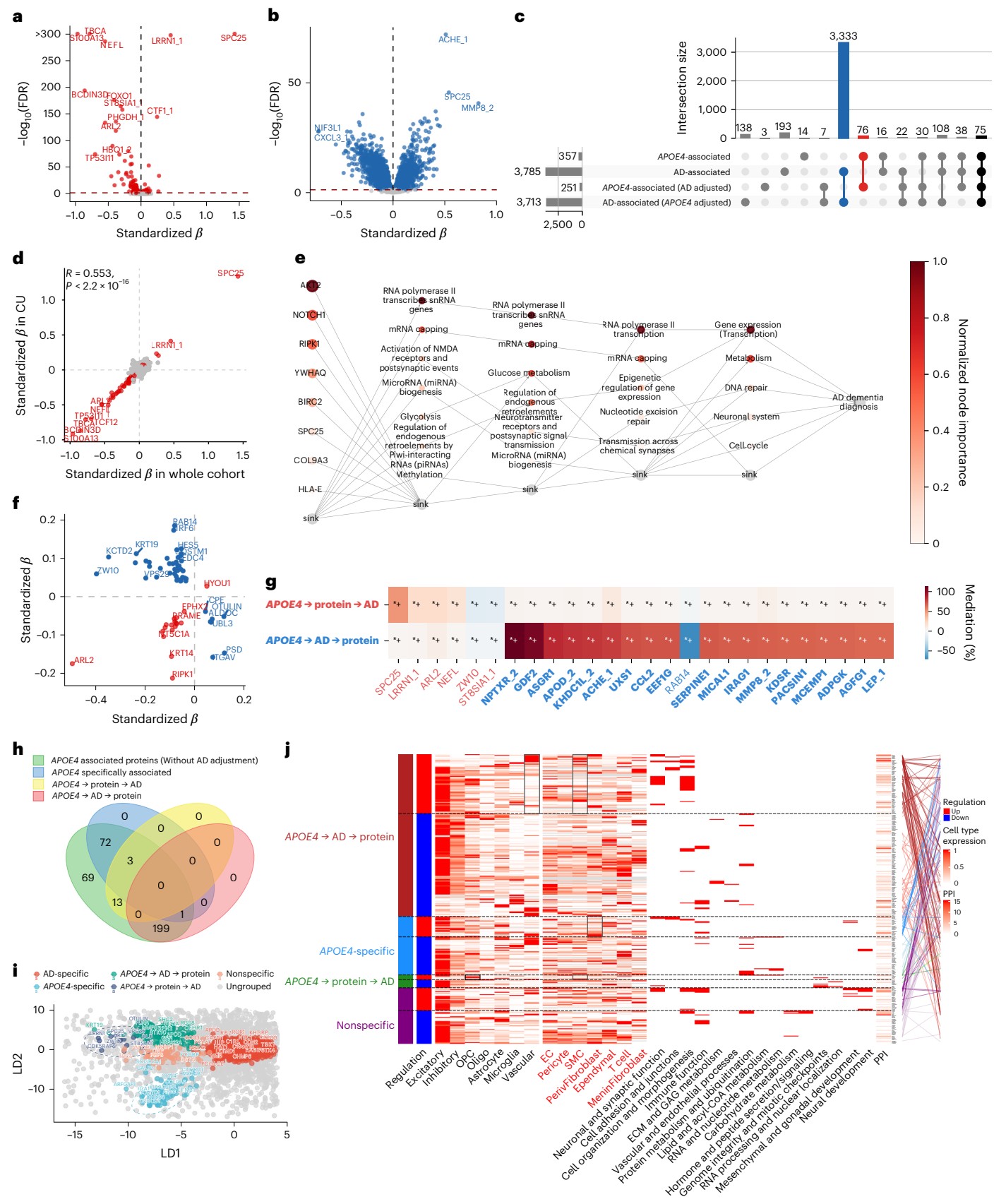

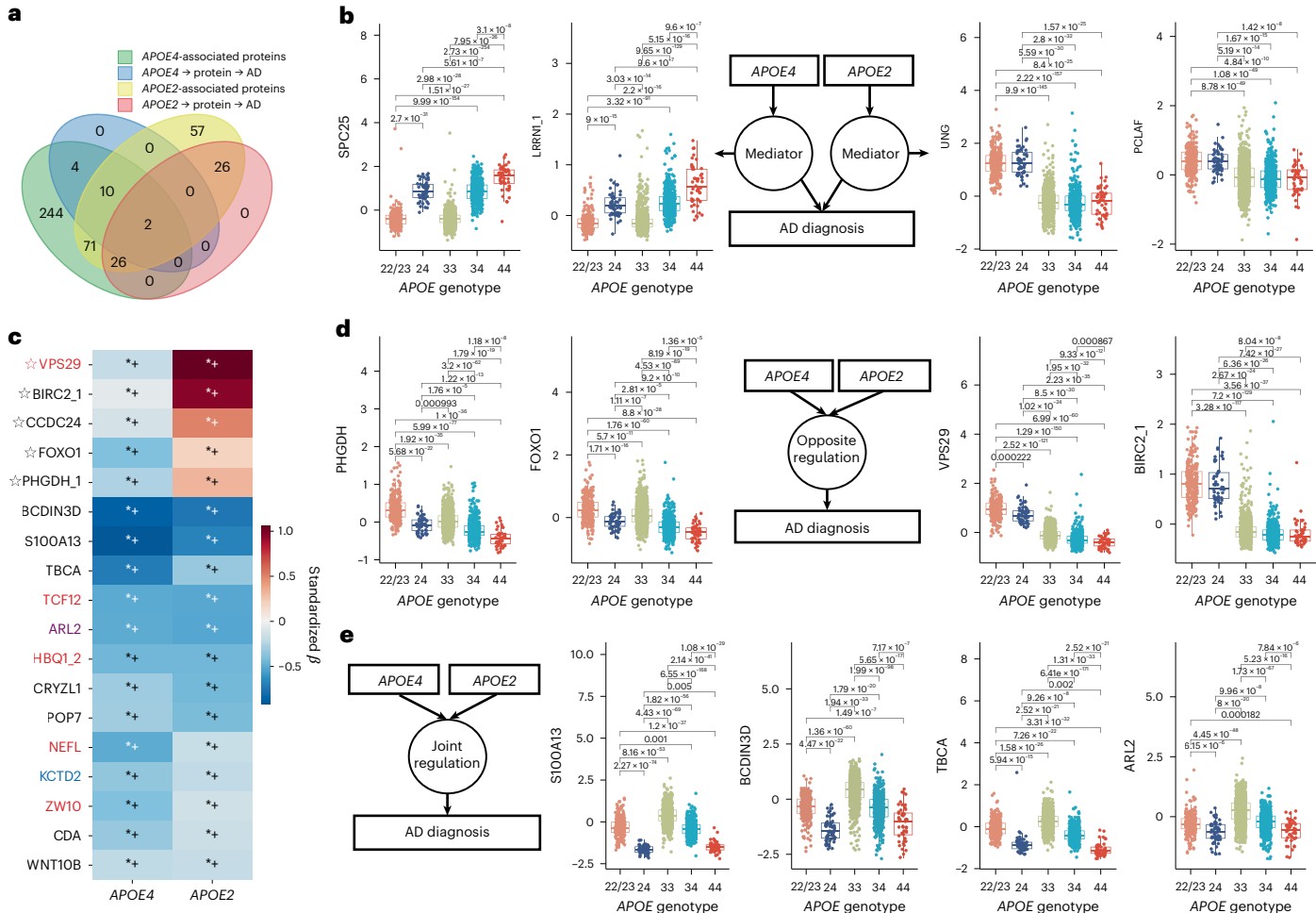

**Fig. 4 | Asymmetric proteomic signatures of *APOE4* and *APOE2*. a**, Venn plot shows proteins mediating *APOE4*'s effect of AD diagnosis and their association with *APOE2*, and proteins mediating *APOE2*'s effect on AD diagnosis and their association with *APOE4*. **b,d,e**, Boxplots show protein level comparison between different *APOE* genotype groups for *APOE4* or *APOE2* related mediators (**b**, left, for ε4 dominated mediators: SPC25 and LRRN1; right, for ε2 dominated mediators: UNG and PCLAF), and for early dysregulated proteins that are opposingly (**d**) or similarly (**e**) regulated by both alleles in the CU group (*N* = 2,069 individuals). On the box plots, the *y* axis represents residual protein levels after adjusting for age, sex, mean protein level, and cohorts. The center line of each box indicates the median (50th percentile); the lower and upper edges of the box represent the 25th and 75th percentiles, respectively. Whiskers extend to the most extreme values within 1.5 times the interquartile range (IQR); data points beyond this range are considered outliers and have been excluded from the plot

display. The *x* axis represents *APOE* genotype; ε2/ε2 and ε2/ε3 carriers were merged into the "22/23" group due to a small sample size of ε2/ε2 carriers. Group differences were assessed using two-sided Welch's *t*-tests, and *P* values were adjusted for multiple comparisons using the Holm–Bonferroni method. Pairwise comparisons were performed across all groups, but only significant adjusted *P* values are shown. **c**, Heat map shows proteins significantly altered by both *APOE4* and *APOE2* in CU individuals. Cell color indicates the effect size of *APOE4* (column 1) and *APOE2* (column 2) extracted from linear models adjusted for age, sex, mean protein level, and cohorts. Asterisks (*+) denote FDR-corrected significance of the effect sizes. Only proteins with average absolute effect size > 0.2 are shown. "☆" denotes opposite effect directions between *APOE4* and *APOE2*. Red, blue, and purple labels indicate AD mediator proteins for *APOE4*, *APOE2*, or both, respectively. See Source Data Fig. 4 for detailed statistical summary.

distinct upstream mediator architectures and differences in regulatory strength across shared networks, giving rise to asymmetric proteomic patterns that may underlie their divergent AD risk.

Finally, 2,396 proteins were associated with AD but not with *APOE4* or *APOE2*, indicating *APOE*-independent processes. Examples include OMG (neuronal growth inhibitor), SMOC1 (extracellular matrix), GPD1 (glycerol metabolism), and POSTN (tissue remodeling). In addition, 12 proteins (for example, CLUAP1 and GAL3ST2) were downstream of AD in both *APOE4* and *APOE2* carriers, reflecting shared AD-driven alterations.

**Replication and the role of *APOE* in Aβ pathology in the BioFINDER-2 cohort**

Building on the proteomic associations identified in GNPC, we next validated these findings in BioFINDER-2, a cohort with

biomarker-confirmed AD cases (via CSF Aβ42/40 ratio), addressing the lack of Aβ measures in GNPC. The analysis included 846 CU individuals and 259 patients with AD with plasma SomaLogic 7K proteomics data (Supplementary Table 1).

Of 156 *APOE4*-associated proteins, 117 (33% of GNPC finding) were replicated; for 83 *APOE2*-associated proteins identified, 50 proteins (26% of GNPC finding) showed consistent associations. Notably, proteins with larger effect sizes in GNPC were more likely to replicate in BioFINDER-2 (Supplementary Fig. 4a,b). Ten *APOE4*-related proteins, including SPC25 and TBCA, were identified as upstream mediators (mediation proportion 5–83%; Fig. 5a). SPC25 was replicated with a full mediation effect.

To better distinguish *APOE*-driven proteomic changes from downstream Aβ-mediated effects, we expanded the analysis to include 316 individuals with mild cognitive impairment (MCI), broadening the age

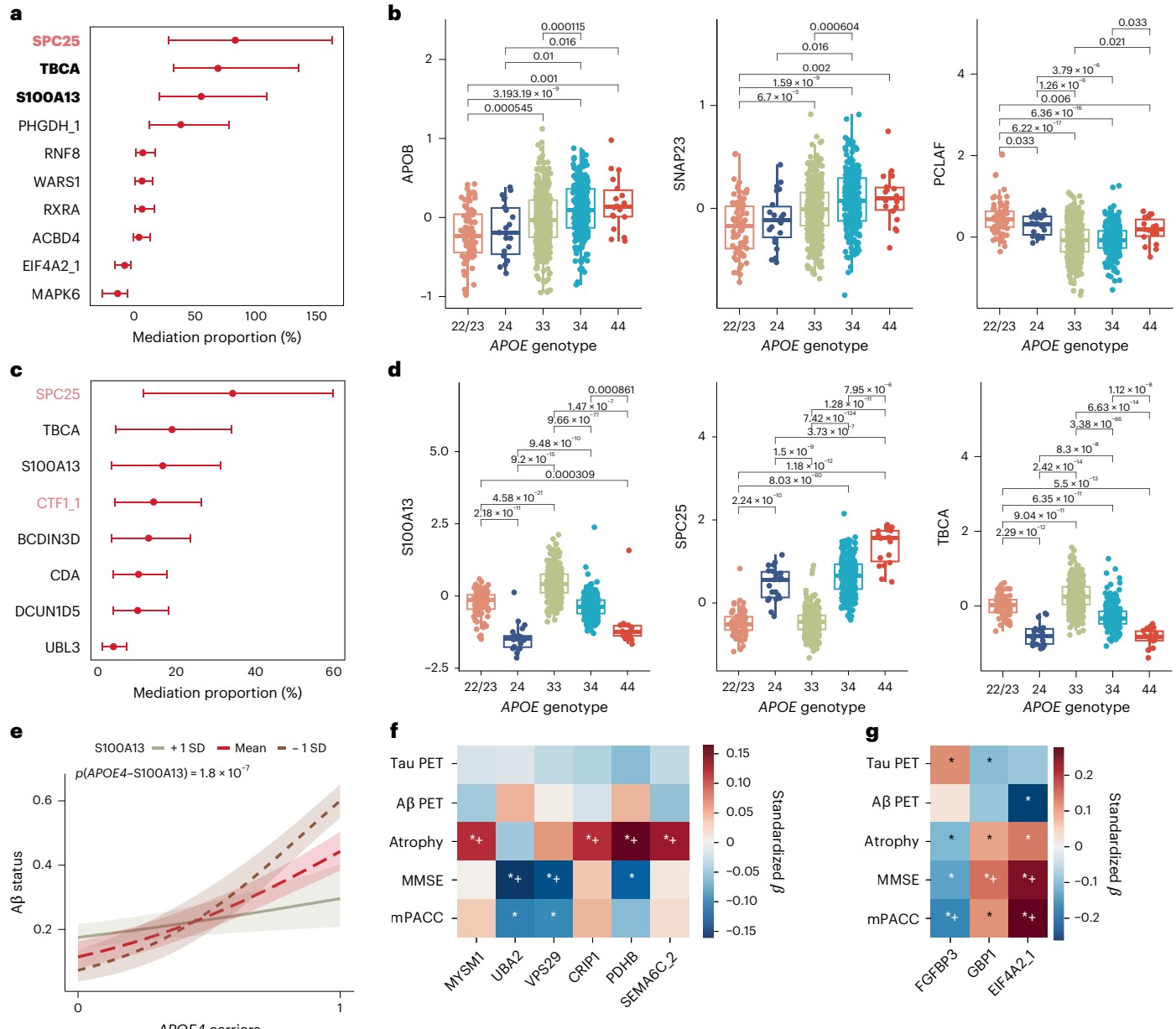

**Fig. 5 | Identification of mediators for Aβ status in the BioFINDER-2 plasma SomaLogic cohort. a,c**, Dot plots with error bars show mediation proportions and 95% confidence intervals for proteins classified as upstream mediators of the *APOE4* effect on AD diagnosis (**a**, N = 998 individuals) or Aβ status (**c**, N = 1,282 individuals). The x axis shows the mediation proportion (%) for each protein. Red dots indicate significant mediation proportions, and horizontal lines indicate the corresponding 95% confidence intervals. Bold labels denote full mediation, and red labels denote mediators replicated in the GNPC cohort. **b,d**, The box plots show residual protein levels (y axis, after adjusting for age, sex, and mean protein level) grouped by *APOE* genotypes for *APOE2*-related Aβ mediators (**b**) and *APOE4*-related Aβ mediators (**d**) in the Aβ⁻ group (N = 715 individuals). The center line of each box indicates the median (50th percentile); the lower and upper edges of the box represent the 25th and 75th percentiles, respectively. Whiskers extend to the most extreme values within 1.5 times the IQR; data points beyond this range are considered outliers and have been excluded from the plot display. The x axis represents *APOE* genotype; ε2/ε2 and ε2/ε3 carriers were merged into the "22/23" group due to a small sample size of ε2/ε2 carriers. Group differences were assessed using two-sided Welch's *t*-tests, and *P* values were adjusted for multiple comparisons using the Holm–Bonferroni

method. Pairwise comparisons were performed across all groups, but only significant adjusted *P* values are shown. **e**, The interaction plot shows the interaction between *APOE4* carrier status (0, noncarriers (ε3/ε3); 1, carriers) and S100A13 protein level in predicting Aβ status in CU and MCI groups (N = 1,038 individuals). Lines represent estimated Aβ status across *APOE4* groups at three levels of S100A13 levels: one standard deviation below the mean (−1 s.d., dashed brown), the mean (dashed red), and one standard deviation above the mean (+1 s.d., solid gray). **f**, Heat map of standardized regression coefficients (β) for associations between *APOE*-associated proteins (x axis) and AD-related features (y axis; tau PET in the temporal meta-ROI, Aβ PET, temporal lobe cortical thickness, MMSE, and mPACC) in Aβ⁻ individuals (N = 715). Standardized β and significance were estimated in linear models with adjustment for age, sex, and mean protein level. Only proteins significantly associated with at least one feature are shown. **g**, The same analysis in Aβ⁺ individuals (N = 706). In **f** and **g**, red indicates positive associations, and blue indicates negative associations, with darker shades indicating stronger associations. Asterisks: * indicates nominal significance, and *+ indicates FDR-corrected significance. See Source Data Fig. 5 for detailed statistical summary.

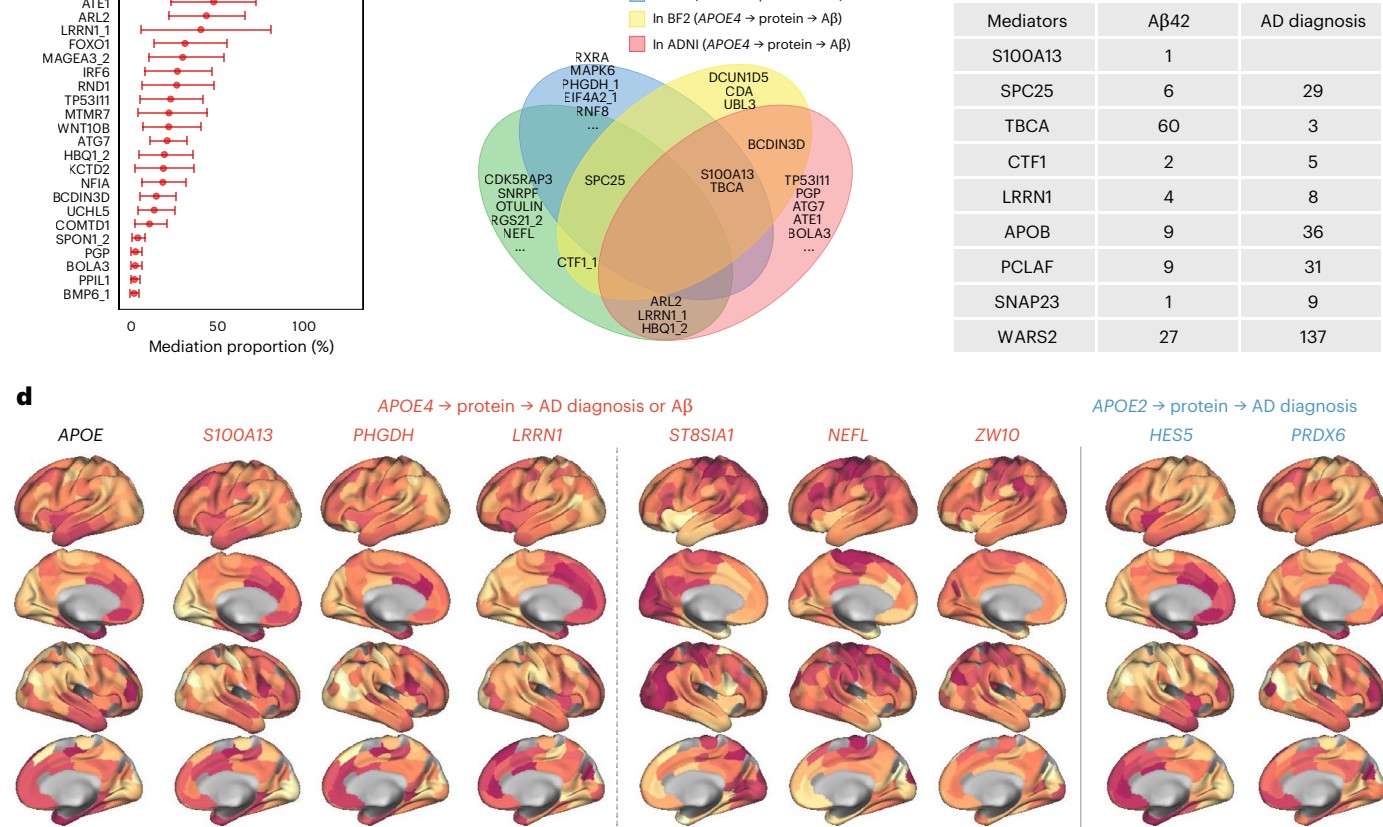

**Fig. 6 | Mediators supported by CSF proteomics, GWAS, and transcriptome.**
**a**, Dot plots with error bars show mediation proportions and 95% confidence intervals for proteins classified as upstream mediators of the *APOE4* effect on Aβ status in ADNI (*N* = 620 individuals). The *x* axis shows the mediation proportion (%) for each protein. Red dots indicate significant mediation proportions, and horizontal lines indicate the corresponding 95% confidence intervals. Bold label denotes full mediation. **b**, Venn plot shows *APOE4*-related mediators identified in 3 SomaLogic proteomics datasets. **c**, The table shows the number of SNPs in the coding gene of key mediators (identified as mediators in at least 2 datasets) that are associated with AD clinical diagnosis or CSF Aβ42 level in external GWAS

studies. Detailed statistical summary for each SNP is provided in the Source data.
**d**, The plot shows brain-wide gene expression patterns, with statistics reflecting spatial correlations between *APOE* expression and coding gene of *APOE4*- or *APOE2*-associated proteins in the human brain. Key genes with significant spatial co-expression with *APOE* are shown. Red indicates coding genes of *APOE4*-mediators; blue denotes coding genes of *APOE2*-mediators. *R* denotes the Pearson correlation coefficient between the regional gene expression profile of each indicated protein and the reference *APOE* expression map across the 200 Schaefer parcels. See Source Data Fig. 6 for detailed statistical summary.

range and disease spectrum. Based on CSF Aβ42/40 ratios, 715 individuals (50.3%) were Aβ⁻ and 706 (49.7%) were Aβ⁺ (Supplementary Table 1).

Of 73 *APOE2*-associated proteins that were identified, 43 (59%) were already dysregulated in Aβ⁻ individuals. Top predictors of Aβ positivity in CU/MCI individuals included APOB, SHH, and BIRC2, enriched in mitochondrial import and gene expression pathways, supporting early *APOE2*-related pathways (Supplementary Fig. 5a). Similarly, 65 of 167 *APOE4*-related proteins (39%) were already altered in the Aβ⁻ group. Key contributors to Aβ prediction in CU/MCI individuals, such as AKT2, HDAC8, APOB, CYP3A4, and SPC25, were enriched in chromatin regulation, drug metabolism, gene expression, and cell cycle pathways (Supplementary Fig. 5b). Furthermore, we found that those early associations were not amplified or diminished by age and Aβ status. Moreover, sex-stratified analyses in Aβ⁻ individuals showed highly concordant *APOE*-associated effect sizes between men and women for these early-altered proteins, and *APOE*–sex interaction analyses did not identify sex-dependent modification (Extended Data Fig. 2a,b).

Importantly, 18 proteins were associated with both alleles in Aβ⁻ individuals, including oppositely regulated VPS29, PHGDH, and FOXO1 and similarly regulated S100A13, TBCA, ARL2, and BCDIN3D, consistent

with GNPC (Extended Data Fig. 2c). These findings underscore the importance of these proteins in explaining divergent AD risks.

Aβ-anchored upstream–downstream mediation framework captured similar proteomics evolution: *APOE2*'s effects were largely independent of Aβ. APOB, SNAP23, WARS2, and PCLAF were replicated from the GNPC findings as upstream mediators, showing full mediation with an ε2-dominant effect in Aβ⁻ individuals (Fig. 5b). No protein was mediated by Aβ pathology.

By contrast, 8 *APOE4*-associated proteins (for example, SPC25, TBCA, S100A13, and BCDIN3D) partially mediated *APOE4*'s effect on Aβ positivity (mediation proportion 4–35% Fig. 5c), with an ε4-dominant and dosage-dependent effect in Aβ⁻ individuals (Fig. 5d). However, more (28) proteins were more strongly mediated by Aβ pathology, supporting a proteomic signature vulnerable to Aβ pathology. An interaction was also observed between *APOE4* and S100A13 levels, whereby lower S100A13 levels enhanced the association between *APOE4* and Aβ positivity in CU and MCI participants (Fig. 5e), highlighting its importance in explaining AD risk.

Sensitivity analyses showed broadly consistent *APOE*-associated protein changes and mediation patterns (Supplementary Fig. 6).

Four mediator proteins, CDA, DCUN1D5, S100A13, and TBCA, consistently showed stronger upstream than downstream mediation, whereas SPC25 was attenuated after adjustment for medication use, largely driven by cholinesterase inhibitor (ChEIs) use. When amyloid burden was modeled using continuous CSF Aβ42/40, 6 of 8 upstream mediators remained significant, whereas none remained significant with continuous Aβ measurement using positron emission tomography (PET). This occurred despite strong concordance between CSF Aβ42/40 ratio and Aβ PET at both categorical and continuous levels. Because the PET sample was enriched for CU and MCI individuals, its narrower Aβ PET range, particularly at early amyloid stages, likely limited sensitivity to detect upstream mediation effects when modeled continuously (Supplementary Notes). Several *APOE*-associated proteins were associated with later AD-related features, including cognition and cortical atrophy (Fig. 5f,g and Supplementary Notes), whereas mediator proteins showed no associations with later AD features, supporting their early role.

### Extending mediation analysis to CSF proteomics

To evaluate whether *APOE*-associated proteins identified in plasma were also detectable in CSF, we next analyzed SomaLogic 7K proteomics data from the Alzheimer's Disease Neuroimaging Initiative (ADNI) cohort (*N* = 666; Supplementary Table 1). We identified 93 *APOE2*- and 684 *APOE4*-associated proteins in this cohort. Although only a subset of plasma-identified proteins reached statistical significance in CSF (71 *APOE4*- and 19 *APOE2*-associated proteins), the directions of *APOE* effects were largely concordant between the CSF and plasma SomaLogic datasets (Supplementary Fig. 4).

Consistent with plasma signatures in terms of disease-stage distribution, most *APOE2*-related alterations in CSF (78%, 73/93) were already evident in Aβ⁻ individuals and were enriched for protein localization, autophagy, and cell–cell communication pathways (Supplementary Fig. 7a). About 34% (231/684) of *APOE4*-related alterations remained significant in Aβ⁻ individuals, with enrichment in extracellular matrix organization, autophagy, meiosis, and cell–cell communication pathways (Supplementary Fig. 7b).

Mediation analysis identified 24 *APOE4*-associated proteins showing stronger evidence at upstream mediation pathway (*APOE4* → protein → Aβ status) with mediation proportion up to 96% (Fig. 6a). More (213) proteins were downstream of Aβ as observed in other cohorts. Several mediators overlapped with plasma results (Fig. 6b): S100A13 and TBCA were consistent across BioFINDER-2 analyses, BCDIN3D was specific to Aβ outcomes. By contrast, SPC25 mediated *APOE4*–AD associations in plasma but not in CSF, indicating compartment-specific differences in mediation patterns. This divergence may partly reflect contextual modulation in plasma, as SPC25 no longer mediated *APOE4*–Aβ associations after adjustment for ChEIs use, while also being consistent with differences in tissue origin and regulatory context between plasma and CSF. No *APOE2* mediators were identified, likely reflecting the limited number of ε2 carriers (*N* = 46) in ADNI.

### Assessing *APOE* signature in other proteomics platforms

To assess the generalizability of *APOE*-associated proteomic signatures beyond the primary SomaLogic analyses, we examined independent CSF and plasma datasets generated using tandem mass tag-based mass spectrometry (TMT-MS) and OLINK (Explore 3072). In ADNI CSF TMT-MS, broad *APOE4*- and Aβ-related protein changes were observed, and a small number of upstream mediator proteins were identified (Extended Data Fig. 3). Direct overlap with the ADNI CSF SomaLogic results was limited because many candidate mediators were not measured or did not pass quality control in the TMT-MS dataset (Supplementary Fig. 8). Among overlapping proteins, most showed concordant effect directions, although some, including S100A13 and TBCA, showed discordant associations across platforms, consistent with weak cross-platform agreement in direct comparison of protein

measurements (Supplementary Fig. 4, Extended Data Fig. 4, and Supplementary Notes).

In BioFINDER-2 CSF OLINK, several SomaLogic candidate mediators were likewise unavailable, and after adjustment for Aβ, only a small number of *APOE4*-associated proteins remained, including SNAP25 and CKAP4, whereas LDLR was the only *APOE2*-associated CSF protein and did not mediate AD or Aβ outcomes (Extended Data Fig. 5). Several key *APOE4* mediator proteins identified in plasma SomaLogic analyses, including S100A13, TBCA, NEFL, ST8SIA1, and LRRN1, showed opposite effect directions in CSF OLINK, and direct comparisons indicated generally weak correlations for measurements of the same proteins across platforms, highlighting substantial proteomic heterogeneity across biofluids and platforms (Extended Data Fig. 6 and Supplementary Notes).

In UK BioBank (UKBB) plasma OLINK, some *APOE*-related signals, including PLA2G7 and LDLR, were directionally consistent with the main plasma findings. PLA2G7 showed allele dose-dependent effects in both ε4 and ε2 carriers, whereas LDLR was consistently elevated in ε2/ε2 carriers across age groups (Extended Data Fig. 7 and Supplementary Notes). Overall, these analyses indicate that although a limited number of *APOE*-associated proteins were detectable across additional proteomic platforms and biofluids, global concordance was modest and strongly influenced by differences in target coverage, assay characteristics, and biological compartment, supporting these datasets primarily as orthogonal context rather than direct one-to-one replication.

### Comparison of NEFL across assays and datasets

To further illustrate cross-biofluid and cross-platform differences at the level of individual *APOE*-related proteins, we examined NEFL across affinity-based proteomic platforms, MS-based proteomics, and high-sensitivity immunoassays (Extended Data Fig. 8). In CSF, NEFL showed positive associations with AD diagnosis or Aβ status across three proteomic platforms and NTK (NeuroToolKit), largely independent of *APOE* genotype, consistent with NEFL tracking amyloid-related neurodegenerative processes. In plasma, by contrast, SomaLogic-measured NEFL showed an *APOE4* gene dose-dependent reduction, with directionally similar decreases also observed by Simoa, although NEFL in that assay remained primarily associated with Aβ status. Although CSF NEFL measured by NTK was strongly correlated with plasma NEFL measured by Simoa, it did not show directionally consistent or *APOE4* gene dose-dependent associations, consistent with *APOE4*-related effects being more readily masked by amyloid-associated variation in CSF than in plasma. Negative associations between CSF NEFL and *APOE* genotypes were also observed in SomaLogic, indicating that *APOE*-related effects remain detectable alongside pronounced amyloid-associated signals (see Supplementary Notes for details). Together, these findings suggest that certain *APOE*-driven reductions in NEFL signals captured by aptamer-based detection might also remain detectable in CSF and are not fully masked by dominant Aβ-associated effects.

### Genetic and transcriptomic support for *APOE*-associated mediators

Having observed limited reproducibility of *APOE*–protein associations across platforms, we next asked whether the role of *APOE*-associated proteins is nevertheless supported by other orthogonal evidence. Published GWAS datasets (AD, GCST002245 study[18]; CSF Aβ42, GCST90129599 study[19]) further supported *APOE*-associated mediator proteins at the genetic level. Among the 8 key *APOE4* mediators identified in at least two datasets, 5 had coding variants associated with AD or CSF Aβ42 (beta effect estimate (*β*) = −0.21–0.25, *P* = 0.003–0.049), while all 4 key *APOE2* mediators identified in multiple datasets were likewise supported by single-nucleotide polymorphism (SNP) associations (*β* = −0.31–0.24, *P* = 3.308 × 10⁻⁵ to 0.049). These results further substantiate their relevance to AD biology.

To further assess brain relevance, we examined spatial co-expression with *APOE* in Allen Brain Atlas transcriptomic data. Although several key mediators, including SPC25, APOB, and PCLAF, were unavailable, 123 of 278 *APOE4*-associated proteins and 58 of 152 *APOE2*-associated proteins with transcriptomic data showed spatial correlation with *APOE* expression in the human brain. These included positive correlations for S100A13, PHGDH, LRRN1, HES5, and PRDX6, and negative correlations for ST8SIA1, NEFL, and ZW10 (Fig. 6d). No spatial correlation was observed between Aβ PET uptake and expression of *APOE* or *APOE*-associated genes.

In the ROSMAP bulk brain RNA-sequencing data, *APOE* genotype was associated with the expression of only a limited subset of mediator genes, whereas AD status represented the dominant source of transcriptional variation in postmortem brain tissue (Supplementary Tables 5 and 6 and Supplementary Notes).

### Exploration of longitudinal *APOE*–protein associations in CSF

To further explore the longitudinal stability of *APOE*–protein associations beyond cross-sectional age stratification, we leveraged the Parkinson's Progression Markers Initiative (PPMI) CSF OLINK cohort ($N = 253$; Supplementary Table 1). Among 53 *APOE4*- and 13 *APOE2*-associated proteins identified in GNPC, only PRSS8 and NEFL showed nominal *APOE*–time interactions; among 26 testable proteins from BioFINDER-2 CSF OLINK, only SIGLEC1 showed such an effect (Supplementary Table 7). These results suggest that most *APOE*–protein associations are stable over time, highlighting its persistent effect.

## Discussion

Through large-scale, cross-platform proteomic profiling of plasma and CSF across multiple cohorts, we systematically mapped *APOE4*- and *APOE2*-associated proteomic signatures to better characterize their relationship with AD susceptibility. One key contribution of our study was the identification of several allele-specific proteins as upstream mediators of the *APOE*–AD association, implicating distinct biological processes such as cell-cycle regulation, cytoskeletal integrity, and mitochondrial function that may underlie the divergent effects of *APOE2* and *APOE4* on AD risk. Our multicohort design further revealed that support for individual *APOE*-associated proteins varied across platforms, tissues, and datasets; however, several key plasma proteins were supported by one or more complementary lines of evidence, including CSF SomaLogic datasets, allele-dominant effects, AD GWAS evidence, and transcriptomic co-expression with *APOE* in the human brain, and remained robust after adjusting for ancestry, vascular pathology, and medication use.

Key proteins mediating the effects of *APOE2* on AD include SNAP23 (endocytic trafficking), APOB (lipid transport and metabolism), WARS2 (mitochondrial translation), and PCLAF (DNA repair), suggesting protection via homeostasis, mitochondrial function, and genomic stability[20–23]. Mediator proteins showed only limited associations with later AD phenotypes, consistent with their upstream positioning and suggesting a role as early molecular modifiers rather than direct determinants of progression.

By contrast, *APOE4* mediators included other proteins, for example, SPC25 (kinetochore assembly, linked to microglial proliferation[24] and elevated in MCI[25]), S100A13 (moderate Aβ positivity in *APOE4* carriers and related to cellular senescence), and ZW10 (spindle checkpoint). Some of these were enriched in OPCs, and pointed to cell-cycle dysregulation as a contributor to AD risk[26–29], especially in glial cells. TBCA and ARL2 further implicate cytoskeletal disruption in *APOE4*-related neurodegeneration[30,31].

SPC25, previously highlighted in SomaLogic-based studies[14,16,32,33], was identified here as an upstream mediator of the *APOE4*–AD association. Its robust association with *APOE4*, including in Aβ⁻ and ChEIs-untreated individuals, supports a treatment-independent genotype effect. By contrast, its association with Aβ lost significance after

adjusting for ChEIs use, suggesting that the observed mediation may partly reflect treatment effects rather than underlying Aβ pathology. SPC25 was not associated with *APOE2*, reinforcing its *APOE4* specificity.

The impact of ChEIs on SPC25 appeared specific to *APOE4* carriers, consistent with previous reports of *APOE* genotype-dependent responses to ChEIs therapy[34–36]. Given that SPC25 is not known to represent a protective or compensatory response in the context of neurodegeneration, this finding does not necessarily indicate therapeutic benefit. Instead, it raises the possibility that *APOE4*-specific molecular responses to ChEIs treatment, reported previously as differential treatment responsiveness in biomarker-unanchored trials, may reflect genotype-driven effects that are not directly linked to AD disease biology. This observation highlights the importance of distinguishing genetic-driven molecular effects from biological changes secondary to disease pathology. Furthermore, as ChEIs commonly overlap with Aβ positivity and clinical AD, treatment adjustment is crucial to disentangle therapeutic from pathophysiological effects.

Beyond individual mediators, *APOE2* and *APOE4* showed distinct large-scale trajectories, consistent with their opposing effects on AD risk. Most *APOE2*-associated proteins functioned as upstream mediators, promoting homeostasis, stress suppression, and immune regulation, with only a small subset remodeled at disease onset. Several *APOE2*-specific proteins not directly linked to AD also reflected tissue maintenance and systemic resilience. Together, these patterns support the notion that *APOE2* protects against AD by sustaining stable, resilience-promoting programs that resist pathological remodeling. By contrast, *APOE4* carriers show widespread proteomic alterations that, although detectable early and independent of age, are highly susceptible to pathology-driven remodeling. While a few (for example, S100A13 and TBCA) acted upstream, the majority emerged as downstream responders, particularly within glial and vascular compartments, implicating enhanced exocytosis and vascular activity but impaired immune trafficking, consistent with AD signature[37–39]. Additional *APOE4*-specific changes, enriched in macrophages, indicate immune and structural activation with loss of homeostasis, amplifying systemic vulnerability. Many *APOE4*-related changes did not simply mirror disease effects but may instead reflect early compensatory or allele-specific programs that are later reshaped by downstream cascades, highlighting a transition from genotype-predominant to pathology-predominant regulation.

The divergent AD risks conferred by *APOE2* and *APOE4* are unlikely to reflect a simple inversion of the same proteomic network. Instead, multiple layers of asymmetry emerge. First, each allele operates through distinct sets of mediator proteins. Second, we identified a pool of early co-dysregulated proteins by *APOE4* and *APOE2* in CU individuals, enriched in autoubiquitination, RNA modification, and autophagy pathways, with signals in basal keratinocytes and club cells—peripheral compartments implicated in amyloid-beta precursor protein expression and systemic Aβ clearance[40–42]. Within this pool, a few oppositely regulated proteins (for example, VPS29, PHGDH, and FOXO1) may serve as "switch nodes" driving divergent trajectories, whereas a broader set (for example, S100A13, BCDIN3D, TBCA, and ARL2) showed ε4-dominant regulation, reflecting amplified vulnerability and possibly explaining reports linking *APOE2* to neurodegeneration under specific conditions[43]. These signatures were already present in Aβ⁻ individuals, indicating early *APOE*-driven changes rather than downstream amyloid effects. Collectively, these early and upstream asymmetries in molecular architecture may underlie the later susceptibility of proteomic programs to AD-driven reprogramming and pathological cascades.

Beyond allele-related features, we also identified downstream AD markers independent of *APOE*. These AD-specific signatures included OMG, a neuronal growth inhibitor linked to axonal regeneration deficits and neurodegeneration[44,45]; SMOC1, an extracellular matrix protein associated with glial activation, tissue remodeling, and AD pathology[46–48]; GPD1, related to glycerol metabolism and

energy imbalance; and POSTN (periostin), involved in tissue remodeling, neuroinflammation, and cognitive decline[49]. We further identified *APOE*-related proteins consistently reshaped by downstream pathology in both *APOE4* and *APOE2* carriers, such as CLUAP1 (ciliary function, linked to CSF α-synuclein[50]) and GAL3ST2 (sphingolipid metabolism, associated with AD risk[51]). Together, these proteins highlight AD-predominant mechanisms independent of *APOE* genotypes and may serve as biomarkers of disease progression.

A key feature of our findings is the substantial heterogeneity of *APOE*-associated proteomic signatures across measurement platforms and biological matrices. While stable and directionally consistent *APOE* effects were reproducibly observed within the same technology, most notably in plasma and CSF samples measured using the SomaLogic platform, consistent with the lower assay variance reported for this technology[52–55], concordance across different platforms was generally limited. This heterogeneity reflects fundamental differences in assay coverage, the molecular features captured, and quantification principles across platforms. In particular, SomaLogic provides broader target coverage, whereas OLINK panels and TMT-MS interrogate a more restricted set of proteins, resulting in a markedly smaller overlap of directly comparable targets. Consequently, many mediator proteins prioritized in SomaLogic analyses were not detectable or not reliably quantified in orthogonal datasets, including both OLINK and TMT-MS, largely due to differences in target selection, dynamic range, and detection rates, thereby limiting direct cross-platform comparability.

Even among overlapping proteins (for example, S100A13 and TBCA), cross-platform concordance was modest when directly comparing measurements across technologies (for example, OLINK versus SomaLogic and TMT-MS versus SomaLogic), consistent with previous reports highlighting the inherent challenges of replicating large-scale proteomic associations across platforms with distinct measurement principles[54,56]. Heterogeneity was also evident within SomaLogic itself, where proteins quantified by multiple aptamers showed only modest correlations (for example, ST8SIA1 and LRRN1), highlighting the importance of target recognition and assay-specific binding properties in shaping observed proteomic signals. Notably, for certain proteins such as TBCA and NEFL, statistically significant associations with *APOE4* were detected across platforms but with opposite effect directions, indicating that these assays may be sensitive to different molecular features or biological contexts rather than capturing directly comparable signals.

Within the broader context of platform heterogeneity, NEFL provides a particularly informative example of how disease context and genotype can jointly shape proteomic signals across biofluids and measurement technologies, without invoking technical artifacts. In our analyses, we observed *APOE*-driven reductions in NEFL levels, even among Aβ⁺ individuals and patients with AD, most prominently in plasma SomaLogic measurements. Previous studies have shown that plasma NEFL measured by SomaLogic is elevated across multiple non-AD neurodegenerative diseases, including Parkinson's disease, frontotemporal dementia, and amyotrophic lateral sclerosis, supporting its validity as a general marker of neuronal injury[14]. The opposing direction observed in AD therefore does not indicate reduced assay sensitivity. Rather, it suggests that in the AD context, where *APOE4* carriers are highly enriched, genotype-driven reduction of NEFL may dominate over, or suppress, the more generic injury-related increases commonly observed across other neurodegenerative conditions. This interpretation is further supported by biofluid-specific patterns. In CSF, NEFL associations were largely dominated by amyloid-related pathology across platforms, whereas *APOE*-related effects were more readily detectable in plasma. *APOE*-associated reductions in NEFL remained detectable in SomaLogic-based CSF measurements, indicating that aptamer-based detection can retain sensitivity to genotype-associated NEFL signals even within a biological matrix strongly influenced by Aβ pathology.

Together, these findings suggest that the observed heterogeneity in *APOE*-associated proteomic effects is primarily shaped by differences in assay coverage, detection sensitivity, and the platform-dependent capture of distinct molecular features. Accordingly, proteomic associations, particularly those that are platform specific, should be interpreted with caution in a context-aware manner that accounts for assay design, biological matrix, and disease anchoring. Future peptide- or isoform-resolved proteomic analyses may therefore be required to further clarify how *APOE* genotype and amyloid pathology differentially shape those proteins across biofluids and disease stages.

Beyond platform heterogeneity, an additional interpretative challenge concerns the tissue origin of *APOE*-associated proteomic changes. Although several *APOE*-associated signals, particularly those related to *APOE4*, enriched for CNS-relevant pathways, colocalized with *APOE* in the brain, and supported by AD GWAS evidence, most findings were derived from plasma proteomics, and their tissue origin therefore remains uncertain. Many proteins are likely of peripheral origin, yet peripheral pathways such as inflammation, lipid metabolism, or Aβ clearance may themselves contribute to AD[41,57–61]. Even brain proteomic datasets may face similar challenges, as some protein changes could reflect secondary effects of Aβ pathology rather than early genotype-driven mechanisms. For example, certain *APOE4*-associated proteins observed in CU or AD individuals may partly reflect the greater Aβ burden typically seen in *APOE4* carriers, making it difficult to disentangle whether levels are primarily driven by genotype, downstream amyloid processes, or their interplay. This nuance has not always been systematically considered, but it is critical for accurate interpretation of *APOE*-related proteomic changes in AD.

Consistent with this complexity, exploratory bulk transcriptomic analyses in postmortem brain tissue revealed only limited *APOE* genotype-associated differences among mediator genes, whereas AD diagnosis was associated with more widespread transcriptional changes. This dissociation likely reflects multiple, nonmutually exclusive factors. First, the proteomic signatures identified here are derived from biofluids, which integrate signals across tissues, cell types, and regulatory processes, and may therefore capture systemic or posttranscriptional effects of *APOE* that are not directly mirrored at the level of bulk brain RNA expression. Second, protein abundance is shaped by additional layers of regulation beyond transcription, including translation efficiency, secretion, degradation, and compartment-specific dynamics[62–64], such that *APOE*-related effects on protein levels may occur independently of detectable changes in steady-state mRNA levels. Together, these considerations suggest that proteomic and transcriptomic measurements capture complementary, rather than interchangeable, layers of *APOE*-related biology across tissues and disease stages. Future studies integrating brain multi-omics with AD biomarkers across disease stages and functional validation may further distinguish central from peripheral origins and establish causal roles in AD risk.

Despite the strengths of our multicohort design, several limitations warrant mention. First, cohort heterogeneity complicates interpretation: in the GNPC cohort, the absence of AD biomarkers limited our ability to confirm clinical diagnoses or exclude potential misdiagnosed non-AD cases (a limitation we addressed by validating all GNPC findings in BioFINDER-2, which includes biomarker-confirmed AD diagnoses), UKBB includes mostly younger CU without AD biomarkers data, while PPMI is PD focused. Our multicohort design increases complementarity but limits strict comparability and therefore requires context-specific interpretation. Second, mediation analyses were cross-sectional and should be viewed as statistical rather than causal; longitudinal results from PPMI should be considered exploratory given modest sample size, follow-up duration, and potential pathology differences, and longitudinal analysis in AD-specific cohorts or experimental validation will be required. Third, AD diagnosis and amyloid status were primarily modeled as binary variables, which may not fully capture the

continuous nature of disease progression and amyloid accumulation. Although sensitivity analyses using the continuous CSF Aβ42/40 ratio largely supported the robustness of our findings, analyses based on continuous Aβ PET measures were limited by a restricted dynamic range in predominantly preclinical samples. Fourth, although sensitivity analyses were performed in BioFINDER-2 (for example, adjusting for ancestry, vascular pathology, and medication use), most cohorts included in this study were predominantly of European ancestry, and harmonized race and ethnicity information were not uniformly available across datasets. As a result, the generalizability of our findings to racially and ethnically diverse populations may be limited. Moreover, residual confounding from other factors (for example, comorbidities and lifestyle) cannot be fully excluded. Fifth, while deep-learning models such as BINN provided a systems-level perspective complementary to Gene Ontology (GO) enrichment and mediation analyses, their generalizability across diverse cohorts is uncertain. In addition, GWAS-based supportive analyses relied on publicly available summary statistics, precluding formal testing of SNP–*APOE* genotype interactions. Finally, our stratification of *APOE*-associated proteins into functional categories is based on statistical criteria and should be interpreted as a heuristic framework, as biological roles may overlap across categories. Collectively, these limitations emphasize the need for validation across assays, populations, and AD-specific longitudinal cohorts and warrant cautious interpretation.

In summary, our study provides a systematic proteomic map revealing the differential AD risk among *APOE* isoforms. *APOE2* promotes cellular resilience potentially through stable constitutive programs, particularly by maintaining homeostasis and suppressing stress and immune responses, whereas *APOE4* may establish vulnerable states prone to pathological remodeling, notably by enhancing proliferation with reduced regulatory control. These asymmetric proteomic changes reflect distinct molecular strategies rather than mirror processes. Despite limitations related to cohort heterogeneity and platform variability, our findings highlight pathways involving cell-cycle regulation, gene expression, cytoskeletal integrity, and mitochondrial health as key mediators of *APOE*-associated AD risk and point to upstream and allele-specific proteins already altered in Aβ⁻ individuals as promising biomarkers and targets for early detection and preventive interventions.

## Methods

### GNPC SomaLogic 7K participants
The GNPC is a multicenter, international proteomics initiative. The cohort composition, sample processing, and proteomics pipeline have been described in detail in a previous study[14]. The present study represents a focused deep dive into the GNPC dataset to investigate the role of *APOE4* and *APOE2* in AD, using the same inclusion criteria and preprocessing steps as in Imam et al.[14]. To ensure independence of discovery and validation cohorts, BioFINDER-2 participants were additionally excluded from the GNPC dataset in the present study.

In total, 3,289 participants were included (aged 27–90 years; median age 76 years; 41% men), comprising 2,069 CU individuals and 1,220 patients with AD dementia. Plasma proteomics were profiled using the SomaScan 7K platform (SomaLogic), which quantifies proteins with DNA-based SOMAmer reagents[65]. Proteomics data underwent the standard SomaScan pipeline, including quality control and adaptive normalization by maximum likelihood. APOE-targeting aptamers were excluded, resulting in 7,285 aptamers corresponding to 6,358 unique proteins (Supplementary Fig. 3a). Unless specified otherwise, results were reported at each aptamer level.

For primary analyses, ε2/ε4 carriers were excluded to avoid confounding[14]. *APOE4* analyses compared ε3/ε4 and ε4ε4 carriers to ε3/ε3 (binary variable), and *APOE2* analyses compared ε2/ε2 and ε2/ε3 carriers to ε3/ε3. ε2/ε4 carriers were only included in descriptive comparisons (conducted for GNPC, BioFINDER-2, and UKBB due to

sufficient samples) of all six genotype groups (ε2/ε2, ε2/ε3, ε2/ε4, ε3/ε3, ε3/ε4, ε4/ε4).

### BioFINDER-2 cohort
The Swedish BioFINDER-2 study (NCT03174938; https://biofinder.se) contains both SomaLogic 7K assay (plasma) and OLINK (CSF) proteomics with detailed AD phenotyping data, for example, Aβ data, including CSF Aβ42/Aβ40 ratio and Aβ PET.

We used BioFINDER-2 plasma SomaLogic data to validate the plasma proteomics of *APOE* and explore the role of these proteins in *APOE*–Aβ associations. Focused on Aβ-related pathological changes across AD progression, we included 846 CU, 316 MCI, and 259 AD individuals with genotyping and SomaLogic proteomics measurements in plasma, resulting in 1,421 individuals (aged 20–93 years, median age 72 years; 47% men; Supplementary Table 1; with 7,285 aptamers and 6,358 unique proteins; Supplementary Fig. 3a) in total. Other dementia types that might introduce confounding neurodegenerative pathways were excluded. The details of these subcohorts have been described previously, including the diagnostic criteria[66] and cognitive staging[67].

To explore the heterogeneity and consistency of *APOE* proteomics across platforms and tissues, we also used CSF proteomics data from OLINK Explore 3072 proximity extension assay. A total of 1,475 individuals (aged 20–93 years, median 72 years; 46% men; Supplementary Table 1) with genotyping and OLINK proteomics were included. Like the plasma SomaLogic dataset, other dementia types were excluded. Protein quality control was conducted as described[46]. Specifically, the normalized protein expression (NPX) value of each protein was compared to its limit of detection (LOD), and only proteins with levels above the LOD in at least 70% of participants were retained. APOE proteins were also excluded. Resulting in 1,391 measurements (1,382 unique proteins; Supplementary Fig. 3a) in total.

Aβ status (positive or negative) was defined using CSF Aβ42/Aβ40 ratio with assay-specific thresholds. For the majority of participants (85% in CSF OLINK dataset, 90% in plasma SomaLogic dataset), CSF Aβ42 and Aβ40 were measured using the Roche Diagnostics Elecsys CSF electrochemiluminescence immunoassay and Roche NTK, respectively, as described before, with a previously established cutoff of 0.080[68,69]. When Roche measurements were unavailable, Aβ status was defined using Lumipulse G assays (cutoff 0.072)[70] or Meso Scale Discovery assays (cutoff <0.077 determined using mixture modeling).

To assess NEFL heterogeneity across platforms, plasma NEFL measured using the Quanterix Simoa assay, CSF NEFL level measured using Roche Elecsys NTK assay were used and were compared to plasma SomaLogic and CSF OLINK measurements.

Ethically, written informed consent was obtained from all participants before they entered the study. Ethical approval was obtained from the Regional Ethical Committee in Lund, Sweden.

### ADNI
The ADNI is a multisite, longitudinal study designed to track progression from normal cognition to MCI and AD using imaging, fluid biomarkers and clinical assessments. Further details about the study are available at www.adni-info.org. For CSF SomaLogic proteomics data, a total of 666 CU, MCI, and individuals with AD dementia (age range 54–91 years, median 74 years; 57% men; Supplementary Table 1) with available *APOE* genotyping and CSF proteomic profiles measured using the SomaScan 7K platform (v4.1) were included. Aβ status was defined using CSF Aβ42 concentrations measured with the Roche Elecsys assay, with Aβ positivity defined by the established cutoff of 880 pg ml⁻¹, which has shown high concordance with amyloid PET classification[71]. The APOE protein itself was excluded from downstream analyses, resulting in a final dataset comprising 7,001 quantified aptamers.

For CSF TMT-based mass spectrometry proteomics data (including 2,024 measurements, 2,015 unique proteins; Supplementary Fig. 3a; see Supplementary Methods for data preprocessing details), 536

participants (253 Aβ⁻ and 283 Aβ⁺; age range 55–91 years, median 74 years; 56% men; Supplementary Table 1) were included.

## UKBB

UKBB is a large-scale, multicenter prospective cohort study that included approximately 500,000 participants aged 40 to 69 years, who were enrolled between 2006 and 2010. For more information on study methods and data collection, see the UKBB online protocol (www.ukbiobank.ac.uk). A total of 4,813 participants from this cohort were included (age range 40–70 years, median 54 years; 54% men; Supplementary Table 1), all of whom had *APOE* genotype data and protein levels in plasma measured using OLINK Explore 3072. The same protein retention criteria as for BioFINDER-2 OLINK dataset were applied, and 1,319 proteins were ultimately included.

## PPMI

To assess longitudinal *APOE*-associated proteomic changes independent of Aβ, we analyzed CSF OLINK data from the PPMI Project 9000 substudy. A full study protocol is available at https://ida.loni.usc.edu/login.jsp. A total of 253 participants (age range 31–85 years, median 65 years; 66% men) with available CSF proteomics and Aβ data were included, comprising 72 patients with Parkinson's disease, 56 prodromal individuals, and 125 healthy controls. The same protein retention criteria as for BioFINDER-2 OLINK dataset were applied, resulting in 826 proteins. Participants provided written informed consent, and institutional review board approval was obtained at each site.

Longitudinal trajectories of *APOE*-associated proteins were evaluated using linear mixed-effects models adjusted for age, sex, mean overall protein level, diagnostic group, and Aβ42–time. Separate models were fitted for *APOE4* and *APOE2* analyses, excluding ε2/ε4 carriers.

## Proteomic preprocessing

Proteomic preprocessing was performed independently for SomaLogic and OLINK datasets under a harmonized framework. After applying cohort-specific inclusion criteria, proteins or samples with >15% missing values were excluded. ADNI SomaLogic data were analyzed using released datasets that had already undergone quality control[72]. In OLINK datasets, proteins with >30% of values below the LOD[46] were removed. APOE proteins were excluded from all datasets. When proteins were measured by multiple SomaScan aptamers or OLINK panels, results were reported at the aptamer or panel level, with corresponding IDs provided in the Source data files.

SomaLogic adaptive normalization by maximum likelihood values were log₂ transformed for downstream analyses, whereas OLINK NPX values were already log₂ transformed and normalized by the manufacturer. To account for global shifts in protein abundance, mean protein abundance was included as a covariate in all linear and mediation models[73]. For OLINK datasets, this mean was calculated using proteins above the LOD in >90% of participants; otherwise it was calculated across all included proteins. Before model fitting, only individuals with complete data for the protein of interest, outcome, and covariates were included. Protein-level deviates >5 s.d. from the mean were excluded.

## Bulk RNA sequencing data

Bulk RNA sequencing data from the dorsolateral prefrontal cortex in the ROSMAP study were analyzed. Gene-level expression values were provided in fragments per kilobase of transcript per million mapped reads (FPKM) format. Genes with expression >0.1 FPKM in at least 85% of samples were retained, converted to transcripts per million (TPM), log₂ transformed after adding a pseudocount of 1, and z-score standardized. Analyses were restricted to individuals with available *APOE* genotype, age at death, postmortem interval, and cognitive diagnosis at death, including 198 CU and 245 AD participants (*N* = 443; 35% men; Supplementary Table 1). Transcriptomic analyses focused on genes encoding mediator proteins identified in SomaLogic analyses.

Of 91 mediator proteins, 76 had corresponding gene expression data. Associations of *APOE* genotype and AD diagnosis with gene expression were tested using linear models adjusted for age at death, postmortem interval, and sex.

## Study design, sample size, randomization, and blinding

This study used data from observational human cohorts. Sample sizes were determined by data availability in each cohort; no statistical methods were used to predetermine them. Participants were not randomized but were classified according to *APOE* genotype, clinical diagnosis, and biomarker status. Proteomic measurements were performed blinded to any demographics or clinical characteristics. Predefined exclusion criteria were applied at the participant and protein levels, as described above. Data distributions were assumed to be approximately normal after log₂ transformation and normalization, but normality was not formally tested for each protein. Welch's *t*-test was used for pairwise comparisons of residual protein levels.

## Main statistical models

Proteins associated with *APOE* allele carrier status (binary, *APOE4* carriers (ε3/ε4 and ε4/ε4) versus ε3/ε3 carriers or *APOE2* carriers (ε2/ε2 and ε2/ε3) versus ε3/ε3 carriers) were analyzed using linear model type 1. Proteins associated with AD diagnosis (binary, CU versus AD dementia) or with Aβ status (binary, Aβ⁻ versus Aβ⁺) were analyzed using linear model type 2. A third model incorporated both *APOE* and AD diagnosis (or Aβ status) as independent variables to account for the confounding effect of the *APOE*–AD or *APOE*–Aβ association:

Model type 1: Protein = $\beta1 + \beta2\,APOE$ + covariates + $\epsilon$
Model type 2: Protein = $\beta3 + \beta4\,AD\,diagnosis\,(or\,A\beta)$ + covariates + $\epsilon$
Model type 3: Protein = $\beta5 + \beta6\,APOE + \beta7\,AD\,diagnosis\,(or\,A\beta)$ + covariates + $\epsilon$

Covariates included age, sex, and mean overall protein level in all cohorts, with study cohort additionally included as a covariate for the GNPC dataset. *APOE4* and *APOE2* analysis were conducted separately. In *APOE4* analysis, *APOE* indicates binary variables of *APOE4* carriers (ε3/ε4 and ε4/ε4) versus ε3/ε3 carriers. In *APOE2* analysis, *APOE* indicates binary variables of *APOE2* carriers (ε2/ε2 and ε2/ε3) versus ε3/ε3 carriers.

To investigate early *APOE* effects, Model type 1 was applied within the CU and Aβ⁻ subgroups. Age-related variation was assessed by identifying *APOE*-associated proteins in younger and older CU subgroups stratified by median age and by adding an *APOE*–age interaction term to Model type 1 within the CU and Aβ⁻ groups.

To investigate *APOE*–protein associations change across different diagnostic or Aβ status groups, we also conducted separate analyses using Model type 1 in the AD dementia (or Aβ⁺) groups. Similarly, an *APOE*–AD dementia diagnosis (or *APOE*–Aβ status) interaction term in Model type 3 for the whole cohort was used to statistically assess whether the effects of *APOE* on proteins differ significantly between CU and AD individuals or between Aβ⁻ and Aβ⁺ individuals.

To evaluate whether protein levels in *APOE4* carriers would alter their Aβ positivity rates in CU and MCI, in BioFINDER-2 plasma SomaLogic cohort, we fit interaction models of the following form: Aβ status = $\beta1 + \beta2\,(APOE4 \times protein) + \beta3\,APOE4 + \beta4\,protein$ + covariates + $\epsilon$. Covariates included age, sex, and mean overall protein level. For proteins showing significant interaction with *APOE4*, we performed simple slopes analysis and calculated Johnson–Neyman intervals using the interactions R package (v1.2.0). In the simple slopes plot, the effect of *APOE4* on Aβ positivity was estimated at three protein expression levels: the mean and ±1 standard deviation, with the slope representing the strength and direction of the association at each level.

All regression analyses were performed in R v4.4.2; statistical tests were two-sided. *P* values were adjusted using the Benjamini–Hochberg

(false discovery rate (FDR)) correction method, and significance was reported using FDR-corrected *P* values. Graphs of individual protein levels grouped by *APOE* genotypes are the residuals after eliminating the effect of covariates on proteins using a regression model; Welch's *t*-test was used to compare residual protein levels between groups. Two-sided *P* values are calculated, and *P* values were adjusted for multiple comparisons using the Holm–Bonferroni method. Data visualizations were generated using the R package ggplot2 v4.0.2 or the seaborn Python package v0.13.2.

### Mediation analysis

To further explore the role of *APOE*-related proteins in AD clinical diagnosis or Aβ pathology and to identify *APOE*-related proteins affected by AD pathology or Aβ pathology, we tested them in different statistical mediation models. The mediation models were used to systematically classify *APOE*-related proteins into two protein groups: proteins showing stronger evidence in upstream mediation pathways (*APOE* → protein → AD/Aβ) were defined as mediator proteins. Proteins showing stronger evidence in downstream mediation pathways (*APOE* → AD/Aβ → protein) were affected by *APOE* through an AD-mediated pathway and defined as AD-mediated proteins. Following the Baron and Kenny's Causal-Steps Test requirement, proteins significantly associated with *APOE* after FDR correction in Model type 1 were further tested for this upstream–downstream mediation framework.

Specifically, *APOE* is positioned as a temporal starting point, as the genotype is fixed at birth. In the upstream mediation pathway, *APOE* was the exposure, AD diagnosis or Aβ was the outcome, and *APOE*-associated proteins were potential mediators. In the downstream mediation pathway, *APOE* was the exposure, *APOE*-associated proteins were outcomes, and AD diagnosis or Aβ was a potential mediator.

Mediation analysis was performed using the "mediation" (v4.5.0) R package where mediation proportions were estimated using causal mediation analysis with 95% confidence intervals based on 1,000 bootstrapping replicates. Estimates such as direct effect, indirect effect, mediation proportion, and *P* values for each estimate were extracted directly from model results. *P* values for these metrics were FDR corrected.

### Subdivision of *APOE*-associated proteins

To functionally stratify *APOE*-associated proteins (FDR < 0.05 in Model type 1), we categorized them into four groups based on their associations with AD diagnosis and mediation relationships:

1. Mediator proteins: Proteins showing significant indirect effects (FDR < 0.05) in the upstream mediation pathway (*APOE* → protein → AD diagnosis) but not in the reverse (*APOE* → AD diagnosis → protein), or for which the absolute mediation proportion was stronger in the upstream than in the downstream direction (when both indirect effects were significant after FDR correction). These proteins likely reflect early, genotype-driven molecular changes that can explain the effect of *APOE* on AD.
2. Pathology-mediated proteins: Proteins meeting the reverse pattern, that is, stronger or exclusive mediation in the downstream pathway, suggesting they are predominantly regulated secondarily by AD.
3. *APOE*-specific proteins: defined as proteins significantly associated with *APOE* genotype (FDR < 0.05) but not with AD diagnosis (FDR > 0.05) and not classified into mediation-related categories. These proteins may reflect *APOE*-driven molecular changes that are not directly associated with clinical AD. Similarly, proteins associated with AD but not with *APOE* were defined as AD-specific proteins.
4. Nonspecific proteins: the remaining *APOE*-associated proteins, which were generally associated with both *APOE* and

AD but did not meet mediation criteria in either direction, possibly due to limited statistical power, modest effect sizes, or context-dependent regulation. Linear discriminant models were applied to further explore their potential classifications.

### Sensitivity analysis

Sensitivity analyses were performed in BioFINDER-2 plasma SomaLogic dataset to assess the robustness of our main findings: adjustment for population structure using the first five genetic principal components calculated in PLINK v2.0; white matter lesions and intracranial volume; medication usages across six classes including platelet inhibitors, antidepressants, anti-inflammatory, hypertension cardioprotective, lipid lowering, and ChEIs; and continuous modeling of Aβ using CSF Aβ42/Aβ40 ratio or Aβ PET rather than binary classification (see Supplementary Methods for details).

### BINN-enriched Reactome pathway analysis

BINNs[74] were used to annotate the biological functions of *APOE*-associated proteins already altered in CU or Aβ− individuals. Analyses were implemented in the Python package binn (v0.1.1), integrating the Reactome hierarchy and Deep SHAP to estimate node importance.

Only proteins associated with *APOE4* or *APOE2* in both the full cohort and the CU or Aβ− subgroup were used as input. When proteins were measured by multiple aptamers, only the aptamer with the smallest adjusted *P* value for *APOE* was retained. In GNPC, BINN models were trained to predict AD diagnosis. In BioFINDER-2 and ADNI, models were trained to predict Aβ positivity in CU and MCI individuals; AD participants were additionally included in ADNI. Data were split into training and validation sets (80:20), and models were trained for 100 epochs, or 500 epochs in ADNI.

### LDA

To evaluate whether the predefined *APOE*-associated protein categories were separable, we performed LDA in GNPC. The model used five class labels as described above: mediator proteins, pathology-mediated proteins, *APOE*-specific proteins, nonspecific proteins, and AD-specific proteins. Feature vectors were constructed from protein residuals adjusted for age, sex, mean protein level, and cohorts. The model was implemented in scikit-learn (v1.3.1) and used to derive two discriminant axes for visualization. The full proteome was then projected into this space to assess separation and within-group compactness.

### Cell type and functional enrichment analysis

To investigate the cellular and functional context of *APOE*-associated proteomic changes, we performed cell-type enrichment analyses for genes encoding proteins in each functional category. Three single-cell RNA sequencing resources were used: the ROSMAP aged prefrontal cortex atlas[75], the BBB atlas[76], and the Human Protein Atlas[77].

For the Human Protein Atlas 81-cell-type reference, enrichment was assessed using expression-weighted cell-type enrichment (EWCE[78]) R package (v1.18.0). For the ROSMAP and BBB atlases, Seurat objects were processed in Seurat (v4.3.0)[79], and average gene expression was calculated for each cell type using AverageExpression. For each protein group, the mean expression of the corresponding genes across cell types was compared with that of 10,000 random gene sets sampled from the SomaLogic background and matched for gene-set size. One-sided *P* values were calculated to test whether the observed expression exceeded that expected by chance and were adjusted for multiple comparisons using the Benjamini–Hochberg method. Unless otherwise specified, the reported results were derived from the 81-cell type atlas.

Functional enrichment analyses used GO biological process terms and were performed with enrichGO in clusterProfiler[80] v4.6.2 and org. Hs.eg.db[81] v3.16.0. To improve interpretability, analyses were performed on two complementary sets: significant proteins alone, and

significant proteins together with their first-degree protein–protein interaction partners from InWeb[82]. We focused on biological processes supported in both analyses and summarized overlapping terms into broader functional categories. Protein–protein interaction networks were additionally constructed using STRING with a minimum confidence score of 0.7. See Supplementary Methods for details.

## Regional gene expression analysis

To examine the brain relevance of *APOE*-associated proteins, we tested whether genes encoding these proteins showed spatial correlation with *APOE* expression or regional Aβ PET across the Schaefer 200 atlas. Regional gene expression data were obtained from the Allen Human Brain Atlas[83] and processed with abagen (v0.1.3)[83–85]. To account for spatial autocorrelation, null models were generated using BrainSMASH (v0.11.0)[86]. For both software, we used the default parameters and followed the same steps as described previously[46]. Nonparametric *P* values were calculated as the proportion of surrogate maps exceeding the observed correlation. *APOE4-* or *APOE2*-related proteins were considered to have gene expression profiles significantly related to *APOE* expression in the brain when BrainSMASH corrected FDR < 0.05.

## Reporting summary

Further information on research design is available in the Nature Portfolio Reporting Summary linked to this article.

## Data availability

GNPC, ADNI, UKBB, and PPMI data used in this manuscript are publicly available from the GNPC harmonized data set (https://www.neuro-proteome.org/), ADNI database (adni.loni.usc.edu), UKBB database (https://www.ukbiobank.ac.uk/), and PPMI database (https://www.ppmi-info.org/) upon request. Bulk brain transcriptomic data from the ROSMAP cohort were obtained from the AMP-AD Knowledge Portal and are publicly available through Synapse (https://www.synapse.org/), subject to data use agreements. GWAS summary data are publicly available at https://www.ebi.ac.uk/gwas/. Single-nucleus RNA sequencing from the Allen Brain Institute is openly available at https://portal.brain-map.org/atlases-and-data/rnaseq. The 81 cell type atlas is openly available at https://www.proteinatlas.org/humanproteome/single+cell/single+cell+type/data#cell_type_data. Pseudonymized data from the BioFINDER study (Principal Investigator: O.H.) are available to qualified academic researchers upon request, specifically for the purpose of replicating the analyses reported in this study. Requests should be directed to N.M.-C. (niklas.mattsson-carlgren@med.lu.se). In accordance with the EU General Data Protection Regulation, access to these data requires a data transfer agreement with Skåne University Hospital (Region Skåne), which outlines provisions for secure data handling, storage, and usage. All proposed analyses must comply with the ethical approvals granted by the Swedish Ethical Review Authority. This controlled access ensures the confidentiality of study participants, who did not consent to unrestricted public data sharing, and ensures that data use remains consistent with the scope of the original ethical approvals.

## Code availability

Essential code for the manuscript and full model results are available via GitHub at https://github.com/Lina0125/APOE_proteomics.

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

## Acknowledgements

We acknowledge all the BioFINDER team members as well as participants in the study and their family members for their dedication. The BioFINDER study was supported by the Alzheimer's Association (grant numbers ZEN 24-I 069572 to O.H., iLEADS-24-1277370 to O.H., ALZSI-26-1523522 to N.M.-C. and SG-231061717 to S.P.), European Research Council (grant number 101096455 to O.H.), Knut and Alice Wallenberg Foundation (grant number KAW 2022.0231 to O.H.), Mats Paulsson Foundation (grant number 2025-0076 to O.H.), Michael J. Fox Foundation (grant numbers MJFF-025741 to O.H. and MJFF-025507 to N.M.-C.), Region Skåne (grant numbers 2025-2026-2024-2426 to O.H. and 2025-2026-2024-2028 to N.M.-C.), Swedish Brain Foundation (grant numbers FO2024-0133-HK-46 to O.H., FO 2025-0055 and FO2025-0055 HK267 to N.M.-C., and FO 2024-0284 to S.P.), Swedish Parkinson Foundation (grant numbers 1589/24 to O.H., 1685/25 to N.M.-C., and 1698/25 to S.P.), Swedish Research Council (grant numbers 2023-06428 to O.H. and 2021-02219 and 2025-02319 to N.M.-C.), the Swedish federal government through the Avtal om Läkarutbildning och Forskning (ALF) agreement (grant numbers 2022-Project0080 to O.H. and 2022-Project0107 to N.M.-C.), Cure Alzheimer's Fund (to N.M.-C.), the Family Rönström Foundation (grant numbers FRS-0013 and AF-1011799 to N.M.-C. and AF-1011949 to S.P.), GHR Foundation (grant numbers 14358 to N.M.-C. and 13943 to S.P.), Global Research Platform, Limited Liability Company (Gates Ventures LLC) (to N.M.-C.), Greta och Johan Kocks Stiftelser (grant number F2024/228 to N.M.-C.), Skåne University Hospital's Foundations and Donations (to N.M.-C. and S.P.), Swedish Alzheimer Foundation (grant numbers A-1032795 to N.M.-C. and AF-1032524 to S.P.), Wallenberg AI, Autonomous Systems, and Software Program (WASP) and Data Driven Life Science (DDLS) (grant number WASP/DDLS22-066 to N.M.-C.), Bundy Academy (to S.P.), EU Commission: The European Partnership for Personalised Medicine (ERA PerMed) (grant number ERAPERMED2021-184 to S.P.), Greta och Johan Kocks stiftelser (to S.P.), Innovative Health Initiative (grant number 101132933 to S.P.), the Kamprad Family Foundation for Entrepreneurship, Research and Charity (grant number 20243058 to S.P.), and National Institute of Aging (grant number R01AG083740-02 to S.P.). J.W.V. was supported by the SciLifeLab & Wallenberg Data Driven Life Science Program (grant number KAW 2020.0239) and the Swedish Research Council (2024-03642). In the BioFINDER-2 cohort, the precursor of ¹⁸F-flutemetamol was sponsored by GE Healthcare. The precursor of ¹⁸F-RO948 was provided by Roche. Computational work for the GNPC cohort was supported by workspace mms-proteomics-kb from the Alzheimer's Disease Data Initiative (https://www.alzheimersdata.org). We acknowledge the use of ChatGPT (OpenAI, San Francisco, CA, USA) for assistance in grammar checking and code debugging. This research has been conducted using the UKBB Resource under Application number 105777. The funding sources had no role in the design and conduct of the study; in the collection, analysis, and interpretation of the data; or in the preparation, review, or approval of the manuscript.

## Author contributions

L.L., O.H., and N.M.-C. conceived and designed the study. L.L. drafted the manuscript with input from N.M.-C. L.L., A.P.B., I.H., B.S., and I.C.-M. performed formal analyses and visualization. V.K. and F.I. coordinated GNPC data harmonization, access, management, and computational resources. L.L., A.P.B., I.H., S.J., B.S., I.C.-M., A.V., B.M., L.A., R.O., V.K., F.I., S.P., J.W.V., E.S., O.H., and N.M.-C. contributed to data interpretation and critical revision of the manuscript. N.M.-C. supervised the work.

## Funding

## Competing interests

N.M.-C. has received speaker/consultancy fees from Biogen, BioArctic, Eli Lilly, and Owkin and Merck. O.H. is an employee of Lund University and Eli Lilly. S.P. has acquired research support (for the institution)

from ki elements/ADDF and Avid. In the past 3 years, he has received consultancy/speaker's fees from Novo Nordisk, BioArctic, Biogen, Esai, Lilly, and Roche. R.O. is currently a full-time employee of Eli Lilly and Company. His contribution to the work presented in this manuscript was performed as an employee of Amsterdam University Medical Centers and Lund University. R.O. has received research funding/support from the European Research Council, ZonMw, Nederlandse Organisatie voor Wetenschappelijk Onderzoek (NWO), National Institute of Health, Alzheimer Association, Alzheimer Nederland, Stichting Dioraphte, Cure Alzheimer's fund, Health Holland, ERA PerMed, Alzheimerfonden, Hjarnfonden, Avid Radiopharmaceuticals, Janssen Research & Development, Roche, Quanterix and Optina Diagnostics; he has given lectures in symposia sponsored by GE Healthcare, received speaker fees from Springer, and was an advisory board/steering committee member for Asceneuron, Biogen, Johnson & Johnson, and Bristol Myers Squibb. All the aforementioned has been paid to Amsterdam University Medical Centers and Lund University. B.M. and A.V. are employees of AbbVie. B.S. is an employee of Johnson & Johnson. J.W.V. has received advisory fees from Manifest Technologies within the past 2 years. The remaining authors declare no competing interests.

## Additional information

**Extended data** is available for this paper at https://doi.org/10.1038/s43587-026-01123-0.

**Correspondence and requests for materials** should be addressed to Lina Lu or Niklas Mattsson-Carlgren.

[1]Clinical Memory Research Unit, Department of Clinical Sciences Malmö, Faculty of Medicine, Lund University, Lund, Sweden. [2]Department of Physiology and Pharmacology, Université de Montréal, Montréal, Quebec, Canada. [3]Centre de recherche de l'institut universitaire de gériatrie de Montréal (CRIUGM), Montréal, Quebec, Canada. [4]Johnson & Johnson, Beerse, Belgium. [5]AbbVie, North Chicago, IL, USA. [6]Department of Clinical Sciences Malmö, SciLifeLab, Lund University, Lund, Sweden. [7]Amsterdam Neuroscience, Neurodegeneration, Vrije Universiteit Amsterdam, Amsterdam, Netherlands. [8]Alzheimer Center Amsterdam, Neurology, Vrije Universiteit Amsterdam, Amsterdam UMC location VUmc, Amsterdam, Netherlands. [9]Gates Ventures, Seattle, WA, USA. [10]Memory Clinic, Skåne University Hospital, Malmö, Sweden. *A list of authors and their affiliations appears at the end of the paper. ✉e-mail: lina.lu@med.lu.se; niklas.mattsson-carlgren@med.lu.se

### The Global Neurodegeneration Proteomics Consortium (GNPC)

**Lina Lu**[1]**, Alexa Pichet Binette**[1,2,3]**, Bart Smets**[4]**, Lijun An**[6]**, Varsha Krish**[9]**, Farhad Imam**[9]**, Jacob W. Vogel**[6]**, Oskar Hansson**[1] **& Niklas Mattsson-Carlgren**[1,10]

A full list of members and their affiliations appears in the Supplementary Information.

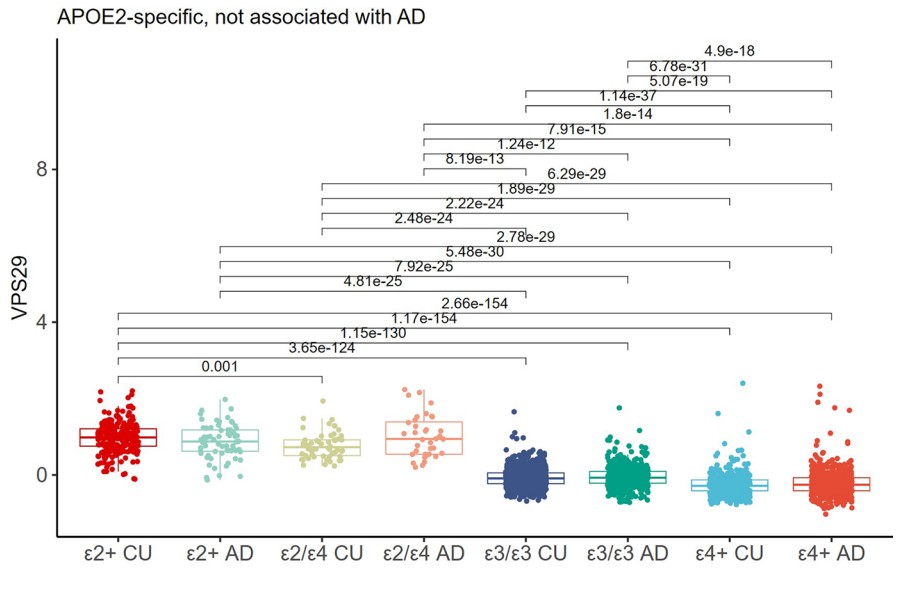

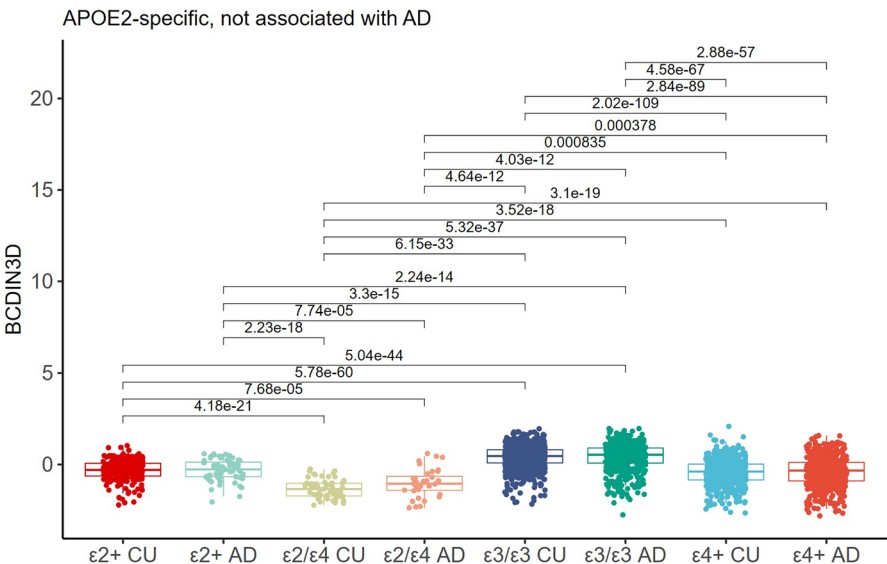

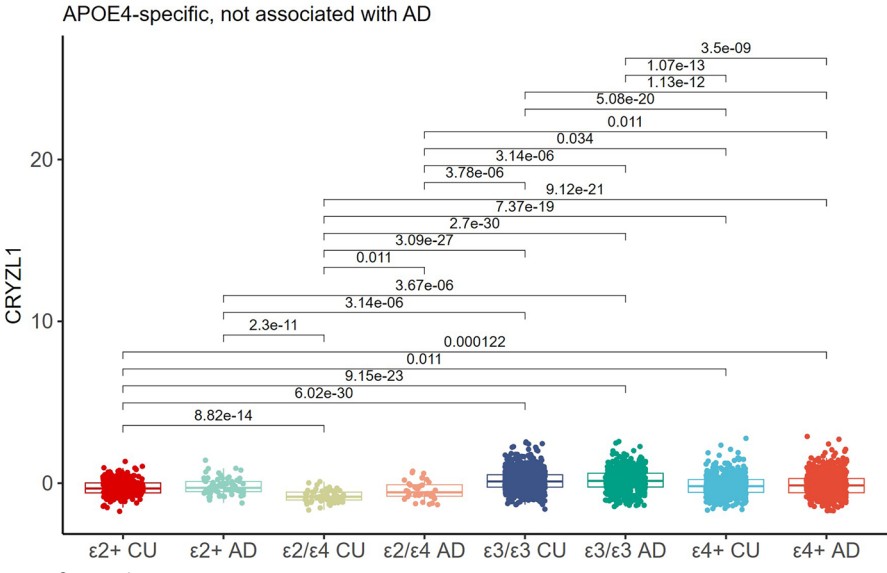

**Extended Data Fig. 1 | See next page for caption.**

**Extended Data Fig. 1 | *APOE*-specific proteomic signatures were not associated with AD (GNPC).** Box-and-whisker plots showing residual protein levels of VPS29 (top), BCDIN3D (middle), and CRYZL1 (bottom), adjusted for covariates, across *APOE* genotype-diagnosis groups (ε2+ CU, ε2+ AD, ε2/ε4 CU, ε2/ε4 AD, ε3/ε3 CU, ε3/ε3 AD, ε4+ CU, ε4+ AD). Group differences were assessed using two-sided Welch's t-tests, and P values were adjusted for multiple comparisons using the Holm–Bonferroni method. Pairwise comparisons were performed across all groups, but only significant adjusted P values are shown. *APOE2*-associated proteins (for example, VPS29 and BCDIN3D) and the *APOE4*-associated protein CRYZL1 showed consistent levels across diagnostic groups, indicating that these differences are allele-specific rather than driven by clinical diagnosis. See Source Data Extended Data Fig. 1 for detailed statistical summary.

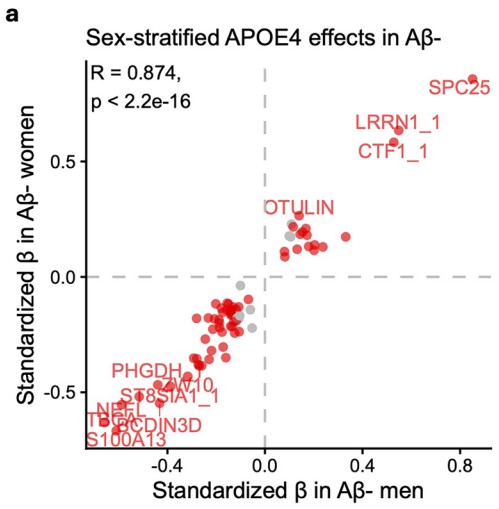

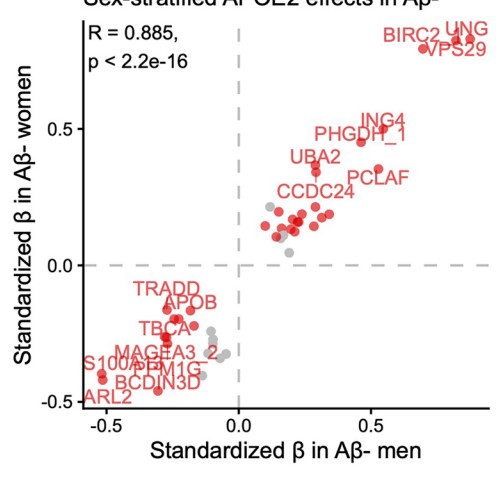

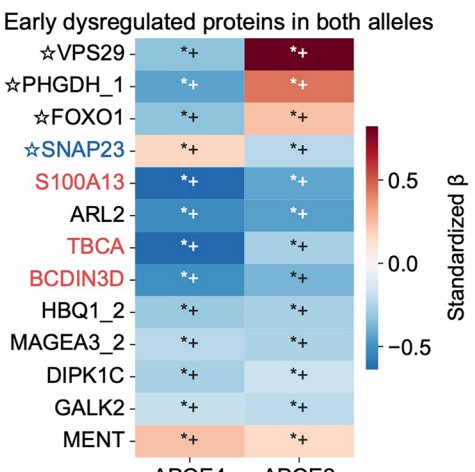

**Extended Data Fig. 2 | Sex-independent *APOE* effect at the Aβ- stages (BioFINDER-2, SomaLogic). a, b**. Sex-stratified *APOE* effect estimates for proteins altered in *APOE4* carriers (**a**) or *APOE2* carriers (**b**) at the Aβ- stage. Scatter plots compare standardized *APOE* effect sizes (standardized β) estimated separately in men (x axis) and women (y axis). Red points denote proteins significantly associated with *APOE* in both men and women in the Aβ- subgroup (N = 715 individuals). Spearman correlation was assessed using a two-sided test. **c**, Heatmap showing proteins significantly altered by both *APOE4* and *APOE2* in Aβ- individuals. Cell color indicates the effect sizes of *APOE4* (column 1) and *APOE2* (column 2), estimated from linear models adjusted for age, sex, and mean protein level. *+ denotes FDR-corrected significance. Only proteins with a mean absolute effect size > 0.2 are shown. ☆ denotes opposite effect directions for *APOE4* and *APOE2*. Red and blue labels indicate proteins identified as *APOE4* or *APOE2* mediators respectively. See Source Data Extended Data Fig. 2 for detailed statistical summary.

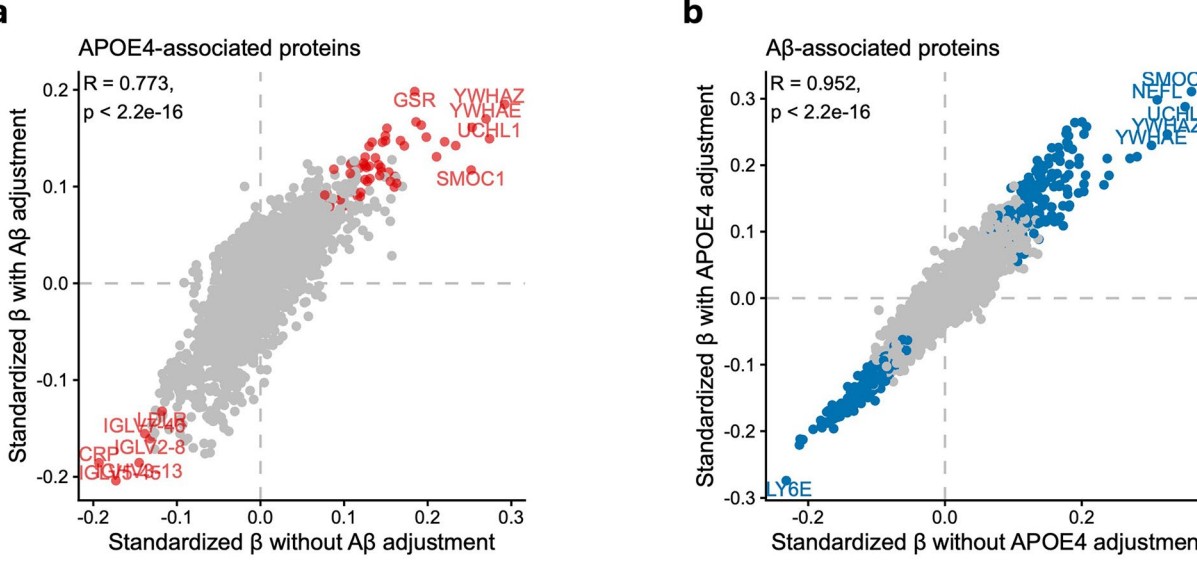

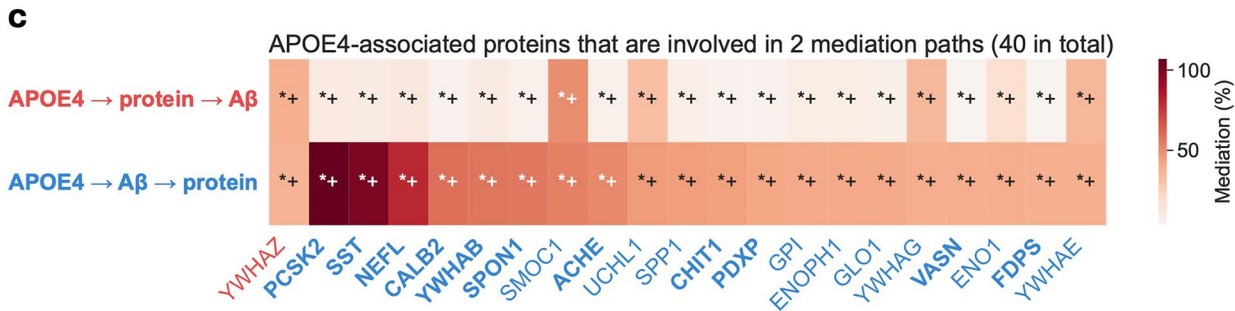

**Extended Data Fig. 3 | CSF *APOE*-proteomics signatures in TMT-MS (ADNI).**
**a**. Scatter plot comparing standardized *APOE4* effect sizes on proteins from models without (x axis) or with (y axis) adjustment for Aβ status. Proteins significantly associated with *APOE4* in both models are shown in red; all other proteins are shown in grey. Spearman correlation was assessed using a two-sided test. **b**. Scatter plot comparing standardized Aβ effect sizes on proteins from models without (x axis) or with (y axis) adjustment for *APOE4*. Proteins significantly associated with Aβ in both models are shown in blue; all other proteins are shown in grey. Spearman correlation was assessed using a two-sided test. **c**. Heatmap summarizing mediation effects and statistical significance for *APOE4*-associated proteins across two mediation pathways: the upstream

pathway (top row, *APOE4* → protein → Aβ; red label) and the downstream pathway (bottom row, *APOE4* → Aβ → protein; blue label). Cell colors represent the proportion mediated. Protein labels on the x axis are color-coded by the dominant mediation direction, with red indicating stronger upstream mediation and blue indicating stronger downstream mediation. Bold labels indicate complete mediation in the dominant pathway, whereas non-bold labels indicate partial mediation. Asterisks indicate statistical significance for mediation proportion estimation (*+ FDR-significant). For clarity, only selected proteins with the largest mediation effects from each pathway are shown; full results are provided in the Source data. See Source Data Extended Data Fig. 3 for detailed statistical summary.

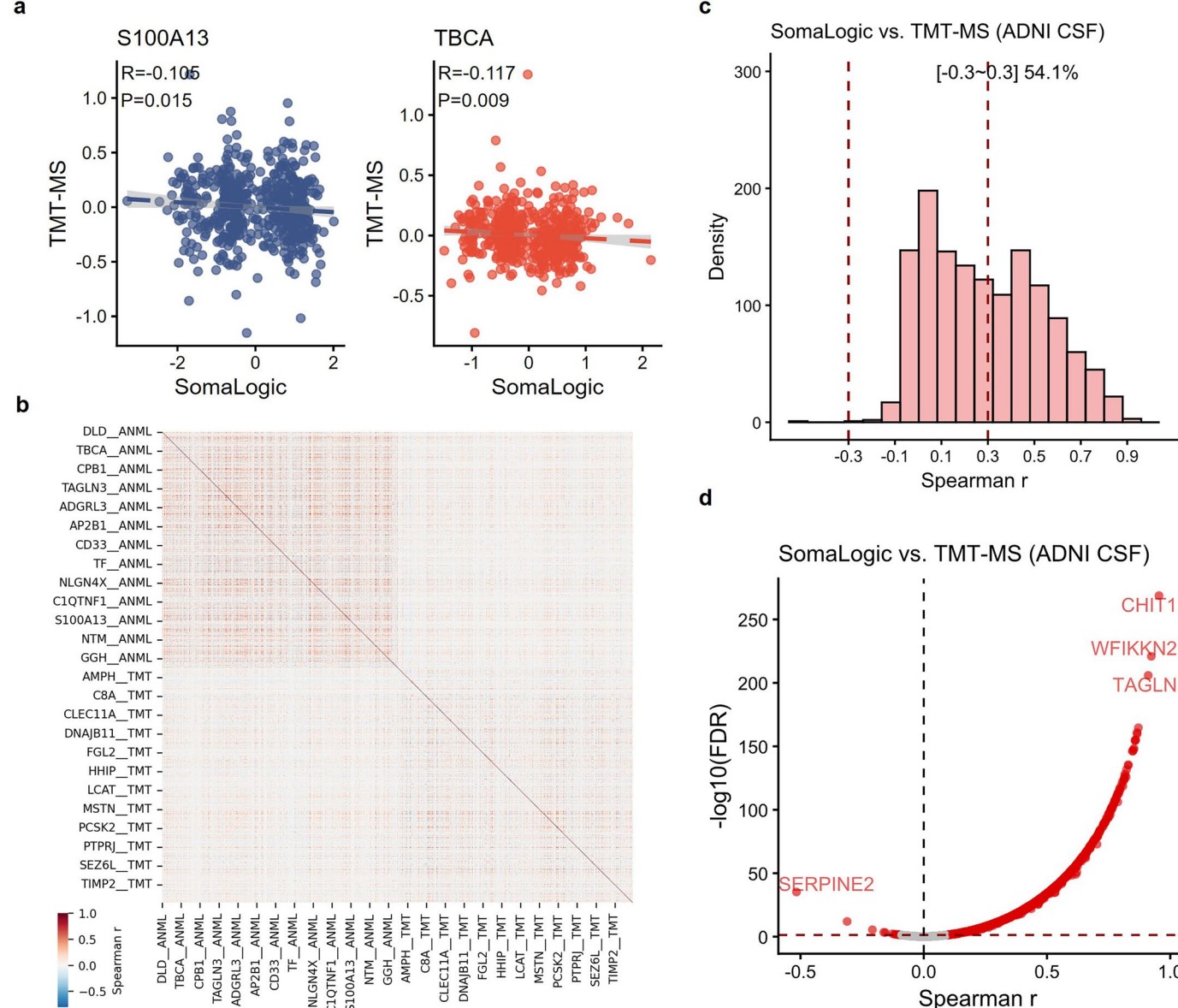

**Extended Data Fig. 4 | Direct comparison of SomaLogic vs. TMT-MS in ADNI (CSF). a.** Representative examples of pairwise correlations between SomaLogic and TMT-MS CSF measurements for selected proteins (S100A13 and TBCA) in 542 individuals from the ADNI cohort. Each dot represents one individual. Spearman correlation was assessed using a two-sided test. **b.** Heatmap of pairwise Spearman correlation coefficients between SomaLogic (ANML panel) and TMT-MS measurements for 1,360 proteins quantified by both platforms in ADNI CSF (N = 542 individuals). **c.** Distribution of pairwise Spearman correlation coefficients across proteins measured by both platforms. **d.** Volcano-style plot showing pairwise Spearman correlation coefficients (x axis) and corresponding statistical significance after FDR correction (y axis) for proteins measured by both platforms. Protein measurements were adjusted for mean protein level before correlation analysis. Similar distributions of correlation coefficients and significance were observed with and without this adjustment. See Source Data Extended Data Fig. 4 for detailed statistical summary.

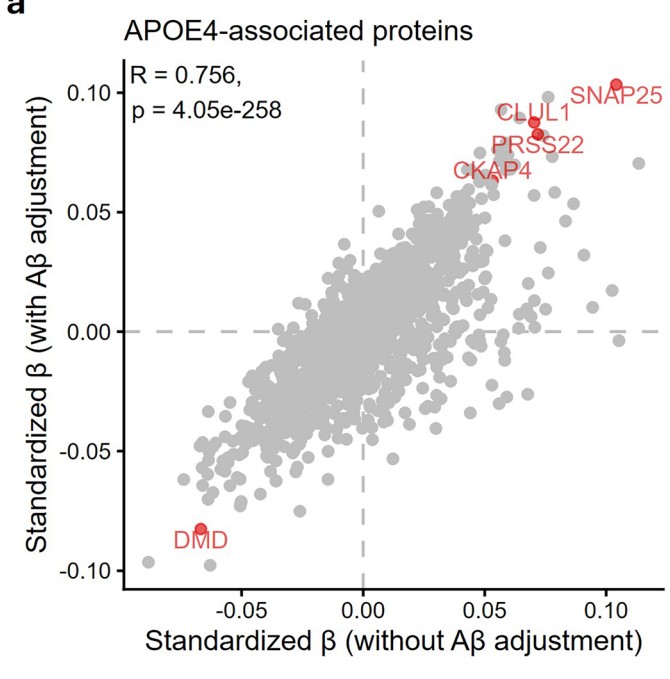

**a** APOE4-associated proteins

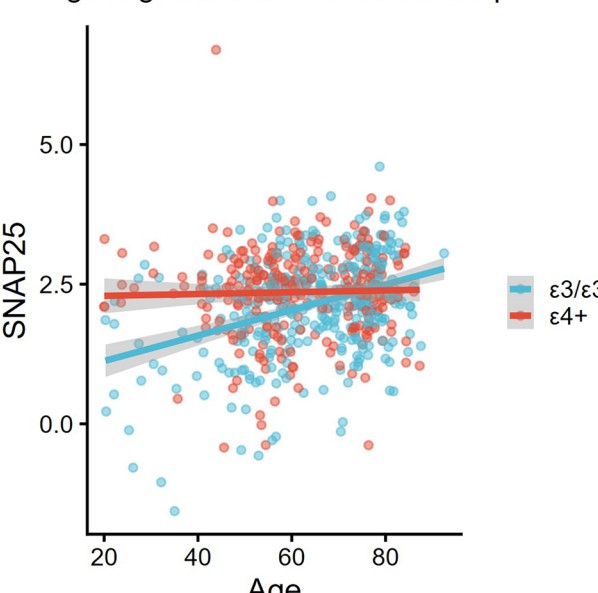

**b** Age-regulated APOE4-effect in Aβ-

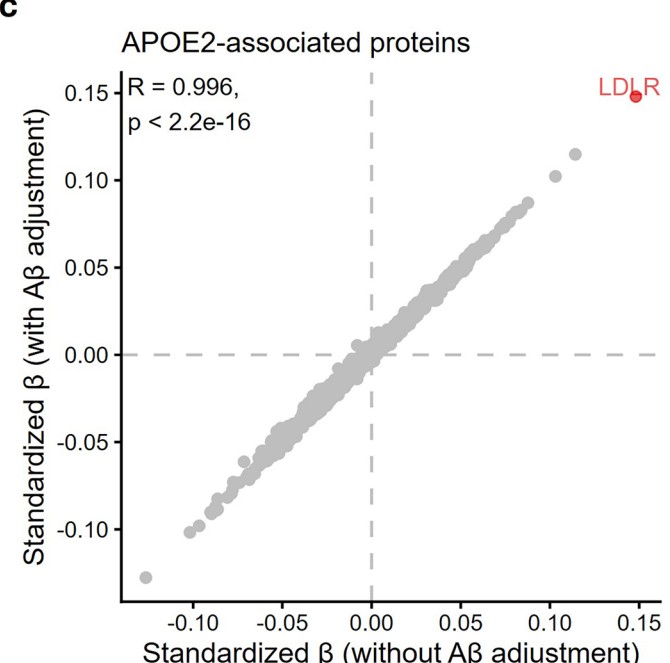

**c** APOE2-associated proteins

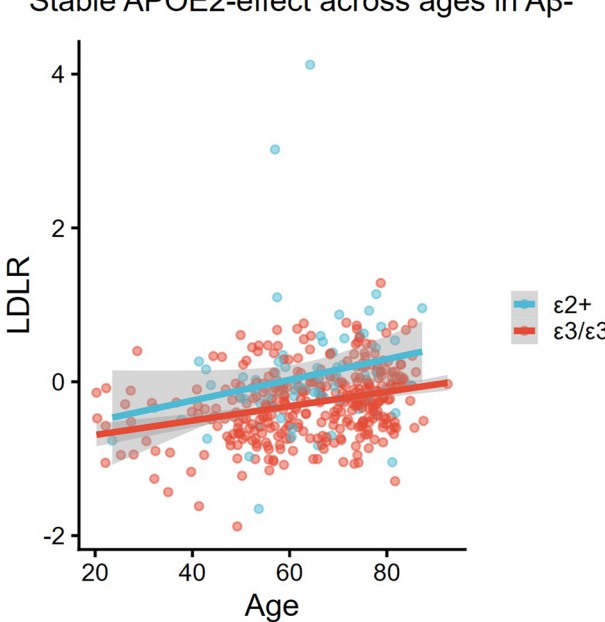

**d** Stable APOE2-effect across ages in Aβ-

**Extended Data Fig. 5 | *APOE*-proteomics changes in BioFINDER-2 CSF OLINK cohort. a**, **c**. Scatter plots comparing standardized *APOE4* (**a**) or *APOE2* (**c**) effect sizes on proteins from models without (x axis) or with (y axis) adjustment for Aβ status. Proteins significantly associated with *APOE4* (**a**) or *APOE2* (**c**) in both models are shown in red; all other proteins are shown in grey. Spearman correlation was assessed using a two-sided test. **b**. Age-dependent *APOE4* effects on SNAP25 levels in Aβ- individuals, stratified by *APOE* genotype (ε3/ε3 versus

ε4+ ). Solid lines indicate fitted regression lines and shaded bands indicate 95% confidence intervals. **d**. Stability of *APOE2* effects on LDLR levels across age in Aβ-individuals, stratified by *APOE* genotype (ε3/ε3 versus ε2+ ). Solid lines indicate fitted regression lines and shaded bands represent regression fits with 95% confidence intervals. See Source Data Extended Data Fig. 5 for detailed statistical summary.

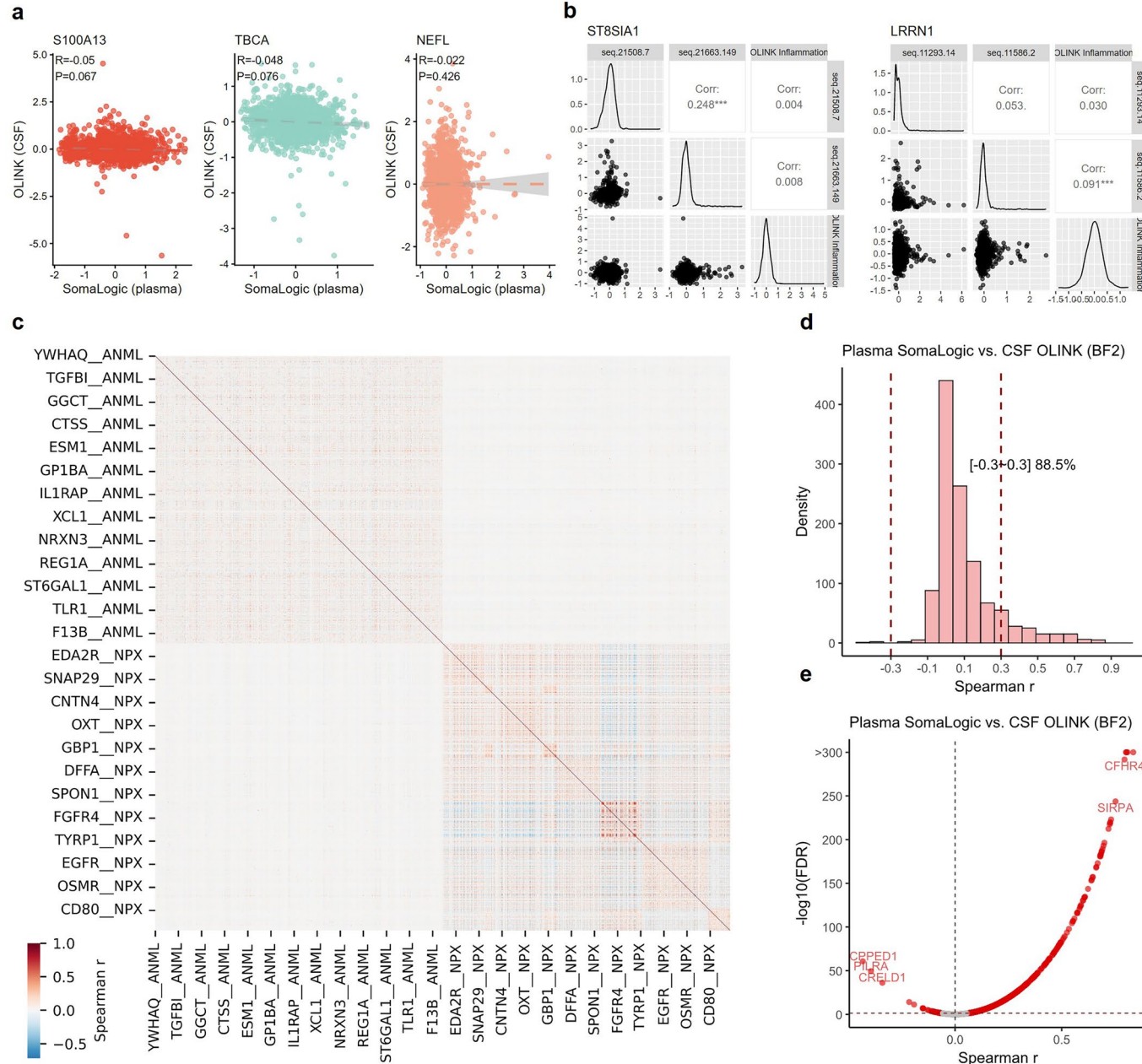

**Extended Data Fig. 6 | Direct comparison of plasma SomaLogic vs. CSF OLINK in BioFINDER-2. a.** Representative examples of pairwise correlations between plasma SomaLogic and CSF OLINK measurements for selected proteins (S100A13, TBCA and NEFL). Each dot represents one individual. Spearman correlation was assessed using a two-sided test. **b.** Scatterplot matrices with marginal density plots illustrating the relationships of ST8SIA1 and LRRN1 across plasma SomaLogic and CSF OLINK measurements. In plasma SomaLogic, each protein was measured by two separate aptamers; in CSF OLINK, both proteins were measured in the Inflammation panel. Pairwise Spearman correlations were assessed using a two-sided test. **c.** Heatmap of pairwise Spearman correlation coefficients between plasma SomaLogic (ANML panel) and CSF OLINK (NPX panel) measurements for 1,169 shared proteins quantified by both platforms in 1,349 individuals. **d.** Distribution of pairwise Spearman correlation coefficients across proteins measured by both platforms. **e.** Volcano-style plot showing pairwise Spearman correlation coefficients (x axis) and corresponding statistical significance after FDR correction (y axis) for proteins measured by both platforms. Protein measurements were adjusted for mean protein level before correlation analysis. Similar distributions of correlation coefficients and significance were observed with and without this adjustment. See Source Data Extended Data Fig. 6 for detailed statistical summary.

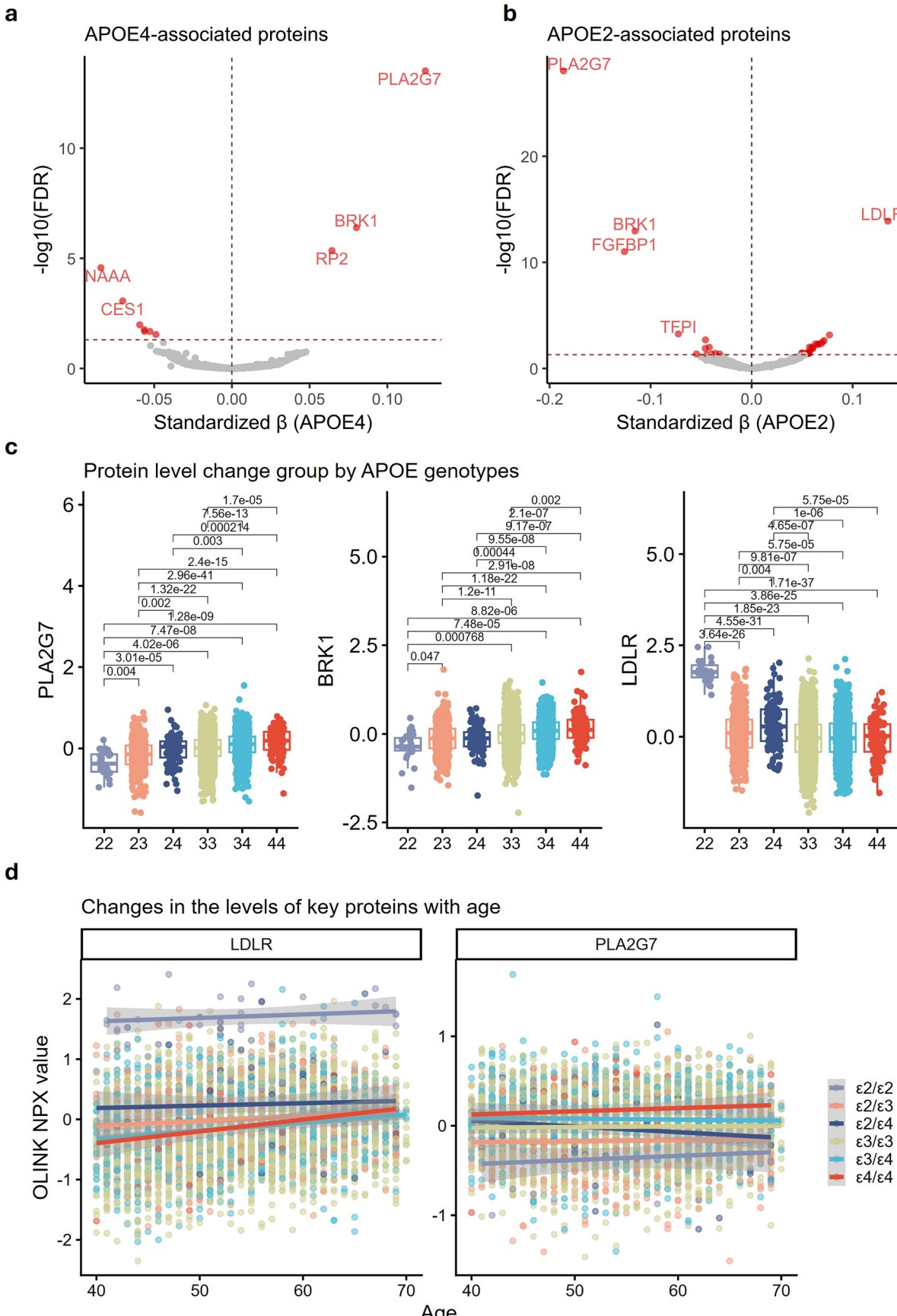

**Extended Data Fig. 7 | See next page for caption.**

**Article** https://doi.org/10.1038/s43587-026-01123-0

**Extended Data Fig. 7 | *APOE*-proteomics changes in UKBB plasma OLINK cohort.** Volcano plots of *APOE4*-associated (**a**) or *APOE2*-associated (**b**) proteins in plasma OLINK from UK Biobank (UKBB), showing standardized effect sizes (β) versus -log10(FDR) from regression models adjusted for age, sex and mean protein level. Red labels indicate proteins significant after FDR correction. **c.** Box plots showing residual protein levels of PLA2G7, BRK1 and LDLR, adjusted for age, sex and mean protein level, across *APOE* genotype groups (22 = ε2/ε2, 23 = ε2/ε3, 24 = ε2/ε4, 33 = ε3/ε3, 34 = ε3/ε4, 44 = ε4/ε4). Boxes represent the interquartile range, center lines indicate the median, whiskers extend to 1.5 × IQR, and dots represent individual samples. Group differences were assessed using two-sided Welch's t-tests, and P values were adjusted for multiple comparisons using the Holm-Bonferroni method. Pairwise comparisons were performed across all groups, but only significant adjusted P values are shown. **d.** Scatter plots showing age versus OLINK NPX values for LDLR and PLA2G7 in UKBB participants stratified by *APOE* genotype (ε2/ε2, ε2/ε3, ε2/ε4, ε3/ε3, ε3/ε4, ε4/ε4). Age at sampling is shown on the x axis and protein levels are shown on the y axis. Each dot represents one participant. Solid lines indicate fitted linear regression lines for each genotype group, and shaded bands indicate 95% confidence intervals. See Source Data Extended Data Fig. 7 for detailed statistical summary.

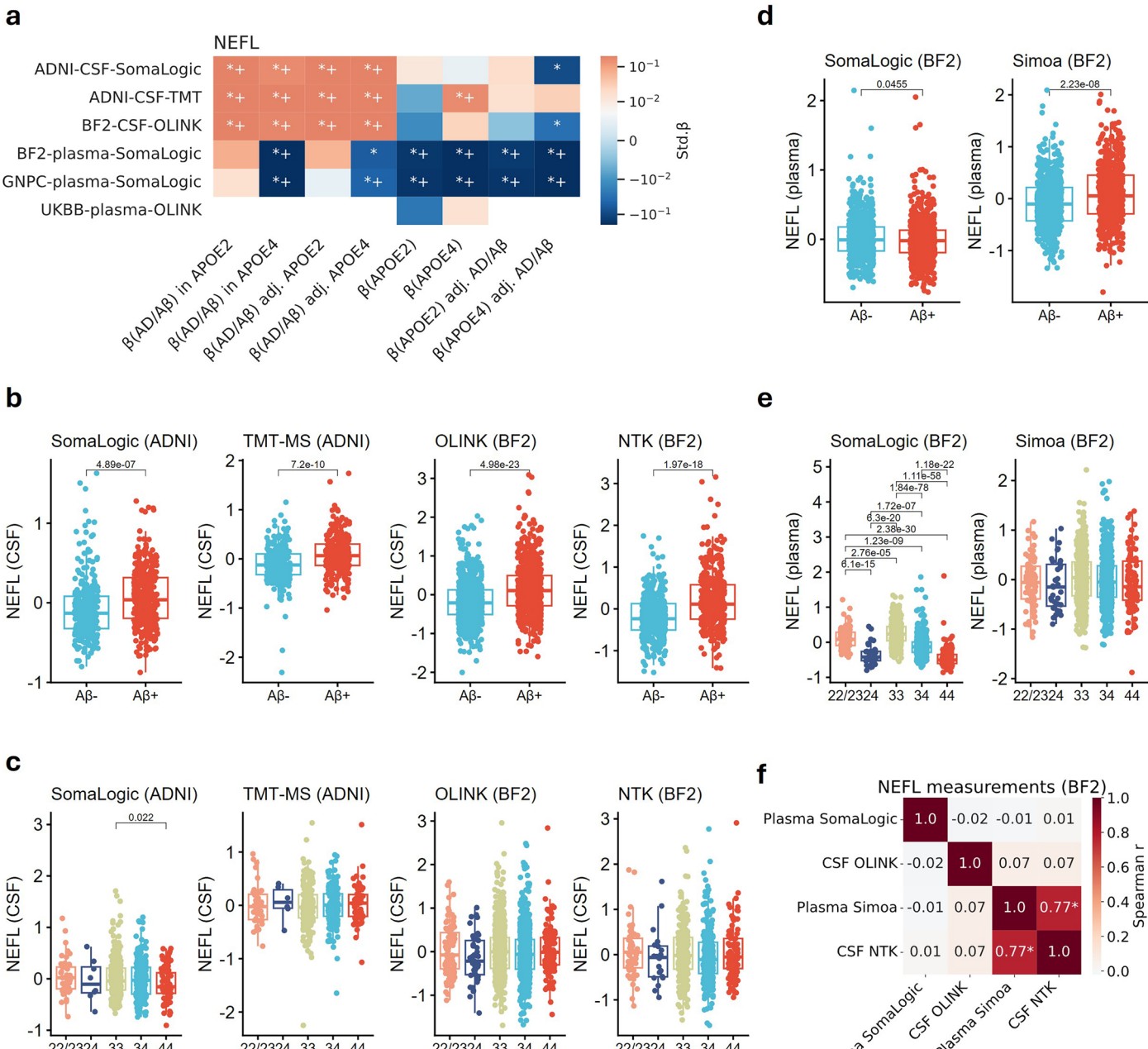

**Extended Data Fig. 8 | Comparison of NEFL across assays and datasets.**
**a**. Heatmap showing standardized regression coefficients (β) for associations of NEFL levels with *APOE* genotype and with AD or Aβ status in *APOE2*- and *APOE4*-focused analyses across six datasets. Rows show cohort-platform combinations and columns show association models. Asterisks indicate significance in each dataset (* P < 0.05; *+ FDR-adjusted P < 0.05). **b. c**. NEFL levels stratified by CSF Aβ status (**b**, Aβ- versus Aβ+) or *APOE* genotype (**c**, 22/23: ε2/ε2 and ε2/ε3; 24: ε2/ε4; 33: ε3/ε3; 34: ε3/ε4; 44: ε4/ε4) in CSF (ADNI SomaLogic, ADNI TMT-MS, BioFINDER-2 OLINK and BioFINDER-2 NTK). **d. e**. NEFL levels stratified by CSF Aβ status (**d**. Aβ- versus Aβ+) or *APOE* genotype (e. 22/23: ε2/ε2 and ε2/ε3; 24: ε2/ε4; 33: ε3/ε3; 34: ε3/ε4; 44: ε4/ε4) in plasma (BioFINDER-2 SomaLogic and BioFINDER-2 Simoa). Points represent individual participants; box plots indicate medians, interquartile ranges and 1.5 × IQR whiskers. Group comparisons were performed using two-sided Welch's t-tests on covariate-

adjusted residuals. Models were adjusted for age and sex; analyses of Aβ effects (**b**, **d**) were additionally adjusted for *APOE4* carrier status and *APOE2* carrier status, whereas analyses of *APOE* effects (**c**, **e**) were additionally adjusted for Aβ status. For proteomics-based assays (SomaLogic, OLINK, and TMT-MS), models were additionally adjusted for mean protein level. For **c** and **e**, P values were adjusted for multiple comparisons using the Holm-Bonferroni method. Pairwise comparisons were performed across all groups, but only significant adjusted P values are shown. **f**. Heatmap of Spearman correlations between NEFL levels measured across platforms in BioFINDER-2 (N = 1,547 individuals). Values and color intensity indicate correlation strength. Two-sided P values were adjusted by the Benjamini-Hochberg method; significant correlations after FDR correction are marked with an asterisk. See Source Data Extended Data Fig. 8 for detailed statistical summary.

# Reporting Summary

## Statistics

For all statistical analyses, confirm that the following items are present in the figure legend, table legend, main text, or Methods section.

| n/a | Confirmed | |
|---|---|---|
| ☐ | ☒ | The exact sample size (*n*) for each experimental group/condition, given as a discrete number and unit of measurement |
| ☐ | ☒ | A statement on whether measurements were taken from distinct samples or whether the same sample was measured repeatedly |
| ☐ | ☒ | The statistical test(s) used AND whether they are one- or two-sided<br>*Only common tests should be described solely by name; describe more complex techniques in the Methods section.* |
| ☐ | ☒ | A description of all covariates tested |
| ☐ | ☒ | A description of any assumptions or corrections, such as tests of normality and adjustment for multiple comparisons |
| ☐ | ☒ | A full description of the statistical parameters including central tendency (e.g. means) or other basic estimates (e.g. regression coefficient) AND variation (e.g. standard deviation) or associated estimates of uncertainty (e.g. confidence intervals) |
| ☐ | ☒ | For null hypothesis testing, the test statistic (e.g. *F*, *t*, *r*) with confidence intervals, effect sizes, degrees of freedom and *P* value noted<br>*Give P values as exact values whenever suitable.* |
| ☒ | ☐ | For Bayesian analysis, information on the choice of priors and Markov chain Monte Carlo settings |
| ☐ | ☒ | For hierarchical and complex designs, identification of the appropriate level for tests and full reporting of outcomes |
| ☐ | ☒ | Estimates of effect sizes (e.g. Cohen's *d*, Pearson's *r*), indicating how they were calculated |

*Our web collection on statistics for biologists contains articles on many of the points above.*

## Software and code

Policy information about availability of computer code

| Data collection | No software was used. |
|---|---|
| Data analysis | Data analysis and visualization were conducted using R (v4.4.2) and Python (v3.13.9). Key packages included interactions v1.2.0, mediation v4.5.0, EWCE v1.18.0, Seurat v4.3.0. BINN analyses were implemented using binn v0.1.1, and linear discriminant analysis was performed using scikit-learn v1.3.1. Pathway enrichment analysis used clusterProfiler v4.6.2 and org.Hs.eg.db database v3.16.0. Regional transcriptomic analyses used abagen v0.1.3. Data visualizations were generated using ggplot2 v4.0.2,or seaborn v0.13.2. All analysis code necessary to reproduce the main findings as well as model results are available in the GitHub repository (https://github.com/Lina0125/APOE_proteomics). |

For manuscripts utilizing custom algorithms or software that are central to the research but not yet described in published literature, software must be made available to editors and reviewers. We strongly encourage code deposition in a community repository (e.g. GitHub). See the Nature Portfolio guidelines for submitting code & software for further information.

# Data

Policy information about availability of data

All manuscripts must include a data availability statement. This statement should provide the following information, where applicable:
- Accession codes, unique identifiers, or web links for publicly available datasets
- A description of any restrictions on data availability
- For clinical datasets or third party data, please ensure that the statement adheres to our policy

Global Neurodegeneration Proteomics Consortium (GNPC), ADNI, UKBB and PPMI data used in this manuscript are publicly available from the GNPC harmonized data set (https://www.neuroproteome.org/), ADNI database (adni.loni.usc.edu), UKBB database (https://www.ukbiobank.ac.uk/) and PPMI database (https://www.ppmi-info.org/) upon request. Bulk brain transcriptomic data from the ROSMAP cohort were obtained from the AMP-AD Knowledge Portal and are publicly available through Synapse (https://www.synapse.org/), subject to data use agreements. SnRNA-seq from the Allen Brain Institute is openly available at https://portal.brain-map.org/atlases-and-data/rnaseq. 81 cell type atlas is openly available at https://www.proteinatlas.org/humanproteome/single+cell/single+cell+type/data#cell_type_data. GWAS summary data was publicly available in https://www.ebi.ac.uk/gwas/. BioFINDER data are available from the principal investigator (OH), pseudonymized data will be shared by request from a qualified academic investigator as long as data transfer is in agreement with EU legislation on the general data protection regulation and decisions by the Ethical Review Board of Sweden and Region Skåne, which should be regulated in a data transfer agreement. Requests should be directed to N.M.-C. (niklas.mattsson-carlgren@med.lu.se).

# Research involving human participants, their data, or biological material

Policy information about studies with human participants or human data. See also policy information about sex, gender (identity/presentation), and sexual orientation and race, ethnicity and racism.

| | |
|---|---|
| Reporting on sex and gender | Sex was self-reported and was included as a covariate in all analyses. |
| Reporting on race, ethnicity, or other socially relevant groupings | Self reported race and ethnicity were not included as confounding factors in this manuscript. In the BioFINDER-2 cohort, ancestry calculated by genetics data were used as a confounding factor in the sensitivity analysis. |
| Population characteristics | Detailed information is given in Supplement Table. 1. |
| Recruitment | Participants included in this study were drawn from six established cohorts: GNPC, BioFINDER-2, ADNI, PPMI, ROSMAP and UK Biobank. No new participants were recruited specifically for this work. GNPC: Participants were enrolled across multiple academic centers participating in the GNPC collaboration. Recruitment followed local institutional review board (IRB) approvals at each site. BioFINDER-2: The sample consisted of patients that had been referred to participating memory clinics (mostly from primary care) and most cognitively unimpaired participants were recruited from the general population in the south of Sweden between 2017 and 2023. Recruitment was conducted at Skåne University Hospital, and participants provided informed consent under the study protocol approved by the Swedish Ethical Review Authority, and they were compensated for each study visit they completed. ADNI (Alzheimer's Disease Neuroimaging Initiative):Participants were recruited across approximately 60 sites through clinical centers specializing in aging and dementia research. Enrollment included cognitively unimpaired, MCI, and AD individuals following standardized diagnostic and imaging criteria. PPMI (Parkinson's Progression Markers Initiative):Participants were enrolled from multiple international clinical sites as part of a prospective longitudinal study of Parkinson's disease and related disorders. The present analysis used a subset of PPMI participants with longitudinal cerebrospinal fluid (CSF) OLINK proteomic data and Aβ42 measurements. UK Biobank:Participants were recruited from the general population across the United Kingdom between 2006 and 2010. Data access for the current analysis was granted under approved UK Biobank Application Number 105777, and all participants provided written informed consent. ROSMAP: ROSMAP data were accessed via the AMP-AD Knowledge Portal. Participants were enrolled and followed longitudinally at Rush University Medical Center under IRB-approved protocols, with written informed consent for clinical assessment and brain donation. Initial recruitment procedures for each cohort are described in their publicly available study protocols and websites. |
| Ethics oversight | All data were obtained from existing, ethically approved cohorts. Each cohort was approved by the relevant institutional review boards or ethics committees. Participants in all studies provided written informed consent. Specifically, BioFINDER-2 was approved by the Regional Ethical Committee of Lund University, UK Biobank by the North West Multi-centre Research Ethics Committee (11/NW/0382), and other cohorts (GNPC, ADNI, PPMI, ROSMAP) under their respective IRB approvals. All analyses in the present work were performed on de-identified data, and no new data collection was conducted. |

Note that full information on the approval of the study protocol must also be provided in the manuscript.

# Field-specific reporting

Please select the one below that is the best fit for your research. If you are not sure, read the appropriate sections before making your selection.

☒ Life sciences          ☐ Behavioural & social sciences          ☐ Ecological, evolutionary & environmental sciences

For a reference copy of the document with all sections, see nature.com/documents/nr-reporting-summary-flat.pdf

# Life sciences study design

All studies must disclose on these points even when the disclosure is negative.

| | |
|---|---|
| Sample size | We did not perform an a priori sample size calculation for this study. Instead, we included all available participants with proteomic or transcriptomic, genetic, and clinical data in each cohorts. Participants were required to have OLINK, SomaLogic or TMT-MS proteomic measurements, APOE genotype information, and either Alzheimer's disease (AD) clinical diagnosis or amyloid-β (Aβ) biomarker status. In total, the study included 3,289 participants from the GNPC cohort, 1,421 participants from the BioFINDER-2 plasma SomaLogic cohort, 666 participants who have SomaLogic proteomics data and 536 participants who have tandem mass tag-based mass spectrometry (TMT-MS) proteomics data from the ADNI cohort, 1,475 participants from the BioFINDER-2 CSF OLINK cohort, 4,813 participants from the UK Biobank cohort, 253 participants who also with longitudinal CSF OLINK proteomic data and $A\beta_{42}$ measurements from the PPMI cohort, and 443 participants with bulk RNA sequencing data from the ROSMAP study. |
| Data exclusions | In the GNPC cohort, participants with multiple clinical diagnoses were excluded. Participants with a unique diagnosis of AD or cognitively unimpaired control but with cognitive scores inconsistent with the assigned diagnosis were considered likely to reflect data entry errors and were removed. Participants with >15% missing values in proteomic measurements were excluded, and proteins with >15% missingness across participants were also removed. For OLINK proteomic data, proteins for which more than 70% of participants had measurements below the limit of detection (LOD) were excluded and not included in subsequent analyses. |
| Replication | Key findings from the discovery GNPC cohort were evaluated for replication in multiple independent cohorts, including BioFINDER-2, ADNI, PPMI, and UK Biobank, covering both plasma and CSF samples and three proteomic platforms (SomaLogic, OLINK, and TMT-MS). In the discovery GNPC cohort, clinical AD diagnosis was used as the mediating anchor because Aβ biomarker data were not available. In the replication cohorts, similar analyses were performed using AD diagnosis when applicable to ensure comparability with the discovery stage. However, since Aβ represents an earlier and more specific AD-related pathology, Aβ status was used as the primary anchor for mediation analyses in replication cohorts where this biomarker was available. In these Aβ-anchored analyses, individuals with mild cognitive impairment (MCI) were included. Results obtained after including MCI participants were highly correlated with those from analyses restricted to CU and AD participants, indicating strong robustness of the findings. APOE4 and APOE2 associated proteins showed consistent direction and significance across the three SomaLogic-based cohorts (GNPC, BioFINDER-2, and ADNI), including key mediators such as SPC25, S100A13, TBCA, APOB, and PCLAF, which were further supported by genetic or transcriptomic evidence. Analyses using the OLINK platform (BioFINDER-2 and UK Biobank) and TMT-MS dataset (ADNI) demonstrated only partial concordance, greater heterogeneity was observed between the proteomic platforms. This heterogeneity likely reflects differences in proteomic technologies, coverage and assay design. Nevertheless, the overall biological patterns and allele-specific associations were reproducible and robust across analyses. |
| Randomization | This study was based on observational human cohorts; therefore, no randomization was performed. Participants were assigned to groups based on APOE genotype, Aβ status, or AD clinical diagnosis for analysis purposes only. |
| Blinding | Proteomic measurements were performed blinded to any demographics or clinical characteristics. |

# Reporting for specific materials, systems and methods

We require information from authors about some types of materials, experimental systems and methods used in many studies. Here, indicate whether each material, system or method listed is relevant to your study. If you are not sure if a list item applies to your research, read the appropriate section before selecting a response.

## Materials & experimental systems

| n/a | Involved in the study |
|---|---|
| ☐ | ☒ Antibodies |
| ☒ | ☐ Eukaryotic cell lines |
| ☒ | ☐ Palaeontology and archaeology |
| ☒ | ☐ Animals and other organisms |
| ☐ | ☒ Clinical data |
| ☒ | ☐ Dual use research of concern |
| ☒ | ☐ Plants |

## Methods

| n/a | Involved in the study |
|---|---|
| ☒ | ☐ ChIP-seq |
| ☒ | ☐ Flow cytometry |
| ☐ | ☒ MRI-based neuroimaging |

## Antibodies

| | |
|---|---|
| Antibodies used | No individual antibodies were used in this study. Proteomic measurements were generated using high-throughput affinity-based |

| Antibodies used | platforms (SomaLogic 7k and OLINK Explore), which utilize proprietary antibody or aptamer-based reagents for protein quantification according to the manufacturers' standardized protocols. |
| Validation | All antibody and aptamer reagents used in the commercial proteomic platforms (SomaLogic 7k and OLINK Explore) have undergone manufacturer validation for specificity, reproducibility, and technical performance, as reported in their technical white papers and previous peer-reviewed studies. No additional in-lab antibody validation was performed. |

# Clinical data

Policy information about clinical studies

All manuscripts should comply with the ICMJE guidelines for publication of clinical research and a completed CONSORT checklist must be included with all submissions.

| Clinical trial registration | This study did not involve a prospective clinical trial and therefore was not registered as one. Clinical, proteomic or transcriptomic data were obtained from previously established observational cohorts, including GNPC, BioFINDER-2, ADNI, PPMI, ROSMAP and UK Biobank. Each contributing cohort has its own ethical approval and, where applicable, clinical registration (e.g., ADNI: ClinicalTrials.gov NCT00106899; PPMI: NCT01141023; BioFINDER-2: NCT03174938). |
| Study protocol | No new study protocol was generated for this work. Analyses were based on data from multiple established cohorts, each conducted under its own approved study protocol. The GNPC operates under a consortium-level research protocol approved by the institutional review boards of participating sites. The methodological details of GNPC are described in related GNPC publications (DOI: https://doi.org/10.1038/s41591-025-03834-0). |
| | Study protocols for other contributing cohorts are publicly available: BioFINDER-2 (www.biofinder.se), ADNI (adni.loni.usc.edu), PPMI (www.ppmi-info.org), ROSMAP (https://adknowledgeportal.synapse.org/Explore/Studies/DetailsPage/StudyDetails?Study=syn3219045), and UK Biobank (www.ukbiobank.ac.uk). |
| Data collection | Proteomic, transcriptomic, genetic, and clinical data were obtained from multiple established cohorts, including the GNPC, BioFINDER-2, ADNI, PPMI, ROSMAP, and UK Biobank. All data were collected under cohort-specific, ethically approved study protocols and were de-identified prior to analysis. |
| | Proteomic profiling was performed using affinity-based platforms (SomaLogic 7k and OLINK Explore) or mass spectrometry-based approaches (tandem mass tag-based mass spectrometry, TMT-MS), following each cohort's standardized sample processing, quality control, and normalization procedures. |
| | In the GNPC cohort, plasma samples and associated clinical data were collected across participating academic centers as part of an IRB-approved consortium protocol. |
| | The BioFINDER-2 study includes both population-based participants and individuals recruited from memory clinics in southern Sweden; biological samples and imaging data were collected at Skåne University Hospital in Lund and Malmö between April 2017 and December 2023. |
| | The ADNI cohort is a multi-site longitudinal study primarily recruiting through academic research centers, with standardized clinical assessments, biospecimen collection, and imaging protocols, and exclusion of major comorbid conditions. Both SomaLogic and TMT-MS proteomic datasets generated within ADNI were included in the present analyses. |
| | Longitudinal CSF proteomic data from the PPMI were obtained from Project 9000, with detailed study procedures described in the publicly available PPMI protocol. |
| | ROSMAP data were obtained through the Accelerating Medicines Partnership-Alzheimer's Disease (AMP-AD) Knowledge Portal. ROSMAP is a longitudinal study conducted by Rush University Medical Center that enrolled older adults from religious communities and the general population, who underwent annual clinical evaluations and agreed to brain donation at death. Bulk RNA sequencing data from postmortem brain tissue, along with corresponding APOE genotyping and clinical data, were used in the present study. All ROSMAP data were generated and released under IRB-approved protocols and accessed through controlled data use agreements. |
| | UK Biobank is a large-scale, multicenter prospective cohort study that enrolled approximately 500,000 participants aged 40-69 years across the United Kingdom between 2006 and 2010. Biological samples, genetic data, and longitudinal health-related information were collected using standardized procedures, as described in the UK Biobank study protocol. |
| | In all cohorts, written informed consent was obtained from participants or their legal representatives, as applicable, and all analyses were conducted using de-identified data. |
| Outcomes | The primary outcomes were the associations between APOE genotypes (ε4 and ε2) and plasma or CSF protein levels across multiple cohorts. Mediation outcomes included both the indirect effects of APOE genotypes on AD diagnosis and Aβ pathology through protein biomarkers (APOE => protein => AD/Aβ), and the reverse mediation effects of APOE genotypes on protein levels through AD diagnosis or Aβ pathology (APOE => AD/Aβ => protein). Based on these mediation results, proteins were categorized into distinct biological groups, including APOE-specific proteins, AD(or Aβ)-specific proteins, upstream mediator proteins, and downstream mediated proteins. |
| | Secondary outcomes included pathway and cell-type enrichment analyses of these protein groups, as well as evaluation of their associations with downstream AD-related phenotypes, including Aβ positron emission tomography (PET), tau PET, cortical thickness measures, and cognitive performance. |
| | In addition, brain transcriptomic analyses were conducted to assess co-expression patterns between APOE and protein-coding genes in the brain. Using ROSMAP bulk brain transcriptomic data, we further examined whether APOE genotype was associated with gene expression levels of mediator proteins and checked the presence of AD-associated genetic variants within these genes reported in previous GWAS studies, thereby assessing their potential central nervous system relevance and genetic support. |

# Plants

| | |
|---|---|
| Seed stocks | *Report on the source of all seed stocks or other plant material used. If applicable, state the seed stock centre and catalogue number. If plant specimens were collected from the field, describe the collection location, date and sampling procedures.* |
| Novel plant genotypes | *Describe the methods by which all novel plant genotypes were produced. This includes those generated by transgenic approaches, gene editing, chemical/radiation-based mutagenesis and hybridization. For transgenic lines, describe the transformation method, the number of independent lines analyzed and the generation upon which experiments were performed. For gene-edited lines, describe the editor used, the endogenous sequence targeted for editing, the targeting guide RNA sequence (if applicable) and how the editor was applied.* |
| Authentication | *Describe any authentication procedures for each seed stock used or novel genotype generated. Describe any experiments used to assess the effect of a mutation and, where applicable, how potential secondary effects (e.g. second site T-DNA insertions, mosiacism, off-target gene editing) were examined.* |

# Magnetic resonance imaging

## Experimental design

| | |
|---|---|
| Design type | MRI data were used only in the BioFINDER-2 cohort to examine associations between APOE-related plasma proteins and cortical thickness as a downstream neurodegeneration marker of AD pathology |
| Design specifications | Cross-sectional observational design |
| Behavioral performance measures | *State number and/or type of variables recorded (e.g. correct button press, response time) and what statistics were used to establish that the subjects were performing the task as expected (e.g. mean, range, and/or standard deviation across subjects).* |

## Acquisition

| | |
|---|---|
| Imaging type(s) | T1-weighted structural MRI for cortical thickness measurement |
| Field strength | 3 Tesla |
| Sequence & imaging parameters | Acquisitions used a 3-D magnetization-prepared rapid gradient echo (MPRAGE) sequence, with 1 mm isotropic voxel resolution. |
| Area of acquisition | A cortical thickness meta-ROI was calculated including entorhinal, inferior temporal, middle temporal and fusiform using FreeSurfer (version 6.0) parcellation, which are areas known to be susceptible to AD-related atrophy |

Diffusion MRI ☐ Used ☒ Not used

## Preprocessing

| | |
|---|---|
| Preprocessing software | FreeSurfer (version 6.0; https://surfer.nmr.mgh.harvard.edu) |
| Normalization | *If data were normalized/standardized, describe the approach(es): specify linear or non-linear and define image types used for transformation OR indicate that data were not normalized and explain rationale for lack of normalization.* |
| Normalization template | *Describe the template used for normalization/transformation, specifying subject space or group standardized space (e.g. original Talairach, MNI305, ICBM152) OR indicate that the data were not normalized.* |
| Noise and artifact removal | *Describe your procedure(s) for artifact and structured noise removal, specifying motion parameters, tissue signals and physiological signals (heart rate, respiration).* |
| Volume censoring | *Define your software and/or method and criteria for volume censoring, and state the extent of such censoring.* |

## Statistical modeling & inference

| | |
|---|---|
| Model type and settings | Linear regression models were applied with cortical thickness as the dependent variable and APOE-related protein levels as the primary predictors. All models were adjusted for age, sex, and mean protein level to control for individual differences in global protein abundance. Outliers (> 5 SD from the mean) were removed. Analyses were conducted separately within Aβ-positive and Aβ-negative subgroups to examine potential effect modification by amyloid status. All statistical analyses were performed in R (version 4.4.2), and multiple testing across proteins was corrected using the Benjamini-Hochberg false discovery rate (FDR < 0.05) procedure. |
| Effect(s) tested | The effect size and significance of APOE-related plasma protein levels on cortical thickness, to determine whether APOE-associated proteins are linked to downstream neurodegeneration and late-stage AD pathology. |

Specify type of analysis: ☐ Whole brain ☒ ROI-based ☐ Both

| | Anatomical location(s) | A cortical thickness meta-ROI was calculated including entorhinal, inferior temporal, middle temporal and fusiform, which are areas known to be susceptible to AD-related atrophy |
|---|---|---|

Statistic type for inference

(See Eklund et al. 2016)

*Specify voxel-wise or cluster-wise and report all relevant parameters for cluster-wise methods.*

Correction

FDR correction for statistical results

## Models & analysis

| n/a | Involved in the study |
|---|---|
| ☒ | ☐ Functional and/or effective connectivity |
| ☒ | ☐ Graph analysis |
| ☐ | ☒ Multivariate modeling or predictive analysis |

Multivariate modeling and predictive analysis

Independent variables: APOE-related plasma protein levels.
Dependent variable: Cortical thickness derived from T1-weighted MRI in predefined cortical ROIs (including entorhinal, inferior temporal, middle temporal and fusiform).
Model type: Multiple linear regression models adjusted for age, sex, and mean proteomic level.
Feature extraction / dimension reduction: ROI-based cortical thickness measures were processed using FreeSurfer; no additional dimensionality-reduction techniques were applied.
Training and evaluation: Analyses were performed in the BioFINDER-2 cohort and stratified by Aβ status to examine effect modification. Model robustness was evaluated by FDR-corrected significance (q < 0.05)

