## [Peer Review File · Nature Aging]

Proteomic signatures of the APOE ϵ 4 and APOE ϵ 2 genetic variants and Alzheimer's disease

Corresponding Author: Dr Niklas Mattsson-Carlgrén

Version 0:

Reviewer comments:

Reviewer #1

(Remarks to the Author)

Lu et al. analyzed the effects of APOE genotype on plasma proteomics using the SomaLogic platform, with particular emphasis on APOE2-related mechanisms. Through mediation analyses, they nominated candidate proteins as potential upstream or downstream mediators. The authors also addressed previous concerns by summarizing the diagnostic criteria across AD cohorts, examining how APOE-associated proteins change with age and their relationship to AD, highlighting differences between CSF and plasma, and discussing platform heterogeneity. The discussion section has addressed these issues and highlighted their limitations.

Platform heterogeneity is a major limitation. Therefore, the authors also provided supplementary analyses at the genetic and RNA levels. To explore potential causal links, the authors examined GWAS data to investigate if coding variants in genes encoding key mediator proteins are associated with A β or AD risk. To make these results stronger, it would be useful for the authors to report the effect size estimates and significance of certain SNPs. Additionally, the authors could test whether these SNPs interact with APOE in influencing AD risk.

The authors also supplemented their data with brain transcriptome and examined the spatial co-expression of APOE expression. This analysis demonstrates that certain proteins exhibit spatial expression patterns related to APOE. It would be interesting to know if the authors considered whether different APOE genotypes influence the transcriptional levels of these proteins at the RNA level.

Minor concerns:

1. On page 7, line 188, the authors mention "192 APOE4-associated proteins." This should probably refer to APOE2 instead of APOE4.
2. The race and ethnicity bias remain insufficiently addressed. The authors should include them as a covariate to further enhance the validity of the study, or mention this as a limitation of study if full demographics are not available for all cohorts.

(Remarks on code availability)

Reviewer #2

(Remarks to the Author)

As a disclaimer, I was reviewer of the previous version of this article submitted originally to Nature Medicine and my assessment reflects that fact. Lu et al. present a modified version of their study about the proteomic signature of APOE4 and APOE2 in the blood and the CSF of Alzheimer's disease patients. For this purpose, they bring together data collected from the GNPC, BioFINDER-2, ADNI, UKBB, and the PPMI studies. This effort brings together more than 10000 samples including both, plasma and CSF samples, studied using affinity-based platforms SomaLogic (aptamers) and Olink (DNA-labeled antibodies). Either the presence of Abeta (detected either in CSF or PET) or a diagnosis of AD were considered as discriminating factors for APOE4 or APOE2 carriers in contrast to cognitively unimpaired subjects. An abundance analysis was conducted and associated to, or mediator proteins were identified for, each APOE haplotype. Key mediator proteins were identified for APOE4 such as ZW10, S100A13, SPC25, TBCA and ARL2, and for APOE2 such as APOB, PCLAF, SNAP23 and WARS2. Finally, co-expressors of APOE in brain tissue were identified among the proteins detected during

the analysis, using publicly available RNA microarray data.

As in the previous version of the manuscript, the analysis is well conducted. The manuscript is clearly written, and the results are well presented. I do see differences in the data reported, perhaps influenced by some changes in number of cases taken from each study, and perhaps due to some modifications on the analysis. The document has gained on the description and interpretation of the findings and a more balanced presentation of ApoE haplotypes effects is now presented.

Some issues remain. Authors still use what they call “A β status” or AD diagnosis as absolute values to assess the impact of ApoE haplotypes in protein levels. This “A β status” stated as either positive or negative, either derives from CSF A β 42/A β 40 ratio in the BioFINDER-2 cohort patients, or from A β -PET status in the ADNI cohort. Up to which point CSF A β levels are compatible with A β -PET measurements to be comparable in their definition of A β positivity, remains to be clarified. Furthermore, both are continuous variables and similarly to the classification of “AD diagnosis”, include individuals at different stages of progression and severity of AD. This fact is not addressed in the analysis, both “A β status” and AD diagnosis are considered as binary categorical variables. This might be beneficial from an analytical point of view, but it might not reflect the actual complexity of the disease. It might be worthwhile to assess them as the continuous variables they are and see how it affects the analysis.

Another issue is the lack of validation of the findings using Olink and Somalogic, beyond their internal validation. A low correlation between both approaches and MS proteomics has been previously reported (10.1101/2025.02.14.638375). In this version of the manuscript the authors mention this issue as one out of several limitations of the study. It is clear for this reviewer that a full orthogonal validation, hopefully including actual MS proteomics, of a study of this scale and incorporating data from several cohorts, is impractical. But the authors have strong interesting claims about the differences between ApoE haplotypes that can be validated in at least in a subset of individuals of the cohort more accessible for them. I invite them to consider doing this.

Finally, also acknowledged by the authors as a limitation, proteins identified in this analysis can have different biological sources at a cellular level and reflect various aspects of the pathology, directly or indirectly influenced by a particular ApoE haplotype. So, even though the study provides a very descriptive picture of the apparent effect of ApoE2 and ApoE4 haplotypes in plasma and CSF proteomics, the absence of mechanistic supportive evidence renders the biological significance of the findings incomplete. Therefore, despite the scale of the study and the complexity of the analysis, I see the contribution to the field as only incremental.

(Remarks on code availability)

Reviewer #3

(Remarks to the Author)

Most of the points raised in my earlier review have now been addressed to the extent permitted by the available datasets, and the authors have substantially strengthened the manuscript. I appreciate the clarifications added regarding platform heterogeneity, cross-fluid inconsistencies, tissue specificity, and causal interpretation, as well as the improved framing of the conclusions. Overall, the revisions meaningfully enhance the rigor and transparency of the work, and I congratulate the authors on their thorough and thoughtful responses.

I have only a few remaining suggestions, all of them minor but important for clarity:

- Lines 363–366: I do not fully understand the statement “These asymmetric effects may help explain the divergent AD risks of the two alleles,” particularly because two sentences earlier it is stated that “only a minority showed opposite regulation.” Please clarify the logic and further elaborate on how asymmetry, despite limited opposite regulation, supports divergent risk profiles. This is central to the biological interpretation. It would benefit from substantial expansion in the discussions to help the reader appreciate its implications.
- In the CSF section (lines 498 and following), a description of overlap or lack of overlap with plasma findings is missing; adding this would strengthen interpretability. A related question: “SPC25 mediated APOE4–AD associations in plasma but not in CSF.” Some discussion on how to interpret this divergence would be valuable.
- The statement that “Together, these results reflect a molecular signature of OLINK that is distinct from SomaLogic in CSF” has major implications and should be more explicitly discussed, particularly in light of the relatively low number of proteins identified in UK Biobank.
- Similarly, the NEFL discrepancies across platforms (“positively correlated in CSF... negatively associated in plasma SomaLogic... not replicated in OLINK or Simoa”) highlight a broader issue: to what extent should proteomic associations, especially platform-specific ones, be interpreted with caution? Further reflection on this point would strengthen the discussion.
- In the section on genetic evidence supporting mediator proteins, it would be helpful to add the strength of association (effect sizes or p-values) shown in Figure 6c to contextualize the relevance of the findings.
- For the reasons noted above, the last sentence in the first paragraph of the Discussion might be nuanced, acknowledging that the support is variable across platforms, tissues, and datasets.
- The finding that ChEI use was linked to higher SPC25 levels only in APOE4 carriers is particularly important. Given that many centers prescribe ChEIs in MCI based on the positive APOE4 subgroup signal in the pivotal trials (which lacked biomarkers), this result raises the question of whether the previous signal reflected AD-related biology or simply APOE-driven effects. This deserves further discussion.

Overall, these are relatively minor clarifications in an otherwise strong and much-improved manuscript.

(Remarks on code availability)

Reviewer #4

(Remarks to the Author)

The APOE locus is the strongest genetic factor for Alzheimer's disease (AD), with $\epsilon 4$ (APOE4) increasing and $\epsilon 2$ (APOE2) decreasing risk, yet the molecular programs reflecting these divergent effects remain unclear. In this study, the authors conducted a large-scale, multi-cohort proteomic analysis across plasma and cerebrospinal fluid (CSF), integrating five independent datasets including the GNPC, BioFINDER-2, ADNI, UKBB, and PPMI, and two affinity-based platforms (SomaLogic, OLINK).

Utilizing amyloid β (A β) or clinical AD anchored upstream (APOE => protein => A β /AD) and downstream (APOE => A β /AD =>protein) mediation framework, multi-cohort design, and verification via GWAS and brain transcriptomics, the authors systematically mapped APOE isoform-specific proteomic signatures. They found that proteomic alterations linked to APOE2 and APOE4 were largely detectable in preclinical stages, even before detectable A β pathology, and remained relatively stable across aging and disease stages.

In APOE2 carriers, most dysregulated proteins were involved in DNA repair, mitochondrial metabolism, proteostasis, and anti-inflammatory regulation, functioning largely as upstream mediators; only a minority showed limited remodeling by pathological cascades, consistent with systemic resilience. In APOE4 carriers, a small subset of proteins enriched in cell-cycle and oligodendrocyte precursor cells (OPCs) may help explain its association with AD risk, whereas a broader set linked to vascular remodeling, immune activation, and proteostasis loss underwent pronounced downstream pathology-driven remodeling, reflecting a shift from genotype-related to pathology-driven regulation.

Their comparative analyses identified three key contributors reflecting divergent AD risk: allele-specific mediators supported by genetic or transcriptomic evidence (e.g., SPC25 for APOE4 , APOB and SNAP23 for APOE2), oppositely regulated "switch-node" proteins (e.g., VPS29, PHGDH), and a broader set of concordantly regulated proteins showing $\epsilon 4$ -dominant effects (e.g., S100A13, TBCA, ARL2).

A key finding of this comprehensive study is that the alteration of these proteins were already evident in A β -negative individuals, underscoring their upstream, genotype-driven nature. Collectively, their findings indicate that asymmetric molecular architectures, rather than mirror-image regulation, underlie the divergent AD risks of APOE2 and APOE4 , and nominate upstream pathways and genetically or transcriptomically supported proteins as promising biomarkers and potential targets for allele-tailored interventions in AD.

Overall, this is an outstanding study with excellent study design, multi- and complementary human cohorts, multi-platforms of samples, and various analytic approaches. The data strongly support the main conclusions. Their important findings point to upstream and allele-specific proteins already altered in A β -negative individuals as promising biomarkers and targets for early detection and preventive interventions.

This reviewer only has the following minor comments to help make the paper stronger and more solid.

- 1) Although it is not necessary, it would be nice to see whether there are any sex effects on the major altered proteins in both APOE2 and APOE4 carriers, especially the mediators of APOE2 and APOE4 in Abeta-negative individuals.
- 2) For major analyses, the statistical methods and n numbers should be provided in each figure legend.
- 3) Line 188, should it be "192 APOE2-associated proteins"?

(Remarks on code availability)

Version 1:

Reviewer comments:

Reviewer #1

(Remarks to the Author)

Overall, the authors have largely addressed the reviewer's concerns and provided a detailed description of cross-platform validation across proteomic technologies (Olink, SomaLogic, and TMT-MS). The inclusion of additional cohort data (ROSMAP), the accompanying and other related analyses are comprehensive. The manuscript is clearly written and highlights an interesting asymmetric protein profile between APOE2 and APOE4 effects. The authors have also carefully discussed the limitations of the study.

Nevertheless, platform heterogeneity remains a potential key limitation, as it may hinder the reproducibility of certain protein-level findings across technologies and complicate subsequent functional interpretation using complementary experimental models. In addition, APOE4-related proteomic signatures have been reported in several prior studies, which may limit the novelty of this work to a certain extent.

Minor concerns:

In several cohorts, the CSF A β 42/40 ratio is binary (A β -/A β +) and used for both discovery and validation. Please briefly clarify the exact thresholds applied in each cohort, the source/rationale for these cutoffs, and whether they were validated or calibrated against A β -PET or other biomarker analysis. Please also include relevant references accordingly.

Reviewer #4

(Remarks to the Author)

The authors have addressed very well all of my comments. I do not have any further comments.

Response letter

Reviewer #1:

Lu et al. analyzed the effects of APOE genotype on plasma proteomics using the SomaLogic platform, with particular emphasis on APOE2-related mechanisms. Through mediation analyses, they nominated candidate proteins as potential upstream or downstream mediators. The authors also addressed previous concerns by summarizing the diagnostic criteria across AD cohorts, examining how APOE-associated proteins change with age and their relationship to AD, highlighting differences between CSF and plasma, and discussing platform heterogeneity. The discussion section has addressed these issues and highlighted their limitations.

Response: We thank the reviewer for the careful evaluation of our revised manuscript and for the constructive comments, which have helped us further strengthen the rigor and clarity of the study. We address each point below.

Platform heterogeneity is a major limitation. Therefore, the authors also provided supplementary analyses at the genetic and RNA levels. To explore potential causal links, the authors examined GWAS data to investigate if coding variants in genes encoding key mediator proteins are associated with A β or AD risk. To make these results stronger, it would be useful for the authors to report the effect size estimates and significance of certain SNPs. Additionally, the authors could test whether these SNPs interact with APOE in influencing AD risk.

Response: We agree with the reviewer that reporting quantitative genetic effect estimates would strengthen the interpretation of the GWAS-based supportive evidence. In the revised manuscript, we now explicitly report the effect sizes (β , as provided in the original GWAS) and corresponding p-values for coding variants in genes encoding key mediator proteins (Supplement Table. 9). In addition, we summarize the range of effect sizes and statistical significance in the Results section to contextualize the magnitude of these associations for both APOE4- and APOE2-related mediators: (page 21, lines 646-651): “Among the 8 key proteins mediating APOE4’s effect on AD diagnosis or A β pathology in at least two datasets (Fig. 6c), 5 (S100A13, SPC25, TBCA, LRRN1, and CTF1) had coding variants linked to A β 42 or AD ($\beta = -0.21 \sim 0.25$, p-value = 0.002 \sim 0.049). All 4 key APOE2 mediators (APOB, PCLAF, SNAP23, WARS2) identified in multiple datasets were also supported by SNP associations with A β 42 or AD ($\beta = -0.31 \sim 0.24$, p-value = $3.308 \times 10^{-5} \sim 0.049$). These results further substantiate their relevance to AD biology”. Note that the number of mediator proteins differs from the previous version because, to ensure comparability of A β definitions between ADNI and BioFINDER-2 (see response to reviewer #2 below), we now restricted A β classification in ADNI to CSF A β 42/40 ratio, which led to more replication on mediation results.

With respect to testing SNP \times APOE genotype interactions, we agree that such analyses would be informative. However, the GWAS results referenced in our study are derived from previously published large-scale GWAS meta-analysis, for which individual-level genotype data

are not accessible, precluding formal interaction testing using summary statistics. We also considered interaction analyses within our own cohorts; however, the available sample sizes are insufficient to provide adequate statistical power for robust testing of SNP × APOE interactions, particularly for low-frequency coding variants. To avoid overinterpretation of underpowered results, we did not pursue this analysis. This limitation has now been explicitly acknowledged in the Discussion (page 28, lines 893-895): “*In addition, GWAS-based supportive analyses relied on publicly available summary statistics, precluding formal testing of SNP * APOE genotype interactions*”.

The authors also supplemented their data with brain transcriptome and examined the spatial co-expression of APOE expression. This analysis demonstrates that certain proteins exhibit spatial expression patterns related to APOE. It would be interesting to know if the authors considered whether different APOE genotypes influence the transcriptional levels of these proteins at the RNA level.

Response: We thank the reviewer for this insightful suggestion. We agree that assessing whether APOE genotype is associated with transcriptional variation in APOE-associated mediator genes provides important complementary context for interpreting the proteomic findings.

In response to this comment, we extended our analyses to examine APOE genotype-associated differences in RNA expression using publicly available post-mortem human brain transcriptomic data from the ROSMAP cohort as described in the revised Methods “Bulk RNA sequencing data” section (pages 32-33, lines 1069-1084). Specifically, we analyzed bulk brain RNA sequencing data from 245 individuals with AD and 198 cognitively unimpaired participants, focusing on 76 mediator proteins identified in the SomaLogic analyses for which corresponding gene expression measurements were available.

As now described in the revised Results (page 22, lines 663-677): “*To directly assess whether APOE genotype influences transcriptional levels of these mediator proteins, we further analyzed bulk brain RNA sequencing data from the ROSMAP cohort (245 AD, 198 CU), focusing on 76 mediator proteins identified in SomaLogic with available gene expression measurements. APOE4 was associated with the expression of a limited subset of mediator genes, with nominal associations observed for RASSF2, UNG, SNAP23, CDA, STK10, and RHOG. Among these, SNAP23, CDA, and STK10 remained significantly associated with APOE4 after adjustment for AD diagnosis. In contrast, APOE2 was associated with the expression of several mediator genes, including WNT10B, PHB2, DMKN, NPY, HDAC6, and CDK5RAP3, with associations observed with or without adjustment for AD diagnosis (Supplement Table. 4). By comparison, AD diagnosis was associated with widespread transcriptional changes across mediator genes, predominantly showing reduced expression in AD. APOE expression itself was not associated with APOE genotype or AD status. Overall, these results indicate that APOE genotype contributes to transcriptional variation in a limited subset of mediator genes, whereas AD status represents the dominant source of transcriptional differences in post-mortem brain tissue*”. These results suggest that proteomic and transcriptomic readouts may capture distinct layers of APOE-related biology rather than directly corresponding molecular signals.

We note that these analyses remain exploratory due to variability in tissue sampling, and disease stage. Accordingly, we have framed these findings cautiously and emphasize that they provide supportive, but not definitive, evidence linking *APOE* genotype to transcriptional regulation of select mediator genes. As discussed further in the Discussion (page 27, lines 855-869): “*Consistent with this complexity, exploratory bulk transcriptomic analyses in post-mortem brain tissue revealed only limited APOE genotype-associated differences among mediator genes, whereas AD diagnosis was associated with more widespread transcriptional changes. This dissociation likely reflects multiple, non-mutually exclusive factors. First, the proteomic signatures identified here are derived from biofluids, which integrate signals across tissues, cell types, and regulatory processes, and may therefore capture systemic or post-transcriptional effects of APOE that are not directly mirrored at the level of bulk brain RNA expression. Second, protein abundance is shaped by additional layers of regulation beyond transcription, including translation efficiency, secretion, degradation, and compartment-specific dynamics⁶¹⁻⁶³, such that APOE-related effects on protein levels may occur independently of detectable changes in steady-state mRNA levels. Together, these considerations suggest that proteomic and transcriptomic measurements capture complementary, rather than interchangeable, layers of APOE-related biology across tissues and disease stages. Future studies integrating brain multi-omics with AD biomarkers across disease stages and functional validation may further distinguish central from peripheral origins and establish causal roles in AD risk*”.

Minor concerns:

1. On page 7, line 188, the authors mention "192 APOE4-associated proteins." This should probably refer to APOE2 instead of APOE4.

Response: We thank the reviewer for identifying this error. This has been corrected to “192 APOE2-associated proteins” in the revised manuscript (page 7, line 188).

2. The race and ethnicity bias remain insufficiently addressed. The authors should include them as a covariate to further enhance the validity of the study, or mention this as a limitation of study if full demographics are not available for all cohorts.

Response: We thank the reviewer for raising this important point. Where data were available, we adjusted genetic ancestry using principal components (PC1-5) as a proxy for population structure and found that *APOE*-protein associations and key mediation results were robust to this adjustment.

However, we acknowledge that most cohorts included in this study are predominantly of European ancestry, and that harmonized race and ethnicity information was not uniformly available across datasets. We have therefore explicitly noted this as a limitation in the Discussion, emphasizing that the generalizability of our findings to more diverse populations remains to be established in future studies (page 28, lines 885-891): “*Fourth, although sensitivity analyses were performed in BioFINDER-2 (e.g., adjusting for ancestry, vascular pathology and medication use), most cohorts included in this study were predominantly of European ancestry, and harmonized race and ethnicity information was not uniformly available*

across datasets. As a result, the generalizability of our findings to racially and ethnically diverse populations may be limited. Moreover, residual confounding from other factors (e.g., comorbidities, lifestyle) cannot be fully excluded”.

Reviewer #2:

As a disclaimer, I was reviewer of the previous version of this article submitted originally to Nature Medicine and my assessment reflects that fact. Lu et al. present a modified version of their study about the proteomic signature of APOE4 and APOE2 in the blood and the CSF of Alzheimer's disease patients. For this purpose, they bring together data collected from the GNPC, BioFINDER-2, ADNI, UKBB, and the PPMI studies. This effort brings together more than 10000 samples including both, plasma and CSF samples, studied using affinity-based platforms Somalogic (aptamers) and Olink (DNA-labeled antibodies). Either the presence of Aβ (detected either in CSF or PET) or a diagnosis of AD were considered as discriminating factors for APOE4 or APOE2 carriers in contrast to cognitively unimpaired subjects. An abundance analysis was conducted and associated to, or mediator proteins were identified for, each APOE haplotype. Key mediator proteins were identified for APOE4 such as ZW10, S100A13, SPC25, TBCA and ARL2, and for APOE2 such as APOB, PCLAF, SNAP23 and WARS2. Finally, co-expressors of APOE in brain tissue were identified among the proteins detected during the analysis, using publicly available RNA microarray data.

As in the previous version of the manuscript, the analysis is well conducted. The manuscript is clearly written, and the results are well presented. I do see differences in the data reported, perhaps influenced by some changes in number of cases taken from each study, and perhaps due to some modifications on the analysis. The document has gained on the description and interpretation of the findings and a more balanced presentation of ApoE haplotypes effects is now presented.

Response: We thank the reviewer for this careful observation. The modest differences in the reported results compared with the previous submission reflect analytical harmonization implemented to improve cross-cohort consistency and avoid potential sample overlap. These updates resulted in minor changes in sample size and effect estimates but did not alter the overall patterns or main conclusions of the study.

Some issues remain. Authors still use what they call “Aβ status” or AD diagnosis as absolute values to assess the impact of ApoE haplotypes in protein levels. This “Aβ status” stated as either positive or negative, either derives from CSF Aβ₄₂/Aβ₄₀ ratio in the BioFINDER-2 cohort patients, or from Aβ-PET status in the ADNI cohort. Up to which point CSF Aβ levels are compatible with Aβ-PET measurements to be comparable in their definition of Aβ positivity, remains to be clarified. Furthermore, both are continuous variables and similarly to the classification of “AD diagnosis”, include individuals at different stages of progression and severity of AD. This fact is not addressed in the analysis, both “Aβ status” and AD diagnosis are

considered as binary categorical variables. This might be beneficial from an analytical point of view, but it might not reflect the actual complexity of the disease. It might be worthwhile to assess them as the continuous variables they are and see how it affects the analysis.

Response: We thank the reviewer for raising these important and closely related concerns regarding the definition and modeling of A β pathology. Given the multiple aspects involved, we address them explicitly and sequentially below.

First, regarding the definition of A β status in ADNI and its comparability with BioFINDER-2. We acknowledge that the description of A β status in the original manuscript was imprecise and may have caused confusion. Upon careful re-examination, we confirm that A β status in the original implementation was defined using both A β -PET and CSF A β measurements. All ADNI participants included in A β -anchored analyses had CSF A β measurements available, whereas 248 individuals did not have A β -PET data. In the original implementation, PET-based A β status was used when available, while CSF-based status was used when PET data were missing; discordance between PET- and CSF-based classification was observed in only a small subset of individuals (N = 77, 12%). To eliminate this ambiguity and to maximize cross-cohort comparability, we have now harmonized the analysis by defining A β status in ADNI exclusively using the CSF A β 42/40 ratio, consistent with BioFINDER-2. Importantly, this refinement did not materially alter any results or conclusions. The Methods section specifies that A β status in ADNI has been revised accordingly to explicitly state this definition (page 30, line 978-979): “A β status was defined using CSF A β 42/40 ratio”.

Second, regarding the use of binary versus continuous measures of A β . We agree that continuous biomarkers can more fully capture the graded and dynamic nature of disease progression. Accordingly, we performed sensitivity analyses modeling A β as a continuous CSF A β 42/40 ratio. These analyses yielded results highly consistent with those obtained using binary A β status (page 16, lines 467-470): “Finally, when modeling A β as a continuous CSF A β 42/40 ratio instead of as a binary variable, 7 of the 8 upstream mediator proteins remained consistent (all except UBL3; Supplement Fig. 5F), supporting the robustness of these findings across A β definitions and potential confounders”.

Third, regarding the comparability of CSF A β and A β -PET measures and the use of continuous PET-based outcomes. In BioFINDER-2, CSF A β 42/40 ratio and A β -PET burden showed strong concordance at both the categorical and continuous levels (Supplement Fig. 5I). Specifically, 93% of individuals exhibited concordant A β classification based on CSF A β 42/40 status and A β -PET status, and CSF A β 42/40 ratio was strongly correlated with A β -PET burden (Spearman $r = -0.72$, $P = 3.3 \times 10^{-184}$). These results indicate that CSF A β and A β -PET are largely compatible in this cohort.

To directly assess whether APOE-associated upstream mediation effects remain detectable when amyloid pathology is modeled using a continuous PET-based measure, in the revised version, we performed sensitivity analyses using continuous A β -PET burden. As described in the Results (page 16, lines 470-480): “We further extended these sensitivity analyses by modeling amyloid burden using continuous A β -PET measures in the subset of individuals with available PET imaging (N=1,147; 23 AD, 829 CU, and 295 MCI). In contrast to CSF-based analyses, no upstream mediator proteins remained statistically significant when A β -PET burden was treated as a continuous outcome (Supplement Fig. 5H). This occurred despite

the strong concordance between CSF A β 42/40 ratio and A β -PET measures at both the categorical and continuous levels (Supplement Fig. 5I). Notably, the PET sample was primarily enriched for CU and MCI individuals, in whom A β -PET values clustered within a relatively narrow range, particularly at early stages of amyloid accumulation. This restricted dynamic range likely reduces statistical sensitivity to detect upstream mediation effects when A β -PET is modeled as a continuous outcome”.

We also emphasize this as a limitation in the revised Discussion (page 28, lines 880-885): “*Third, AD diagnosis and amyloid status were primarily modeled as binary variables, which may not fully capture the continuous nature of disease progression and amyloid accumulation. Although sensitivity analyses using the continuous CSF A β 42/40 ratio largely supported the robustness of our findings, analyses based on continuous A β -PET measures were limited by a restricted dynamic range in predominantly preclinical samples”.*

In summary, by harmonizing A β definitions across cohorts using the CSF A β 42/40 ratio and by performing complementary sensitivity analyses using continuous CSF A β and continuous A β -PET measures, we have directly addressed concerns regarding A β definition, dichotomization, and disease-stage heterogeneity. Together, these analyses indicate that our main conclusions are not driven by the use of binary A β status, while also highlighting that the detectability of upstream mediation depends on the biomarker modality and the disease-stage distribution captured by continuous measures.

Another issue is the lack of validation of the findings using Olink and Somalogic, beyond their internal validation. A low correlation between both approaches and MS proteomics has been previously reported (10.1101/2025.02.14.638375). In this version of the manuscript the authors mention this issue as one out of several limitations of the study. It is clear for this reviewer that a full orthogonal validation, hopefully including actual MS proteomics, of a study of this scale and incorporating data from several cohorts, is impractical. But the authors have strong interesting claims about the differences between ApoE haplotypes that can be validated in at least in a subset of individuals of the cohort more accessible for them. I invite them to consider doing this.

Response: We thank the reviewer for this constructive suggestion and agree that orthogonal validation across proteomic platforms is important for interpreting large-scale proteomic findings. We also appreciate the reviewer’s recognition that comprehensive orthogonal validation across multiple cohorts and technologies is challenging in studies of this scale.

We carefully evaluated the feasibility of additional orthogonal validation within cohorts included in this study. While mass spectrometry-based proteomic data are available for the GNPC cohort, as described in Imam et al. 2025⁴, corresponding individual-level genotype information required for *APOE*-stratified analyses is not available in this sub-cohort, precluding direct validation of *APOE* haplotype-specific effects using MS proteomics.

Within the scope of available data, we incorporated orthogonal validation using an independent, untargeted mass spectrometry-based CSF proteomics dataset from ADNI. As described in the Methods under “*ADNI CSF TMT-based mass spectrometry proteomics*” section (page 32, lines 1039-1067), we analyzed CSF proteomic profiles measured by tandem mass tag-based mass spectrometry (TMT-MS) from the Emory University ADNI CSF TMT-MS dataset. After batch effect correction, quality control, restriction to individuals with available CSF

A β measurements, and exclusion of $\epsilon 2/\epsilon 4$ carriers and APOE-related entries, the final analysis included 536 participants (253 A β - and 283 A β +), with 2,024 proteins retained for downstream analyses. Importantly, this subset overlapped with the ADNI CSF SomaLogic cohort, enabling within-individual cross-platform comparisons.

Correspondingly, in the Results section (page 19, line 549: “Assessing APOE-signature in other proteomics platforms”), we explicitly evaluated APOE-associated effects in this orthogonal TMT-MS dataset (page 19, lines 554-559): “In this dataset, widespread CSF protein abundance changes were associated with APOE4 and A β , with effect estimates remaining broadly concordant before and after adjustment for each other (Supplement Fig. 7A-B). Consistent with other cohorts, most APOE4-associated CSF protein changes were more strongly mediated by A β ; nevertheless, three upstream mediator proteins were identified, among which YWHAZ showed a partial mediation proportion of up to 38% (Supplement Fig. 7C)”, providing independent support for APOE-associated CSF alterations at a systems level.

At the same time, direct replication of individual mediator proteins across platforms was constrained by limited target overlap and protein-level quality-control filtering. As reported in the manuscript (page 19, lines 559-569): “Most mediator proteins identified in SomaLogic analyses (including SPC25, ZW10, ARL2 etc.) were either not targeted by the TMT-MS or were excluded during quality-control filtering, limiting direct cross-platform comparison (Supplement Fig. 3_3A-B). Among overlapping proteins, 57 were significantly associated with APOE4 in both ADNI cohorts. While most showed concordant effect directions, a few exhibited opposite directions across platforms, exemplified by S100A13 and TBCA (Supplement Fig. 3_2A); direct comparisons of S100A13 and TBCA levels revealed negative correlations between SomaLogic and TMT-MS measurements (Supplement Fig. 8A). Consistent with this, pairwise comparisons across all shared proteins indicated generally low cross-platform concordance, with 54.1% of proteins showing Spearman correlations between -0.3 and 0.3 (Supplement Fig. 8B-D)”.

Together, these analyses indicate that orthogonal MS-based data support APOE-associated CSF proteomic alterations at a global level, while platform-specific differences in coverage and measurement principles limit direct replication of all individual proteins. We therefore interpret platform heterogeneity as an inherent feature of large-scale proteomic integration, motivating our emphasis on within-platform replication, systematic cross-platform comparison, and complementary genetic evidence, as discussed in the revised manuscript (Discussion, page 27, lines 833-840): “Together, these findings suggest that the observed heterogeneity in APOE-associated proteomic effects is primarily shaped by differences in assay coverage, detection sensitivity, and the platform-dependent capture of distinct molecular features. Accordingly, proteomic associations, particularly those that are platform-specific, should be interpreted with caution in a context-aware manner that accounts for assay design, biological matrix, and disease anchoring. Future peptide- or isoform-resolved proteomic analyses may therefore be required to further clarify how APOE genotype and amyloid pathology differentially shape those proteins across biofluids and disease stages”.

Finally, also acknowledged by the authors as a limitation, proteins identified in this analysis can have different biological sources at a cellular level and reflect various aspects of the pathology, directly or indirectly influenced by a particular ApoE haplotype. So, even though the study provides a very descriptive picture of the apparent effect of ApoE2 and ApoE4 haplotypes in

plasma and CSF proteomics, the absence of mechanistic supportive evidence renders the biological significance of the findings incomplete. Therefore, despite the scale of the study and the complexity of the analysis, I see the contribution to the field as only incremental.

Response: We appreciate the reviewer's thoughtful assessment and agree that proteins identified in large-scale proteomic analyses may originate from diverse cellular sources and reflect multiple biological processes, some of which are indirectly influenced by *APOE* genotype or downstream pathology.

We also agree that the absence of direct mechanistic experimentation represents an important limitation, which we explicitly acknowledged and discussed in the Discussion (page 27, lines 841-869): "*Beyond platform heterogeneity, an additional interpretative challenge concerns the tissue origin of APOE-associated proteomic changes..... Future studies integrating brain multi-omics with AD biomarkers across disease stages and functional validation may further distinguish central from peripheral origins and establish causal roles in AD risk*".

However, we respectfully emphasize that the primary goal of this study was to define the system-level architecture of *APOE2*- and *APOE4*-associated proteomic alterations across biofluids, disease stages, and analytical frameworks, thereby contextualizing proteomic alterations associated with *APOE2* and *APOE4* in relation to their divergent AD risk in humans. Within this scope, our contribution extends beyond a descriptive catalog of associations. By integrating multi-cohort, multi-platform proteomics with amyloid-anchored mediation analyses, genetic support from AD GWAS, and brain transcriptomic context, we provide a structured framework to distinguish genotype-driven upstream alterations from pathology-driven downstream remodeling—an aspect that has not been systematically addressed in prior *APOE* proteomic studies.

Importantly, the identification of allele-specific upstream mediators detectable in A β -negative individuals, together with asymmetric regulatory patterns between *APOE2* and *APOE4*, offers testable hypotheses and prioritizes candidate pathways for future mechanistic investigation. In this sense, we view the study not as an endpoint, but as a foundational resource that delineates the boundaries of robust *APOE*-associated signals and informs the design of targeted experimental and translational follow-up studies.

Reviewer #3:

Most of the points raised in my earlier review have now been addressed to the extent permitted by the available datasets, and the authors have substantially strengthened the manuscript. I appreciate the clarifications added regarding platform heterogeneity, cross-fluid inconsistencies, tissue specificity, and causal interpretation, as well as the improved framing of the conclusions. Overall, the revisions meaningfully enhance the rigor and transparency of the work, and I congratulate the authors on their thorough and thoughtful responses.

Response: We thank the reviewer for the careful re-evaluation of our revised manuscript and for the positive assessment. We are grateful for the reviewer's constructive feedback, which has helped improve the clarity and balance of the manuscript.

I have only a few remaining suggestions, all of them minor but important for clarity:

- Lines 363–366: I do not fully understand the statement “These asymmetric effects may help explain the divergent AD risks of the two alleles,” particularly because two sentences earlier it is stated that “only a minority showed opposite regulation.” Please clarify the logic and further elaborate on how asymmetry, despite limited opposite regulation, supports divergent risk profiles. This is central to the biological interpretation. It would benefit from substantial expansion in the discussions to help the reader appreciate its implications.

Response: We thank the reviewer for pointing out this lack of clarity. We agree that the original statement appeared abrupt and insufficiently motivated by the preceding text. To address this, we revised the Results to more explicitly articulate the logic underlying the observed asymmetry (page 13, lines 371-375): *“These findings indicate that APOE2 and APOE4 do not simply act as mirror images by regulating the same proteins in opposite directions. Instead, their proteomic effects are shaped by distinct upstream mediator architectures and differences in regulatory strength across shared networks, giving rise to asymmetric proteomic patterns that may underlie their divergent AD risk”*.

- In the CSF section (lines 498 and following), a description of overlap or lack of overlap with plasma findings is missing; adding this would strengthen interpretability. A related question: “SPC25 mediated APOE4–AD associations in plasma but not in CSF.” Some discussion on how to interpret this divergence would be valuable.

Response: We thank the reviewer for this helpful suggestion. In response, we have revised the Results section to explicitly describe the overlap and lack of overlap between plasma and CSF findings (Results, page 18, lines 526-530): *“93 APOE2 and 684 APOE4 associated proteins were identified. Although only a subset of plasma-identified proteins reached statistical significance in CSF (71 APOE4 and 19 APOE2 associated proteins), the directions of APOE effects (both with and without adjustment for AD diagnosis or A β) were largely concordant between the CSF and plasma SomaLogic datasets (Supplement Fig. 3_2A-D)”*.

We also directly address the reviewer's question regarding SPC25 in the revised Results (page 19, lines 542-546): *“In contrast, SPC25 mediated APOE4-AD associations in plasma but not in CSF, indicating compartment-specific differences in mediation patterns. This divergence may partly reflect contextual modulation in plasma, as SPC25 no longer mediated APOE4-A β associations after adjustment for ChEIs uses, while also being consistent with differences in tissue origin and regulatory context between plasma and CSF”*.

Together, these additions are intended to strengthen interpretability by explicitly situating CSF findings in relation to plasma results and by clarifying that observed divergences reflect biofluid-specific and context-dependent regulation.

- The statement that “Together, these results reflect a molecular signature of OLINK that is distinct from SomaLogic in CSF” has major implications and should be more explicitly discussed, particularly in light of the relatively low number of proteins identified in UK Biobank.

Response: We thank the reviewer for this important comment and agree that the original statement could be interpreted too broadly, particularly given differences in target coverage and sample composition across cohorts. To avoid overgeneralization, we have removed this sentence from the Results section.

Instead, we revised the Results to provide a more explicit, data-driven comparison of platform- and biofluid-specific APOE-associated effects, focusing on effect directions and measurement concordance. As now described in the revised manuscript (page 20, lines 583-592): “Notably, several key APOE4 mediator proteins identified in plasma SomaLogic analyses, including S100A13, TBCA, NEFL, ST8SIA1, and LRRN1, showed opposite effect directions in CSF OLINK analyses (Supplement Fig. 3_2A, C). Direct comparisons revealed weak or absent correlations between plasma SomaLogic and CSF OLINK measurements for these proteins (Supplement Fig. 10A-B). Even within the SomaLogic platform, correlations between independent aptamers targeting the same protein were weak for ST8SIA1 and LRRN1, underscoring assay- and target-dependent measurement variability (Supplement Fig. 10B). Consistent with this, pairwise comparisons across all 1,169 shared proteins showed that 88.5% exhibited weak cross-fluid correlations (Spearman’s r between -0.3 and 0.3 ; Supplement Fig. 10C-E), highlighting substantial proteomic heterogeneity across biofluids and platforms”.

In response to the reviewer’s concern regarding the relatively limited protein coverage in UK Biobank (we note that the UK Biobank analyses were updated following the annual withdrawal of seven participants, but this update did not materially affect any results or conclusions reported), we further clarified that reduced cross-biofluid concordance within the OLINK platform likely reflects both cohort composition and platform-specific design. As described in the revised Results (page 20, lines 603-608): “At the same time, global cross-biofluid concordance of APOE-associated effects within the OLINK platform was more limited than that observed in SomaLogic-based plasma-CSF comparisons. This likely reflects differences in cohort composition between the largely younger UKBB population and the disease-enriched BioFINDER-2 cohort, as well as platform-specific target coverage (Supplement Fig. 3_1A). In addition, higher assay variability reported for OLINK may further contribute to reduced concordance across biofluids⁵⁻⁸”.

- Similarly, the NEFL discrepancies across platforms (“positively correlated in CSF... negatively associated in plasma SomaLogic... not replicated in OLINK or Simoa”) highlight a broader issue: to what extent should proteomic associations, especially platform-specific ones, be interpreted with caution? Further reflection on this point would strengthen the discussion.

Response: We thank the reviewer for raising this important point regarding the interpretation of platform-specific proteomic associations. We agree that the observed discrepancies for NEFL across biofluids and platforms underscore a broader methodological issue, namely the need for cautious, context-aware interpretation of proteomic findings, particularly when associations appear platform-specific. To address this, we revised both the Results and Discussion to

explicitly frame NEFL as an illustrative example of how *APOE*- and amyloid-associated effects can manifest differently depending on biological compartment and measurement technology, rather than as evidence of conflicting biology.

Specifically, in the revised Results under “*Comparison of NEFL across assays and datasets*” section (pages 20-21, lines 613-636), we now present a structured, data-driven comparison of NEFL associations across platforms, distinguishing *APOE*-driven from A β -driven effects. Briefly, NEFL showed consistent positive associations with A β pathology or AD diagnosis in CSF across platforms, whereas in plasma, *APOE* genotype remained a significant influence, most prominently detected using SomaLogic. These findings highlight that the relative contribution of *APOE* genotype and amyloid pathology to NEFL levels differs across biofluids and is differentially captured by distinct assay technologies.

In the revised Discussion (pages 26-27, lines 791-840), we explicitly emphasize that the limited cross-platform concordance primarily reflects differences in assay design, target coverage, detection sensitivity, and biological context, rather than technical error. We further use NEFL as a representative example to illustrate how individual protein signals can differ across platforms without implying underlying biological inconsistency.

Firstly, we now explicitly emphasize that differences in target coverage across proteomic technologies inherently limit strict cross-platform comparability (page 26, lines 791-803): “*A key feature of our findings is the substantial heterogeneity of APOE-associated proteomic signatures across measurement platforms and biological matrices.....SomaLogic provides broader target coverage, whereas OLINK panels and TMT-MS interrogate a more restricted set of proteins, resulting in a markedly smaller overlap of directly comparable targets. Consequently, many mediator proteins prioritized in SomaLogic analyses were not detectable or not reliably quantified in orthogonal datasets, including both OLINK and TMT-MS, largely due to differences in target selection, dynamic range, and detection rates, thereby limiting direct cross-platform comparability*”.

Importantly, we clarify that discordant *APOE*-associated effects across platforms do not necessarily reflect irreproducible findings. Rather, they highlight a general challenge in large-scale proteomics, whereby even direct comparison on measurements may show weak or inverse correlations across technologies. Such discrepancies do not imply technical failure of any individual platform; instead, they reflect differences in assay design, target recognition, and the specific molecular features captured by each technology (page 26, lines 804-814): “*Even among overlapping proteins (e.g., S100A13 and TBCA), cross-platform concordance was modest when directly comparing measurements across technologies (e.g., OLINK vs. SomaLogic and TMT-MS vs. SomaLogic)..... Heterogeneity was also evident within SomaLogic itself, where proteins quantified by multiple aptamers showed only modest correlations (e.g., ST8SIA1 and LRRN1), highlighting the importance of target recognition and assay-specific binding properties in shaping observed proteomic signals. Notably, for certain proteins such as TBCA and NEFL, statistically significant associations with APOE4 were detected across platforms but with opposite effect directions, indicating that these assays may be sensitive to different molecular features or biological contexts rather than capturing directly comparable signals*”.

Using NEFL as an illustrative example, we emphasize that platform-specific associations may capture different components of the same underlying biological process and therefore

should not be interpreted as directly interchangeable across platforms. In this context, NEFL demonstrates how *APOE*- and amyloid-associated effects can vary across biofluids and technologies without implying biological inconsistency (pages 26-27, lines 815-832): “*Within the broader context of platform heterogeneity, NEFL provides a particularly informative example of how disease context and genotype can jointly shape proteomic signals across biofluids and measurement technologies, without invoking technical artefacts. In our analyses, we observed APOE-driven reductions in NEFL levels, even among A β + individuals and patients with AD, most prominently in plasma SomaLogic measurements. Importantly, prior studies have shown that plasma NEFL measured by SomaLogic is elevated across multiple non-AD neurodegenerative diseases, including Parkinson’s disease, frontotemporal dementia, and amyotrophic lateral sclerosis, supporting its validity as a general marker of neuronal injury⁴. The opposing direction observed in AD therefore does not indicate reduced assay sensitivity. Rather, it suggests that in the AD context, where APOE4 carriers are highly enriched, genotype-driven reduction of NEFL may dominate over, or suppress, the more generic injury-related increases commonly observed across other neurodegenerative conditions. This interpretation is further supported by biofluid-specific patterns. In CSF, NEFL associations were largely dominated by amyloid-related pathology across platforms, whereas APOE-related effects were more readily detectable in plasma. Notably, APOE-associated reductions in NEFL remained detectable in SomaLogic-based CSF measurements, indicating that aptamer-based detection can retain sensitivity to genotype-associated NEFL signals even within a biological matrix strongly influenced by A β pathology*”. We note that other GNPC⁴ study have reported increased plasma NEFL measured by SomaLogic in non-AD neurodegenerative diseases (e.g., PD, FTD, ALS, in their Supplementary Table 5), supporting that SomaLogic plasma NEFL in general has properties as an indicator of neuronal injury. However, as clarified in our study, this effect is overcome by *APOE*-specific modulation of NEFL levels.

Finally, we provide a unifying interpretation (page 27, lines 833-840): “*Together, these findings suggest that the observed heterogeneity in APOE-associated proteomic effects is primarily shaped by differences in assay coverage, detection sensitivity, and the platform-dependent capture of distinct molecular features. Accordingly, proteomic associations, particularly those that are platform-specific, should be interpreted with caution in a context-aware manner that accounts for assay design, biological matrix, and disease anchoring. Future peptide- or isoform-resolved proteomic analyses may therefore be required to further clarify how APOE genotype and amyloid pathology differentially shape those proteins across biofluids and disease stages*”.

Together, these revisions clarify that proteomic associations, particularly those that are platform-specific, should be interpreted with caution and within the context of assay characteristics, biological matrix, and disease anchoring. We now explicitly frame platform heterogeneity as an inherent feature of large-scale proteomic integration and provide guidance on how such findings should be interpreted, thereby strengthening the conceptual interpretation of our results.

- In the section on genetic evidence supporting mediator proteins, it would be helpful to add the strength of association (effect sizes or p-values) shown in Figure 6c to contextualize the relevance of the findings.

Response: We thank the reviewer for this helpful suggestion. We agree that reporting the strength of genetic associations is important for contextualizing the relevance of these findings. Owing to the large number of SNPs evaluated in the genetic analyses shown in Fig. 6c, it was not feasible to list individual effect sizes and p-values for all variants in the main text.

To address this, we have added quantitative information on association strength in the revised manuscript by (i) explicitly reporting representative effect size ranges and significance levels in the revised Results (page 21, lines 646-651): “Among the 8 key proteins mediating APOE4’s effect on AD diagnosis or A β pathology in at least two datasets (Fig. 6c), 5 (S100A13, SPC25, TBCA, LRRN1, and CTF1) had coding variants linked to A β 42 or AD ($\beta = -0.21 \sim 0.25$, p -value = 0.002 \sim 0.049). All 4 key APOE2 mediators (APOB, PCLAF, SNAP23, WARS2) identified in multiple datasets were also supported by SNP associations with A β 42 or AD ($\beta = -0.31 \sim 0.24$, p -value = $3.308 \times 10^{-5} \sim 0.049$)”, and (ii) providing the full set of SNP-level statistics in Supplement Table. 9, as specified in the legend of Fig. 6 (page 23, lines 688-690): “c. The table shows the number of SNPs in the coding gene of key mediators (identified as mediators in at least 2 dataset) that are associated with AD clinical diagnosis or CSF A β 42 level in external GWAS studies. Supplement Table. 9 provides a detailed statistical summary for each SNP”. This approach allows readers to assess the magnitude and statistical support of genetic evidence while maintaining readability of the main text.

- For the reasons noted above, the last sentence in the first paragraph of the Discussion might be nuanced, acknowledging that the support is variable across platforms, tissues, and datasets.

Response: We agree with the reviewer and have revised the last sentence of the first paragraph of the Discussion to provide a more nuanced interpretation. Specifically, we added a clarifying sentence (page 24, lines 712-717): “Our multi-cohort design further revealed that support for individual APOE-associated proteins varied across platforms, tissues, and datasets; however, several key plasma proteins were supported by one or more complementary lines of evidence, including CSF SomaLogic datasets, allele-dominant effects, AD GWAS evidence, and transcriptomic co-expression with APOE in the human brain, and remained robust after adjusting for ancestry, vascular pathology, and medication use”.

This revision explicitly acknowledges variability in support across platforms, tissues, and datasets, while clarifying the basis on which prioritized APOE-associated proteins are discussed.

- The finding that ChEI use was linked to higher SPC25 levels only in APOE4 carriers is particularly important. Given that many centers prescribe ChEIs in MCI based on the positive APOE4 subgroup signal in the pivotal trials (which lacked biomarkers), this result raises the question of whether the previous signal reflected AD-related biology or simply APOE-driven effects. This deserves further discussion.

Response: We thank the reviewer for highlighting this important point and agree that our findings raise a relevant interpretative question regarding the biological basis of previously

reported *APOE4* subgroup signals in ChEI trials. Distinguishing *APOE*-driven molecular effects from secondary to AD pathology is a central motivation of our study.

In response to this comment, we revised the Discussion to clarify that *APOE4*-specific molecular responses to ChEI treatment reported previously may reflect genotype-driven effects that are not directly linked to downstream AD disease biology (pages 24-25, lines 738-745): *“Given that SPC25 is not known to represent a protective or compensatory response in the context of neurodegeneration, this finding does not necessarily indicate therapeutic benefit. Instead, it raises the possibility that APOE4-specific molecular responses to ChEI treatment, reported previously as differential treatment responsiveness in biomarker-unanchored trials, may reflect genotype-driven effects that are not directly linked to AD disease biology. This observation highlights the importance of distinguishing genetic-driven molecular effects from biological changes secondary to disease pathology”*.

Overall, these are relatively minor clarifications in an otherwise strong and much-improved manuscript.

Reviewer #4:

The *APOE* locus is the strongest genetic factor for Alzheimer’s disease (AD), with $\epsilon 4$ (*APOE4*) increasing and $\epsilon 2$ (*APOE2*) decreasing risk, yet the molecular programs reflecting these divergent effects remain unclear. In this study, the authors conducted a large-scale, multi-cohort proteomic analysis across plasma and cerebrospinal fluid (CSF), integrating five independent datasets including the GNPC, BioFINDER-2, ADNI, UKBB, and PPMI, and two affinity-based platforms (SomaLogic, OLINK).

Utilizing amyloid β ($A\beta$) or clinical AD anchored upstream (*APOE* => protein => $A\beta$ /AD) and downstream (*APOE* => $A\beta$ /AD => protein) mediation framework, multi-cohort design, and verification via GWAS and brain transcriptomics, the authors systematically mapped *APOE* isoform-specific proteomic signatures. They found that proteomic alterations linked to *APOE2* and *APOE4* were largely detectable in preclinical stages, even before detectable $A\beta$ pathology, and remained relatively stable across aging and disease stages.

In *APOE2* carriers, most dysregulated proteins were involved in DNA repair, mitochondrial metabolism, proteostasis, and anti-inflammatory regulation, functioning largely as upstream mediators; only a minority showed limited remodeling by pathological cascades, consistent with systemic resilience. In *APOE4* carriers, a small subset of proteins enriched in cell-cycle and oligodendrocyte precursor cells (OPCs) may help explain its association with AD risk, whereas a broader set linked to vascular remodeling, immune activation, and proteostasis loss underwent pronounced downstream pathology-driven remodeling, reflecting a shift from genotype-related to pathology-driven regulation.

Their comparative analyses identified three key contributors reflecting divergent AD risk: allele-specific mediators supported by genetic or transcriptomic evidence (e.g., SPC25 for APOE4 , APOB and SNAP23 for APOE2), oppositely regulated “switch-node” proteins (e.g., VPS29, PHGDH), and a broader set of concordantly regulated proteins showing ϵ 4-dominant effects (e.g., S100A13, TBCA, ARL2).

A key finding of this comprehensive study is that the alteration of these proteins were already evident in A β -negative individuals, underscoring their upstream, genotype-driven nature. Collectively, their findings indicate that asymmetric molecular architectures, rather than mirror-image regulation, underlie the divergent AD risks of APOE2 and APOE4 , and nominate upstream pathways and genetically or transcriptomically supported proteins as promising biomarkers and potential targets for allele-tailored interventions in AD.

Overall, this is an outstanding study with excellent study design, multi- and complementary human cohorts, multi-platforms of samples, and various analytic approaches. The data strongly support the main conclusions. Their important findings point to upstream and allele-specific proteins already altered in A β -negative individuals as promising biomarkers and targets for early detection and preventive interventions.

Response: We thank the reviewer for the very positive and thoughtful evaluation of our work, and for the constructive suggestions to further strengthen the manuscript. We address each point below.

This reviewer only has the following minor comments to help make the paper stronger and more solid.

1) Although it is not necessary, it would be nice to see whether there are any sex effects on the major altered proteins in both APOE2 and APOE4 carriers, especially the mediators of APOE2 and APOE4 in Abeta-negative individuals.

Response: We thank the reviewer for this helpful suggestion. In response, we performed additional analyses stratified by sex in A β - individuals for major *APOE2*- and *APOE4*-associated proteins in the BioFINDER-2 plasma SomaLogic cohort, with a particular focus on prioritized proteins already altered at A β - stages, including mediator proteins. We additionally tested *APOE* * sex interaction effects for these proteins.

As described in the revised Results (page 15, lines 431-434): “Moreover, sex-stratified analyses in A β - individuals demonstrated highly concordant *APOE*-associated effect sizes between males and females for these early-altered proteins, and *APOE* * sex interaction analyses did not identify sex-dependent modification (Supplement Fig. 4C, D, Supplement Table. 10)”, we did not observe evidence for sex-specific modification of *APOE* effects for these key proteins at the A β -negative stage. *APOE* effect sizes were highly comparable between males and females, with strong correlations between sex-stratified estimates (Spearman’s $r = 0.87$ for *APOE4* and $r = 0.89$ for *APOE2*, Supplement Fig. 4C, D), indicating largely concordant *APOE*-associated proteomic effects across sexes among A β -negative individuals.

2) For major analyses, the statistical methods and n numbers should be provided in each figure legend.

Response: We agree with the reviewer and have revised all major figure legends (Figs. 2-6) to explicitly report the statistical methods used and the number of participants included in each analysis.

We note that although proteins with high missingness were excluded according to predefined quality-control criteria (as described in the Method section) prior to model fitting, retained proteins could still contain sporadic missing values. Handling of these missing observations and extreme outliers during model fitting led to minor variation in the effective sample size across proteins. Consequently, a single uniform sample size is not applicable to all protein-level analyses. To ensure transparency and reproducibility, we therefore report model-level information in Supplement_Table_2 and Supplement_Table_8, including standard errors and residual degrees of freedom for each protein and statistical model, from which effective sample sizes can be inferred if needed. Together, these revisions collectively clarify the analytical framework and sample size upon which major analyses are based.

3) Line 188, should it be “192 APOE2-associated proteins”?

Response: We thank the reviewer for catching this error. The text has been corrected to “192 APOE2-associated proteins” in the revised manuscript (page 7, line 188).

1. Dong, J. M. & Zhong, H. Systematic review: Proteomics-driven multi-omics integration for Alzheimer’s disease pathology and precision medicine. *Neurol. Int.* **17**, 197 (2025).
2. Ren, P. *et al.* Atlas of Proteomic signatures of brain structure and its links to brain disorders. *Nat. Commun.* **16**, 5092 (2025).
3. Zhang, J.-G. *et al.* Identify gene expression pattern change at transcriptional and post-transcriptional levels. *Transcription* **10**, 137–146 (2019).
4. Imam, F. *et al.* The Global Neurodegeneration Proteomics Consortium: biomarker and drug target discovery for common neurodegenerative diseases and aging. *Nat. Med.* **31**, 2556–2566 (2025).
5. Rooney, M. R. *et al.* Plasma proteomic comparisons change as coverage expands for SomaLogic and Olink. *medRxiv* 2024.07.11.24310161 (2024).
6. Katz, D. H. *et al.* Proteomic profiling platforms head to head: Leveraging genetics and

clinical traits to compare aptamer- and antibody-based methods. *Sci. Adv.* **8**, eabm5164 (2022).

7. Eldjarn, G. H. *et al.* Large-scale plasma proteomics comparisons through genetics and disease associations. *Nature* **622**, 348–358 (2023).
8. Kirsher, D. Y. *et al.* Current landscape of plasma proteomics from technical innovations to biological insights and biomarker discovery. *Commun. Chem.* **8**, 279 (2025).

Response to reviewers

Reviewer #1:

Overall, the authors have largely addressed the reviewer's concerns and provided a detailed description of cross-platform validation across proteomic technologies (Olink, SomaLogic, and TMT-MS). The inclusion of additional cohort data (ROSMAP), the accompanying and other related analyses are comprehensive. The manuscript is clearly written and highlights an interesting asymmetric protein profile between APOE2 and APOE4 effects. The authors have also carefully discussed the limitations of the study.

Nevertheless, platform heterogeneity remains a potential key limitation, as it may hinder the reproducibility of certain protein-level findings across technologies and complicate subsequent functional interpretation using complementary experimental models. In addition, APOE4-related proteomic signatures have been reported in several prior studies, which may limit the novelty of this work to a certain extent.

Minor concerns:

In several cohorts, the CSF A β 42/40 ratio is binary (A β -/A β +) and used for both discovery and validation. Please briefly clarify the exact thresholds applied in each cohort, the source/rationale for these cutoffs, and whether they were validated or calibrated against A β -PET or other biomarker analysis. Please also include relevant references accordingly.

Response: We thank the reviewer for raising this important point. In the BioFINDER-2 cohort, A β status was determined using the CSF A β 42/A β 40 ratio with assay-specific thresholds depending on the assay platform used to quantify CSF biomarkers. For the majority of participants (85% in the CSF OLINK dataset and 90% in the plasma SomaLogic dataset), CSF A β 42 and A β 40 were measured using the Roche Elecsys platform, and A β positivity was defined using the previously established cutoff of 0.080¹. When Roche measurements were unavailable, Lumipulse G (cutoff 0.072²) or Meso Scale Discovery (MSD; cutoff <0.077 determined using mixture modeling) assays were used. These definitions were applied consistently across datasets derived from the same BioFINDER-2 cohort.

In the ADNI cohort, CSF A β status was defined using CSF A β 42 concentrations measured with the Roche Elecsys assay. A β positivity was determined using the established ADNI cutoff of 880 pg/mL for CSF A β 42, which has previously been validated against amyloid-PET³ and shown to have high concordance with PET-defined amyloid status.

We note that the description of A β status in the ADNI cohort in the previous version of the manuscript was not sufficiently detailed. We have now clarified the assay platforms and corresponding thresholds in the Methods section and provided the appropriate references. This clarification does not affect any analyses or results.

Reviewer #4:

The authors have addressed very well all of my comments. I do not have any further comments.

Response: We thank the reviewer for the positive and encouraging feedback. We are pleased that our revisions have addressed the reviewer's concerns.

1. Quadalti, C. *et al.* Clinical effects of Lewy body pathology in cognitively impaired individuals. *Nat. Med.* **29**, 1964–1970 (2023).
2. Gobom, J. *et al.* Validation of the LUMIPULSE automated immunoassay for the measurement of core AD biomarkers in cerebrospinal fluid. *Clin. Chem. Lab. Med.* **60**, 207–219 (2022).
3. Hansson, O. *et al.* CSF biomarkers of Alzheimer's disease concord with amyloid- β PET and predict clinical progression: A study of fully automated immunoassays in BioFINDER and ADNI cohorts. *Alzheimers. Dement.* **14**, 1470–1481 (2018).